# Solving Zero-Sum Convex Markov Games

**Fivos Kalogiannis** [1]  **Emmanouil-Vasileios Vlatakis-Gkaragkounis** [2]  **Ian Gemp** [3]  **Georgios Piliouras** [3]

## Abstract

We contribute the first provable guarantees of global convergence to Nash equilibria (NE) in two-player zero-sum convex Markov games (cMGs) by using independent policy gradient methods. Convex Markov games, recently defined by Gemp et al. (2024), extend Markov decision processes to multi-agent settings with preferences that are convex over occupancy measures, offering a broad framework for modeling generic strategic interactions. However, even the fundamental min-max case of cMGs presents significant challenges, including inherent nonconvexity, the absence of Bellman consistency, and the complexity of the infinite horizon. Our results follow a two-step approach. First, leveraging properties of hidden-convex–hidden-concave functions, we show that a simple nonconvex regularization transforms the min-max optimization problem into a nonconvex–proximal Polyak-Łojasiewicz (NC-pPL) objective. Crucially, this regularization can stabilize the iterates of independent policy gradient methods and ultimately lead them to converge to equilibria. Second, building on this reduction, we address the general constrained min-max problems under NC-pPL and two-sided pPL conditions, providing the first global convergence guarantees for stochastic nested and alternating gradient descent-ascent methods, which we believe may be of independent interest.

## 1. Introduction

The field of multi-agent reinforcement learning (MARL)—often framed as Markov games (MGs) (Littman, 1994)—studies how multiple agents interact within a shared, dynamic environment to optimize their individual cumulative rewards (Silver et al., 2017; Lanctot et al., 2019). However, many real-world applications require a more expressive formulation of agent preferences that do not simply decompose additively over time (Puterman, 2014; Zahavy et al., 2021). To address this limitation, the emerging framework of convex Markov games (cMGs) (Gemp et al., 2024) has been proposed as a principled yet flexible model for capturing complex multi-agent interactions in dynamic environments. Unlike traditional MGs, cMGs allow for convex utility functions over the state-action occupancy measure, enabling a richer set of players' preferences that better reflect practical applications. In this context, the occupancy measure of each player reflects the frequency of visiting any particular state of their corresponding Markov decision processes (MDPs), over a potentially infinite time horizon.

In practice, cMGs are useful for modeling a variety of challenging multi-agent settings, including:

(i) Fostering *creativity* in machine gameplay, such as discovering novel strategies in chess (Zahavy et al., 2022; 2023; Bakhtin et al., 2022), (ii) multi-step language model alignment (Wu et al., 2025), (iii) enhancing multi-agent *exploration* in robotic systems (Burgard et al., 2000; Rogers et al., 2013; Tan et al., 2022), (iv) ensuring *safety* in autonomous driving (Shalev-Shwartz et al., 2016), (v) enabling *imitation learning* from expert demonstrations, and (vi) promoting *robustness* and *fairness*, in multi-agent decision-making (Hughes et al., 2018). While the former (i-iii) are direct instantiations of cMGs, the latter (iv-vi) will profit when from a cMG formulation.

In general, the authoritative desirable outcome of a multi-agent scenario is some sort of game theoretic *equilibrium*. Given the plethora of cMG applications, a natural question arises: *are there algorithmic solutions for provable equilibrium computation in these games already?* Surprisingly, this is not the case! Notwithstanding the model's appeal, an array of technical challenges arise that impede a straightforward algorithmic solution. Yet, empirical results suggest that variants of policy gradient methods (Williams, 1992) can actually lead to favorable outcomes.

Following this thread, we consider the simplest reasonable setting, *competition between two agents*, and pose our

[1]Department of Computer Science & Engineering, University of California, San Diego, La Jolla, CA, USA [2]Department of Computer Sciences, University of Wisconsin-Madison, Madison, WI, USA and Archimedes, Athena Reaserch Center, Greece [3]Google DeepMind, London, UK. Correspondence to: Fivos Kalogiannis <fkalogiannis@ucsd.edu>.

*Proceedings of the 42$^{nd}$ International Conference on Machine Learning*, Vancouver, Canada. PMLR 267, 2025. Copyright 2025 by the author(s).

central question:

> *Do policy gradient methods converge in zero-sum convex Markov games?* 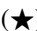 (★)

We answer this question in the affirmative. More explicitly, we provide an algorithmic framework that is *efficient*, adheres to the *independent learning* desideratum (Daskalakis et al., 2020), and is simultaneously *easy-to-implement*. This effectiveness is powered by novel technical insights to address numerous technical challenges native to cMGs.

Getting into the weeds, we enumerate the technical challenges that cMGs pose. As cMGs are a strict generalization of cMDPs, the Bellman consistency of the agents' utility functions fails to hold (Zhang et al., 2020)—in short, we cannot define individual *value* and *action-value* functions. Remarkably, this is a family of long-horizon strategic interactions where agents *cannot use dynamic programming* as opposed to conventional MGs. We note that, even though most policy optimization methods rely on a gradient-based approach, their majority implicitly performs an approximate dynamic programming subroutine such as value-iteration (Jin et al., 2021; Wei et al., 2021; Ding et al., 2022; Erez et al., 2023).

Consequently, the absence of state-wise value-functions translates into a failure of the elementary arguments that were used to prove the existence of Nash equilibria (Fink, 1964), even in two-player zero-sum Markov games (Shapley, 1953). In light of the latter, Gemp et al. (2024) prove the existence of Nash equilibria in cMGs by going beyond the fundamental toolbox of the Brouwer and the Kakutani fixed point theorems. What renders these conventional theorems obsolete is the inherent *nonconvexity of the individual best-response mappings* in terms of players' policies—where, each best-response mapping yields the set of the agent's set of utility maximizing strategy deviations.

The careful reader might already grasp that the failure of Bellman consistency rules out the seamless application of the majority of MARL algorithms; most of them rely on computing *value* and *action-value* functions. We direct our hopes to policy gradient methods which theory of cMDPs and practice of MARL suggest can work. Directly optimizing a policy means facing an optimization landscape that is nonconvex. However, since *all local optima are also global in cMGs* should bring some hope. Nonetheless, why should policy gradient methods work in cMGs when vanilla gradient following methods cycle (Bailey & Piliouras, 2018) and exhibit non-convergent chaotic trajectories even in zero-sum two-player normal-form games? Observe that the latter are nothing but cMGs on a single state with individually linear utilities. Further, even if we stabilize gradient dynamics towards Nash equilibria, attaining strong convergence guarantees of an algorithm requires the deepening of our understanding of the optimization landscape of cMDPs and cMGs. For instance, in the min-max case, the computational complexity of the general problem of computing a saddle-point of a nonconvex-nonconcave smooth function remains far from being settled (Daskalakis et al., 2021), while no algorithm is yet guaranteed to efficiently compute them without additional structural assumptions.

Searching for structure, we observe that the utilities in cMGs exhibit a property known as *hidden convexity* (Ben-Tal & Teboulle, 1996; Li et al., 2005; Wu et al., 2007; Xia, 2020; Vlatakis-Gkaragkounis et al., 2021; Fatkhullin et al., 2023). However, the existing results on hidden convexity either focus on single-objective minimization or, in the context of continuous games, make assumptions that do not hold in cMGs (separability of the hidden mapping and unconstrained variables). Having identified the key difficulties of zero-sum cMGs, in the following section we offer a detailed account of how we tackle them. Briefly, we manage to overcome the aforementioned challenges using some distinct combination of the following algorithmic techniques: (i) a conceptually simple but *nonconvex regularization* of the utility function; (ii) *alternating gradient iterations* (Tammelin et al., 2015; Chavdarova et al., 2021; 2019; Zhang & Yu, 2019; Chambolle & Pock, 2011); (iii) a careful *timescale separation* of the individual step-size, and lastly (iv) *nested-gradient iterations*.

## 1.1. Technical Overview

The proof of the proposed theoretical guarantees comes after a combination of several key observations—old and new. We believe that exposing them in the following itemized manner will serve in conveying the technical level of our work and an intuitive overview of why our techniques work.

- Hidden convex functions are nonconvex functions that can be expressed as a convex function of a reparametrization of their original arguments, termed the "hidden (latent) mapping," while the original arguments are called "control variables". Yet, in some settings, like reinforcement learning (RL), this mapping is not at all hidden but directly computable or observable.

- More specifically, in cMGs, players' utilities are hiddenly concave through the state-action occupancy measure, which is similarly computable (in planning) or observable (in RL).

- Although analyzing these games via occupancy measures yields a quasi-variational inequality problem (Kinderlehrer & Stampacchia, 2000)–unnecessarily increasing the technical challenges (Bernasconi et al., 2024)—yet, two crucial facts remain: (i) the utility is concave with respect to the hidden mapping, and (ii) the hidden mapping is accessible. These properties ensure that after regular-

ization, the hidden function becomes strongly concave. In turn, the resulting function satisfies the proximal Polyak-Łojasiewicz (pPL) condition, with respect to the control variables (*i.e.*, the policies).

- Consequently, each player's utility remains pPL while the feasibility sets for their control variables remain independent. This observation wraps up the discussion on hidden concavity and naturally shifts our focus on the constrained optimization of min-max nonconvex-pPL objectives. In other words, we reduce solving two-player zero-sum convex Markov games to the problem of computing saddle-points of functions that are nonconvex-pPL over constrained domains.

- Leveraging this reduction, our final steps center on developing policy gradient methods that can provably converge to saddle-points of nonconvex-pPL functions over constrained domains, a technical development in optimization theory of independent interest as well.

The proofs are deferred to the Appendix for clarity.

### 1.2. Our Contributions

Before proceeding to enumerate all parts of our contributions, we remark that we deliver a definitive answer to question (★):

**Theorem.** *There exist decentralized policy gradient methods (Algorithms 1 and 2) that compute an $\epsilon$-approximate Nash equilibrium for any $\epsilon > 0$ in any two-player zero-sum convex Markov game using iterations and samples that are:*

$$\mathsf{poly}\left(\frac{1}{\epsilon}, |\mathcal{S}|, |\mathcal{A}| + |\mathcal{B}|, \frac{1}{1-\gamma}\right),$$

*with $\mathcal{S}$ denoting the cMG's state-space, $\mathcal{A}, \mathcal{B}$ the two players' action-spaces, and $\gamma > 0$ the discount factor.*

Our contributions span two key areas: constrained nonconvex min-max optimization and equilibrium computation in convex Markov games.

- In Theorem 4.1 we demonstrate that the best-response mapping for an objective $f(x, y)$, $x \mapsto y^\star(x) := \arg\max_{y \in \mathcal{Y}} f(x, y)$, is *Lipschitz continuous* in $x$ for regularized hidden-convex—and more generally, NC-pPL games. The significance of this result lies in guaranteeing the stability of the iterates of policy gradient methods and serves as a suggestion to practitioners looking for a regularization technique that is intuitive, simple, and easily implementable.

- In Section 4, we provide the first provable guarantees of convergence to a Nash equilibrium for two policy gradient algorithms `Nest-PG` and `Alt-PGDA` (Theorem 4.3).

**Key ingredients.** To establish our results, we leverage a set of incorporating the distinct combination of the following non-trivial components: (i) an intuitive stability-inducing specialized *regularization* of the utility function and (ii) *alternating* or (iii) *nested-gradient iterations*.

A noteworthy element of our approach is that it ensures convergence even with *inexact* gradients on top of being *stochastic*. The algorithm's robustness to gradient inexactness preserves each player's autonomy, allowing them to optimize independently without exchanging private policy information. *I.e.*, unlike the single-agent setting, where exact gradients can be stochastically estimated, our framework's regularizer depends on *both* players' policies, making exact estimation infeasible without policy sharing. Consequently, each player must rely solely on an inexact estimation of their own gradient.

Finally, a noteworthy product of our approach is the Lipschitz continuity of the best-response mapping (which returns the optimal strategy given the opponent's choices) in the hidden-convex case–and more generally nonconvex-pPL—despite the opponent's utility being nonconvex—contrasting with the typical $\frac{1}{2}$-Hölder continuity of maximizers in constrained optimization (Li & Li, 2014). Indeed, while (Kalogiannis et al., 2024) employ a similar technique of regularizing the value function through the occupancy measure perspective, their result only establishes a weaker notion of continuity due to the *coupledness* of individual players' state-action occupancy measure. This claim has been independently supported by (Papadimitriou et al., 2023, Robust Berge Theorem), which considers general nonconvex-strongly concave functions where the feasibility sets of two different strategies depend on each other in a Hausdorff continuous manner. To significantly strengthen the continuity of the best-response mapping, we leverage the pPL condition in the individual policy spaces of agents that remain *uncoupled*. This result was only known for the significantly simpler case of unconstrained two-sided PL functions (Nouiehed et al., 2019).

## 2. Preliminaries

**Notation.** In general, $x, y, z, u, v, w$ and $\lambda, \lambda_1, \lambda_2$ will denote vectors. Scalars will be denoted using $\alpha, \beta, \gamma, \delta, \epsilon, \zeta, \kappa, \mu, \nu$ and $a, b, c, d$. Matrices will be denoted with bold uppercase letters. The probability simplex supported on a finite set $\mathcal{M}$ will be denote with $\Delta(\mathcal{M})$. For a compact convex set $\mathcal{Z} \subseteq \mathbb{R}^d$, we will denote its Euclidean diameter as $D_{\mathcal{Z}}$, *i.e.*, $D_{\mathcal{Z}} := \max_{x,y \in \mathcal{Z}} \|x - y\|_2$. Lastly, the global optimum of a function $f$ will be denoted as $f^\star$.

### 2.1. Convex Markov Games

In this subsection, we define two-player zero-sum convex Markov games (Gemp et al., 2024) and introduce necessary notation. We then present the *occupancy measure* and remark its continuity properties. Subsequently, we review the *policy gradient theorem* for convex MDPs and describe a stochastic policy gradient estimator. Finally, we define the perturbed utility function $U^\mu$, obtained by adding a regularization term to the original utility function $U$.

**Definition 1** (Two-player zero-sum cMG). An infinite-horizon zero-sum convex Markov game is a tuple $\Gamma = (\mathcal{S}, \mathcal{A}, \mathcal{B}, \mathbb{P}, F, \gamma, \varrho)$:

- a maximizing and a minimizing player,
- a finite state space $\mathcal{S}$, and an initial state distribution $\varrho \in \Delta(\mathcal{S})$,
- finite action spaces $\mathcal{A}, \mathcal{B}$,
- a state transition function $\mathbb{P} : \mathcal{S} \times \mathcal{A} \times \mathcal{B} \to \Delta(\mathcal{S})$,
- a discount factor $\gamma \in [0, 1)$, and
- two continuous utilities $F_1, F_2$ functions corresponding to each player's occupancy measure, *i.e.*, there exist concave $F_1, F_2$

$$F_1 : \Delta(\mathcal{S} \times \mathcal{A}) \times \Delta(\mathcal{S} \times \mathcal{B}) \to \mathbb{R};$$
$$F_2 : \Delta(\mathcal{S} \times \mathcal{A}) \times \Delta(\mathcal{S} \times \mathcal{B}) \to \mathbb{R},$$

with $-F_1 = F_2 =: F$.

With all this in hand, we adopt the following standard assumptions, which are widely used in prior work (Zhang et al., 2020):

**Assumption 1.** For the initial state distribution, it holds that $\varrho(s) > 0, \forall s$.

Additionally, we assume *direct policy parametrization* and define the Markovian and stationary policy of the minimizing and the maximizing player to be $x \in \Delta(\mathcal{A})^{|\mathcal{S}|} =: \mathcal{X}$ and $y \in \Delta(\mathcal{B})^{|\mathcal{S}|} =: \mathcal{Y}$ respectively. Throughout, we only consider Markovian stationary policies. After the agents fix their policies, $x, y$, the transition matrix $\mathbb{P}(x, y) \in \mathbb{R}^{|\mathcal{S}| \times |\mathcal{S}|}$ dictates how they traverse the state space. The occupancy measures are defined as:

$$\lambda_1^{s,a} := (1 - \gamma)\mathbb{E}_{x,y}\left[\sum_h^\infty \gamma^h \mathbb{1}\{s^{(h)} = s, a^{(h)} = a\}|\varrho\right];$$
$$\lambda_2^{s,b} := (1 - \gamma)\mathbb{E}_{x,y}\left[\sum_h^\infty \gamma^h \mathbb{1}\{s^{(h)} = s, b^{(h)} = b\}|\varrho\right].$$

**Further notation.** Further, we denote $\lambda$ to be the state-joint-action occupancy measure, $\lambda \in \Delta(\mathcal{S} \times \mathcal{A} \times \mathcal{B})$. Overloading notation, $\lambda(x, y)$ stands for the unique occupancy measure that corresponds to the policy pair $x, y$. Additionally, $\lambda_1 \in \Delta(\mathcal{S} \times \mathcal{A}), \lambda_2 \in \Delta(\mathcal{S} \times \mathcal{B})$ will signify the marginal occupancy measures with respect to the minimizing and the maximizing player respectively. Again,

overloading notation, $\lambda_1(x, y)$ and $\lambda_2(x, y)$ are the unique occupancy measures for a policy pair $x, y$. Finally, we will at times suppress the notation $F_1(\lambda_1(x, y); y)$ in place of $F_1(\lambda_1(x, y), \lambda_2(x, y))$ and similarly for $F_2$.

Crucially, both the occupancy measure and its inverse operators satisfy the following continuity properties:

**Lemma 2.1** (Continuity of the occupancy measure). *Let $\lambda \in \Delta(\mathcal{S} \times \mathcal{A} \times \mathcal{B})$ be the occupancy measure in a (convex) Markov game and let $\lambda_1^{-1} : \Delta(\mathcal{S} \times \mathcal{A}) \to \mathcal{X}$ and $\lambda_2^{-1} : \Delta(\mathcal{S} \times \mathcal{B}) \to \mathcal{Y}$ be the occupancy-to-policy mapping such that: $\lambda_1^{-1}(\lambda_1(x, y)) = x; \lambda_2^{-1}(\lambda_2(x, y)) = y$. Then,*

- *$\lambda$ is $L_\lambda$-Lipschitz continuous and has an $\ell_\lambda$-Lipschitz continuous gradient with respect to the policy pair $(x, y)$ in $\mathcal{X} \times \mathcal{Y}$. Specifically, for all $(x, y)$ and $(x', y')$,*

$$\|\lambda(x, y) - \lambda(x', y')\| \le L_\lambda \|(x, y) - (x', y')\|;$$
$$\|\nabla\lambda(x, y) - \nabla\lambda(x', y')\| \le \ell_\lambda \|(x, y) - (x', y')\|,$$

*where $L_\lambda := \frac{|\mathcal{S}|^{\frac{1}{2}}(|\mathcal{A}| + |\mathcal{B}|)}{(1 - \gamma)^2}$, and $\ell_\lambda := \frac{2\gamma|\mathcal{S}|^{\frac{1}{2}}(|\mathcal{A}| + |\mathcal{B}|)^{\frac{3}{2}}}{(1 - \gamma)^3}$.*

- *For any fixed $y$ (respectively, $x$), $\lambda^{-1}$ is $L_{\lambda^{-1}}$-Lipschitz continuous with respect to $\lambda_1$ (respectively, $\lambda_2$), i.e., for all $\lambda_1(x, y), \lambda_1(x', y)$—respectively, $\lambda_2(x, y), \lambda_2(x, y')$,*

$$\|x - x'\| \le L_{\lambda^{-1}} \left\|\lambda_1^{-1}(\lambda_1(x, y) - \lambda_1(x', y))\right\|;$$
$$\|y - y'\| \le L_{\lambda^{-1}} \left\|\lambda_2^{-1}(\lambda_2(x, y) - \lambda_2(x, y'))\right\|,$$

*with $L_{\lambda^{-1}} := \frac{2}{\min_s \varrho(s)(1 - \gamma)}$.*

Next, our solution concept corresponds to the following min-max Nash equilibrium of the following function $U : \Delta(\mathcal{A})^{|\mathcal{S}|} \times \Delta(\mathcal{B})^{|\mathcal{S}|} \to \mathbb{R}$ be such that:

$$U(x, y) := F(\lambda_1(x, y), \lambda_2(x, y)).$$

**Definition 2** ($\epsilon$-NE). A policy profile $(x^\star, y^\star) \in \mathcal{X} \times \mathcal{Y}$ is said to be an $\epsilon$-approximate Nash equilibrium ($\epsilon$-NE), if for any $x \in \mathcal{X}$ and any $y \in \mathcal{Y}$, it holds that:

$$U(x^\star, y) - \epsilon \le U(x^\star, y^\star) \le U(x, y^\star) + \epsilon. \qquad (\epsilon\text{-NE})$$

Finally, we will denote $U^\mu$ to be: $U^\mu(x, y) := U(\lambda(x, y)) - \frac{\mu}{2}\|\lambda_2(x, y)\|^2$. The following Lemma is a direct consequence of hidden-strong-convexity.

**Lemma 2.2.** *When $\mu > 0$, $U^\mu(x, \cdot)$ has a unique maximizer $y^\star(x)$, for all $x$.*

### 2.2. Policy Gradient Estimators

As discussed, the inexistence of value or action-value functions for general utility MDPs, leads us to focus on policy gradient methods; the direct application of vanilla Q-learning methods is out of the question. To compute

the policy gradient of a utility that is nonlinear in $\lambda_1$, we make use of the (Williams, 1992) along the chain rule of differentiation to write:

$$\nabla_x U(x,y) = \sum_{s \in \mathcal{S}} \sum_{a \in \mathcal{A}} \frac{\partial F_1}{\partial \lambda_1^{s,a}(x,y)} \nabla_x \lambda_1^{s,a}(x,y).$$

By sampling trajectories, each agent can stochastically estimate the policy gradient using the following estimator.

**Definition 3** (Gradient Estimator). Given a trajectory $\xi = \left(s^{(0)}, a^{(0)}\right), s^{(1)}, \ldots, s^{(H-1)}, a^{(H-1)}\right)$ of length $H$ sampled under a policy profile $x, y$ and initial distribution $\varrho$, the gradient estimator, $\hat{g}_x(\xi|x,z)$, is defined as:

$$\hat{g}_x(\xi|x,z) :=$$
$$\sum_{h=0}^{H-1} \gamma^h z \left(s^{(h)}, a^{(h)}\right) \left(\sum_{h'=0}^{h} \nabla_x \log x \left(a^{(h')}|s^{(h')}\right)\right),$$

with $z = \nabla_{\lambda_1} F_1(\lambda_1(x,y); y)$.

**Sufficient exploration** In order to ensure that the agents sufficiently explore the environment and ultimately control the variance of the estimators, we assume that both agents are following $\varepsilon$-greedy direct policy parametrization, *i.e.,* for a given parameter value $x \in \Delta^{|\mathcal{S}|}(\mathcal{A})$, the agent plays according to $\pi_x = (1 - \varepsilon)x + \frac{\varepsilon \mathbf{1}}{|\mathcal{A}|}$, where $\mathbf{1}$ is an all-ones vector of appropriate dimension.

### 2.3. Optimization Theory

Next, we introduce several key concepts from nonconvex and min–max optimization, focusing on hidden convexity and gradient domination conditions. We demonstrate how strong hidden convexity implies the proximal *Polyak-Łojasiewicz condition* (pPL) and the *quadratic growth condition* (QG). We then show that when a min–max objective satisfies a two-sided gradient domination condition, it enjoys a zero duality gap.

**Definition 4** (Hidden Convex Function). Consider a function $f : \mathcal{X} \to \mathbb{R}$ where $\mathcal{X}$ is a compact convex set such that $f(x) := H\left(c(x)\right), \forall x \in \mathcal{X}$ for some mapping $c$ and function $H$. If the following conditions are satisfied:

- the mapping $c$ is invertible and its inverse $c^{-1}$ is $1/\mu_c$ Lipschitz continuous.
- the set $\mathcal{U} := c(\mathcal{X})$ is convex and the function $H : \mathcal{U} \to \mathbb{R}$ is $\mu_H$-strongly convex.

The function is said to be $(\mu_c, \mu_H)$-*hidden strongly convex* (HSC), while for $\mu_H = 0$, it is referred merely as *hidden convex* (HC).

Notably, the convergence analysis of our proposed methods begins with the following claim, which connects cMG utilities to hidden convexity.

**Claim 2.3.** *The function $U$ is hidden-convex (resp., hidden-concave) for the min. player (resp., max. player), given a fixed action of the opponent. Similarly, the perturbed utility function $U^\mu$ is $(\mu, L_{\lambda^{-1}})$-hidden strongly concave for the max player due to the structure of the regularizer.*

*Proof.* Follows from Definitions 1 and 4 & Lemma 2.1. □

**Proposition 2.4** (HC implies gradient domination; (Fatkhullin et al., 2023)). *Let $f : \mathcal{X} \to \mathbb{R}$ be an $\ell$-smooth and $(\mu_c, \mu_H)$-hidden convex function and $I_\mathcal{X}$ be the indicator function of the set $\mathcal{X}$. Further, let $F(\cdot) := f(\cdot) + I_\mathcal{X}(\cdot)$. Also, assume that the map $c(\cdot)$ is continuously differentiable on $\mathcal{X}$.*

*(i) If the set $\mathcal{X} = c(\mathcal{X})$ is bounded with diameter $D_\mathcal{U}$, then*

$$\inf_{s_x \in \partial F(x)} \|s_x\| \geq \frac{\mu_c}{D_\mathcal{U}} (F(x) - F^*), \quad \forall x \in \mathcal{X}.$$

*(ii) If $f(\cdot)$ is $(\mu_c, \mu_H)$-hidden strongly convex, then*

$$\inf_{s_x \in \partial F(x)} \|s_x\|^2 \geq 2\mu_c^2 \mu_H (F(x) - F^*), \quad \forall x \in \mathcal{X}.$$

**Definition 5** (pPL). Let an $\ell$-smooth function $f : \mathcal{X} \to \mathbb{R}$ defined over the convex and compact set $\mathcal{X} \subseteq \mathbb{R}^d$. Let $\mathcal{D}_\mathcal{X}(\cdot, \ell)$ be defined as:

$$\mathcal{D}_\mathcal{X}(x, \ell) := -2\ell \min_{y \in \mathcal{X}} \left\{ \langle \nabla f(x), y - x \rangle + \frac{\ell}{2} \|x - y\|^2 \right\}.$$

Then, $f$ is said to satisfy the proximal Polyak-Łojasiewicz condtion with modulus $\mu$ if for all $x \in \mathcal{X}$, it holds true that:

$$\frac{1}{2} \mathcal{D}_\mathcal{X}(x, \ell) \geq \mu (f(x) - f^*).$$

Our following lemma establishes a variant of the "gradient dominance" for the case of convex Markov games. Namely we show that an approximate constrained first-order stationary point ensures approximately optimal policies in our game.

**Lemma 2.5** (Gradient Dominance). *For a zero-sum convex Markov game, it holds that:*

$$\max_{x' \in \mathcal{X}} \langle \nabla_x U(x,y), x - x' \rangle \geq \mu_x (U(x,y) - U(x^\star, y));$$
$$\max_{y' \in \mathcal{Y}} \langle \nabla_y U(x,y), y' - y \rangle \geq \mu_y (U(x, y^\star) - U(x,y)),$$

*for $\mu_x, \mu_y = \frac{(1-\gamma) \min_s \rho(s)}{2\sqrt{2}}$.*

*Proof.* Fix an arbitrary $y \in \mathcal{Y}$. By the hidden convexity of $U(\cdot, y)$ and Proposition 2.4, we have:

$$\frac{(1-\gamma) \min_s \varrho(s)}{2\sqrt{2}} \left( U(x,y) - \min_{x^\star \in \mathcal{X}} U(x^\star, y) \right)$$
$$\leq \min_{s_x \in \partial_x U(x,y) + \partial_x I_\mathcal{X}(x)} \|s_x\|.$$

Where, $\sqrt{2}$ is the diameter of the state-action occupancy measure. Applying (Rockafellar & Wets, 2009, Proposition 8.32), we obtain:

$$\min_{s_x \in \partial_x U(x,y) + \partial_x I_{\mathcal{X}}(x)} \|s_x\| = \max_{\substack{x' \in \mathcal{X}, \\ \|x-x'\| \leq 1}} \langle \nabla U(x,y), x - x' \rangle$$

$$\leq \max_{x' \in \mathcal{X}} \langle \nabla U(x,y), x - x' \rangle .$$

Thus, we set $\mu_x := \frac{(1-\gamma)\min_s \varrho(s)}{2\sqrt{2}}$, and by symmetry, the same holds for $\mu_y$. $\qquad\square$

Finally, in the Appendix we show that both HSC and pPL imply QG, which states that the optimality gap at any point $x$ is lower-bounded by a quadratic term in its distance from the minimizer. This ensures that progress toward optimality can bound the proximity to the solution.

**Proposition 2.6** (QG from HSC and pPL). *Let $f : \mathcal{X} \to \mathbb{R}$ be $\ell$-smooth and satisfy either: (i) $(\mu_c, \mu_H)$-hidden strong convexity (HSC), or (ii) proximal Polyak-Łojasiewicz (pPL) with modulus $\mu$. Then, $f$ satisfies the* quadratic growth (QG) *condition:*

$$f(x) - f(x^\star) \geq \frac{\mu_{QG}}{4} \|x - x_p^\star\|^2, \quad \forall x \in \mathcal{X},$$

*where $x_p^\star$ is the closest minimizer in $\mathcal{X}^\star = \arg\min_{x \in \mathcal{X}} f(x)$, and*

$$\mu_{QG} = \mu_c^2 \mu_H \quad \text{(HSC)}, \quad \mu_{QG} = \mu \quad \text{(pPL)}.$$

**Piecing the Framework Together.** At a high level, the standard tool in proving convergence to Nash equilibria for gradient-based methods is the construction of a potential—or *Lyapunov*—function (See (Bof et al., 2018)). A Lyapunov function needs to be lower-bounded and decreasing for consecutive iterates of the algorithm. However, proving the latter beyond the convex-concave setting, requires a careful examination of HSC and pPL. As such, in order to shed light on the *structured* nonconvex landscape, we note:

- Starting from the taxonomy of structured nonconvex functions, we show in the appendix that HSC and pPL are equivalent given a proper transformation of their moduli.

- Hence, the stationarity surrogate $\mathcal{D}_{\mathcal{X}}(x, \ell)$ also serves as a surrogate for optimality $f(x) - \min_{x' \in \mathcal{X}} f(x')$. More importantly, by Claim 2.3, in the min-max setting, for a fixed opponent strategy, a sufficiently small $\mathcal{D}_{\mathcal{X}}(x, \ell)$ indicates that the player has found an approximate best response.

Furthermore, we highlight an additional crucial detail. Since, the pPL condition already guarantees quadratic growth (with an equal modulus), the contribution of HSC in this regard may appear redundant. Nonetheless, HSC is

translated to the pPL condition going through an argument which degrades the modulus of HSC $\mu$ to a pPL modulus of $\mu_{PL} = O(\mu^2)$. Hence, directly guaranteeing QG from HSC improves convergence rates of the next sections by at least an order of $O(\epsilon^{3/2})$.

# 3. New Insights in Structured Nonconvex Min-Max Optimization

As a preliminary step of independent interest, we present our results on optimization schemes for constrained min-max problems under nonconvex-pPL and pPL-pPL conditions. To maintain completeness, we restate our solution concept under constrained min-max optimization perspective:

**Definition 6** ($\epsilon$-SP). *Assume a smooth function $f : \mathcal{X} \times \mathcal{Y} \to \mathbb{R}$ where $\mathcal{X}, \mathcal{Y}$ are two compact and convex sets. Then, $(x^\star, y^\star) \in \mathcal{X} \times \mathcal{Y}$ is an $\epsilon$-saddle-point ($\epsilon$-SP) of $f$ if,*

$$\begin{cases} \max_{x \in \mathcal{X}} \langle \nabla_x f(x^\star, y^\star), x^\star - x \rangle & \leq \epsilon; \\ \max_{y \in \mathcal{Y}} \langle \nabla_y f(x^\star, y^\star), y - y^\star \rangle & \leq \epsilon \end{cases} \quad (\epsilon\text{-SP})$$

We adopt a context-agnostic approach, assuming each player receives a "black-box" representation of their gradient vector. These hypotheses remain intentionally abstract, as we impose no specific modeling assumptions on how players' payoff signals are generated. In this sense, they serve as an "inexact gradient oracle" (Devolder et al., 2014) that captures a wide range of settings. Reflecting on our MARL scenario, this framework allows each player to independently select and apply their preferred gradient estimator, as in (Zhang et al., 2020; 2021; Barakat et al., 2023).

**Model** (Stochastic Inexact First-Order Oracle). For any $t$, the gradient estimators $\hat{g}_x(x_t, y_t)$ and $\hat{g}_y(x_t, y_t)$ satisfy:

$$\mathbb{E}[\hat{g}_x(x_t, y_t)] = g_x(x_t, y_t), \quad \mathbb{E}[\hat{g}_y(x_t, y_t)] = g_y(x_t, y_t);$$
$$\mathbb{E}\|\hat{g}_x(x_t, y_t)\|^2 \leq \sigma_x^2, \qquad \mathbb{E}\|\hat{g}_y(x_t, y_t)\|^2 \leq \sigma_y^2.$$

Additionally, scalars $\delta_x, \delta_y > 0$ bound the inexactness error:

$$\|g_x(x_t, y_t) - \nabla_x f(x_t, y_t)\| \leq \delta_x,$$
$$\|g_y(x_t, y_t) - \nabla_y f(x_t, y_t)\| \leq \delta_y.$$

Hence, $\hat{g}_x$ and $\hat{g}_y$ provide unbiased estimates of $g_x$ and $g_y$, respectively, which in turn serve as inexact approximations of $\nabla_x f(x_t, y_t)$ and $\nabla_y f(x_t, y_t)$. As a result, we encounter two types of errors: (i) *systematic bias* (non-zero mean), bounded by $\delta_x, \delta_y$, and (ii) *random noise* (zero mean), with variance bounded by $\sigma_x^2, \sigma_y^2$.

## 3.1. Alternating Methods

A widely used stabilization technique in nonconvex min-max optimization, including adversarial training and GANs,

is *alternating* updates between players (Lu et al., 2019; Nouiehed et al., 2019; Gidel et al., 2019; Bailey et al., 2020; Wibisono et al., 2022; Cevher et al., 2023; Lee et al., 2024). Building on this approach, we analyze nested and alternating gradient iteration schemes.

### 3.1.1. NESTED GRADIENT ITERATIONS

As a warm-up, we focus on solving the minimax problem ($\epsilon$-SP) under the assumption that we have access to an *oracle for approximate inner maximization*, similar to (Jin et al., 2020, Section 4). Specifically, for any given $x$, the oracle provides a $y'$ such that: $f(x, y') \geq \max_y f(x, y) - \epsilon$.

$$\begin{cases} y_{t+1} \leftarrow \texttt{ARGMAX}(f(x_t, \cdot), \epsilon_y); \\ x_{t+1} \leftarrow \Pi_{\mathcal{X}}\left(x_t - \eta \nabla f(x_t, y_{t+1})\right) \end{cases} \quad \text{(GDmax)}$$

**Theorem 3.1** (NC-pPL; Informal version of Theorem D.5)**.** *Let* $f : \mathcal{X} \times \mathcal{Y}$ *be an* $L$-*Lipschitz and* $\ell$-*smooth function and* $\mathcal{X}, \mathcal{Y}$ *be compact convex sets with diameters* $\mathrm{D}_{\mathcal{X}}, \mathrm{D}_{\mathcal{Y}}$ *respectively. Also, assume that* $f(x, \cdot)$ *satisfies the pPL condition with modulus* $\mu > 0$*. Then, the iteration scheme (GDmax) run with a tuning of*

- *step-sizes:* $\tau_x = \Theta\left(\frac{1}{\ell\kappa}\right)$ *and* $\tau_y = \Theta\left(\frac{1}{\ell}\right)$

- *batch-sizes* $M_x = \Theta\left(\frac{\sigma_x^2}{\epsilon^2}\right)$ *and* $M_y = \Theta\left(\frac{\kappa\sigma_y^2}{\epsilon^2}\right)$,

*where* $\kappa := \frac{\ell}{\mu}$*, outputs an* $(\epsilon + \delta_x + \delta_y)$-*saddle-point of* $f$ *after a total number of outer-loop and inner-loop iterations,* $T$*, that is at most*

$$T = O\left(\frac{\kappa^3 L\left(\mathrm{D}_{\mathcal{X}} + \mathrm{D}_{\mathcal{Y}}\right)}{\epsilon^2} \log\left(\frac{1}{\epsilon}\right)\right).$$

**Theorem 3.2** (pPL-pPL; Informal version of Theorem D.6)**.** *Let* $f : \mathcal{X} \times \mathcal{Y}$ *be an* $L$-*Lipschitz and* $\ell$-*smooth function and* $\mathcal{X}, \mathcal{Y}$ *be compact convex sets with diameters* $\mathrm{D}_{\mathcal{X}}, \mathrm{D}_{\mathcal{Y}}$ *respectively. Further, assume that* $f$ *satisfies the two-sided pPL condition with moduli* $\mu_x, \mu_y$*. Then, the iteration scheme (GDmax) run with a tuning of*

- *step-sizes:* $\tau_x = \Theta\left(\frac{1}{\ell\kappa}\right)$ *and* $\tau_y = \Theta\left(\frac{1}{\ell}\right)$

- *batch-sizes* $M_x = \Theta\left(\frac{\kappa_x\sigma_x^2}{\epsilon}\right)$ *and* $M_y = \Theta\left(\frac{\ell\kappa_y\sigma_y^2}{\epsilon^2}\right)$,

*outputs an* $\left(\epsilon + \sqrt{\ell_{\Phi}/\mu_x}(\delta_x + \delta_y)\right)$-*saddle-point of* $f$ *after a number of iterations,* $T$*, that is at most*

$$T = O\left(\frac{\ell^2}{\mu_x\mu_y} \log\frac{\ell L\kappa_x\mathrm{D}_{\mathcal{X}}}{\epsilon} \log\frac{\ell L\kappa_y\mathrm{D}_{\mathcal{Y}}}{\epsilon}\right),$$

*where* $\kappa_x := \frac{\ell}{\mu_x}$ *and* $\kappa_y := \frac{\ell}{\mu_y}$*.*

### 3.1.2. ALTERNATING GRADIENT DESCENT ASCENT

Given the simplicity and practical superiority of single-loop methods over two-loop alternatives, a natural question arises: *can we achieve comparable convergence rates*

*without resorting to multi-loop procedures?* To this end, Alt-GDA leverages the sequential computation of $x_{t+1}$ and $y_{t+1}$, ensuring that each update benefits from fresher gradient information.

$$\begin{cases} x_t \leftarrow \Pi_{\mathcal{X}}\left(x_{t-1} - \tau_x \hat{g}_x(x_{t-1}, y_{t-1})\right) \\ y_t \leftarrow \Pi_{\mathcal{Y}}\left(y_{t-1} - \tau_y \hat{g}_y(x_t, y_{t-1})\right) \end{cases} \quad \text{(Alt-GDA)}$$

**Theorem 3.3** (NC-pPL; Informal version of Theorem D.7)**.** *Let* $f : \mathcal{X} \times \mathcal{Y}$ *be an* $L$-*Lipschitz and* $\ell$-*smooth function and* $\mathcal{X}, \mathcal{Y}$ *be compact convex sets with diameters* $\mathrm{D}_{\mathcal{X}}, \mathrm{D}_{\mathcal{Y}}$ *respectively. Also, assume that* $f(x, \cdot)$ *satisfies the pPL condition with modulus* $\mu > 0$ *for all* $x \in \mathcal{X}$*. Then, the iteration scheme (Alt-GDA) run with a tuning of*

- *step-sizes:* $\tau_x = \Theta\left(\frac{1}{\ell\kappa^2}\right)$ *and* $\tau_y = \Theta\left(\frac{1}{\ell}\right)$

- *batch-sizes* $M_x = \Theta\left(\frac{\ell^2\kappa^2\sigma_x^2}{\epsilon^2}\right)$ *and* $M_y = \Theta\left(\frac{\kappa^2\sigma_y^2}{\epsilon^2}\right)$,

*outputs an* $(\epsilon + \delta)$-*saddle-point of* $f$ *after a number of iterations,* $T$*, that is at most*

$$T = O\left(\frac{\kappa^2\ell L\left(\mathrm{D}_{\mathcal{X}} + \mathrm{D}_{\mathcal{Y}}\right)}{\epsilon^2}\right).$$

**Theorem 3.4** (pPL-pPL; Informal version of Theorem D.8)**.** *Let* $f : \mathcal{X} \times \mathcal{Y}$ *be an* $L$-*Lipschitz and* $\ell$-*smooth function and* $\mathcal{X}, \mathcal{Y}$ *be compact convex sets with diameters* $\mathrm{D}_{\mathcal{X}}, \mathrm{D}_{\mathcal{Y}}$ *respectively. Also, assume that* $f$ *satisfies the pPL condition in* $x$ *and* $y$ *with moduli* $\mu_x, \mu_y > 0$ *respectively. Then, the iteration scheme (Alt-GDA) run with a tuning of*

- *step-sizes:* $\tau_x = \Theta\left(\frac{\mu_y}{\ell^3}\right)$ *and* $\tau_y = \Theta\left(\frac{1}{\ell}\right)$

- *batch-sizes* $M_x = \Theta\left(\frac{\sigma_x^2}{\mu_x\epsilon}\right)$ *and* $M_y = \Theta\left(\frac{\sigma_y^2}{\mu_x\mu_y^2\epsilon}\right)$,

*outputs an* $\epsilon$-*saddle-point of* $f$ *after a number of iterations,* $T$*, that is at most*

$$T = O\left(\frac{\ell^3}{\mu_x\mu_y^2} \log\frac{L(\mathrm{D}_{\mathcal{X}} + \mathrm{D}_{\mathcal{Y}})}{\epsilon}\right).$$

## 4. Main Results – Convex Markov Games

In this section we present our main results regarding the convex Markov games (cMGs). We demonstrate that a utility function that is hidden concave in the occupancy measure satisfies the proximal-PL condition with respect to the player's policy. Then, we show that the maximizers of $U^{\mu}(x, \cdot)$ are Lipschitz continuous in the minimizing player's policy $x$. Finally, we describe two policy gradient algorithms that enjoy convergence to an approximate Nash equilibrium using a finite number of samples and iterations. Namely, (i) nested policy gradient (Nest-PG), and (ii) alternating policy gradient descent-ascent (Alt-PGDA).

Throughout, $\delta_x$ is implicitly tuned through the coefficient of the regularizer, $\mu$. Also, $\delta_y = 0$ since the maximizing player has access to unbiased stochastic estimates of the gradients. To see why $\mu$ controls $\delta_x$, we note that by the $L_{\text{reg}}$-Lipschitz continuity of the regularizer, the norm of its gradient is bounded. As such, $\delta_x = O(\mu L_{\text{reg}})$ and since we also pick $\mu = O(\epsilon)$, the bound on $\delta_x$ follows suit.

**Theorem 4.1** (Continuity of maximizers). *Consider the mapping $x \mapsto y^\star(x) := \arg\max_{y \in \mathcal{Y}} f(x,y)$. Then, for any two points $x_1, x_2$ it holds true that:*

$$\|y^\star(x_1) - y^\star(x_2)\| \le L_\star \|x_1 - x_2\|,$$

*where $L_\star := \frac{\ell}{\sqrt{\mu \mu_{\text{QG}}}}$.*

**Corollary 4.2.** *Define the function $\Phi^\mu : \mathcal{X} \to \mathbb{R}$ as $\Phi^\mu := \max_y \left\{ U(x,y) - \frac{\mu}{2}\|\lambda_2(x,y)\|^2 \right\}$. Then, for any $x, x' \in \mathcal{X}$, the following inequality holds:*

$$\|\nabla_x \Phi^\mu(x) - \nabla_x \Phi^\mu(x')\| \le \ell_{\Phi,\mu} \|x - x'\|.$$

*with $\ell_{\Phi,\mu} := \ell(1 + L_\star)$.*

### 4.1. Nested Policy Gradient

---
**Algorithm 1** `Nest-PG`: Nested Policy Gradient
---
**input** $(x_0, y_0)$, step-sizes $\tau_x, \tau_y$, regul. coeff. $\mu \ge 0$
   **for** $t = 1$ **to** $T_{\text{out}}$ **do**
      $y_{t,0} \leftarrow y_{t-1}$
      **for** $s = 1$ **to** $T_{\text{in}}$ **do**
         $y_{t,s} \leftarrow \Pi_{\mathcal{Y}} \left( y_{s-1} + \tau_y \hat{\nabla}_y U^\mu(x_{t-1}, y_{t,s-1}) \right)$
         $y_t \leftarrow y_{t,T_{\text{in}}}$
      **end for**
      $x_t \leftarrow \Pi_{\mathcal{X}} \left( x_{t-1} - \tau_x \hat{\nabla}_x U(x_{t-1}, y_t) \right)$
   **end for**
   Pick $t^\star \in \{1, \ldots, T\}$ of the best iterate.
**output** $(x_{t^\star}, y_{t^\star + 1})$.
---

**Theorem 4.3.** *Consider a two-player zero-sum cMG, $\Gamma$, and let $\epsilon$ be a desired accuracy $\epsilon > 0$. Then, Algorithm 1 run with appropriately tuned step-sizes $\tau_x, \tau_y > 0$, $\varepsilon_x, \varepsilon_y > 0$ batch-sizes $M_x, M_y > 0$ outputs an $\epsilon$-approximate Nash equilibrium after a number of iterations that is at most:*

$$O\left(\frac{1}{\epsilon^{\frac{11}{2}}}\right) \text{poly}\left(\frac{1}{\min_s \varrho(s)}, \gamma, \frac{1}{1-\gamma}, |\mathcal{S}|, |\mathcal{A}| + |\mathcal{B}|\right).$$

**Theorem 4.4.** *Consider a two-player zero-sum cMG, $\Gamma$, with with utility functions that are hidden strongly concave with moduli $\mu_x, \mu_y > 0$ respectively. Additionally, let $\epsilon$ be a desired accuracy $\epsilon > 0$. Then, Algorithm 1 run with approprite step-sizes $\tau_x, \tau_y > 0$, exploration parameters $\varepsilon_x, \varepsilon_y > 0$, and batch-sizes $M_x, M_y > 0$ outputs an $\epsilon$-approximate Nash equilibrium after a number of iterations*

*that is at most:*

$$O\left(\log \frac{1}{\epsilon}\right) \text{poly}\left(\frac{1}{\mu_x}, \frac{1}{\mu_y}, \frac{1}{\min_s \varrho(s)}, \gamma, \frac{1}{1-\gamma}, |\mathcal{S}|, |\mathcal{A}| + |\mathcal{B}|\right).$$

### 4.2. Alternating Policy Gradient Descent Ascent

---
**Algorithm 2** `Alt-PGDA`: Alternating Policy GDA
---
**input** $(x_0, y_0)$, step-sizes $\tau_x, \tau_y, T > 0$, regul. coeff. $\mu \ge 0$
   **for** $t = 1$ **to** $T$ **do**
      $x_t \leftarrow \Pi_{\mathcal{X}} \left( x_{t-1} - \tau_x \hat{\nabla}_x U^\mu(x_{t-1}, y_{t-1}) \right)$
      $y_t \leftarrow \Pi_{\mathcal{Y}} \left( y_{t-1} + \tau_y \hat{\nabla}_y U(x_t, y_{t-1}) \right)$
   **end for**
   Pick $t^\star \in \{1, \ldots, T\}$ of the best iterate.
**output** $(x_{t^\star}, y_{t^\star})$.
---

**Theorem 4.5** (Informal Version of Theorem E.12). *Consider a two-player zero-sum cMG, $\Gamma$, and let $\epsilon$ be a desired accuracy $\epsilon > 0$. Then, Algorithm 2 run with appropriately step-sizes $\tau_x, \tau_y > 0$, exploration parameters $\varepsilon_x, \varepsilon_y > 0$ batch-sizes $M_x, M_y > 0$, and a regularization coefficient $\mu = O(\epsilon)$, outputs an $\epsilon$-approximate Nash equilibrium after a number of iterations that is at most:*

$$O\left(\frac{1}{\epsilon^6}\right) \text{poly}\left(\frac{1}{\min_s \varrho(s)}, \gamma, \frac{1}{1-\gamma}, |\mathcal{S}|, |\mathcal{A}| + |\mathcal{B}|\right).$$

**Theorem 4.6** (Informal Version of Theorem E.13). *Consider a two-player zero-sum cMG, $\Gamma$, with utility functions that are hidden strongly concave with moduli $\mu_x, \mu_y > 0$ respectively. Additionally, let $\epsilon$ be a desired accuracy $\epsilon > 0$. Then, Algorithm 2 run with step-sizes $\tau_x, \tau_y > 0$, and batch-sizes $M_x, M_y > 0$ outputs an $\epsilon$-approximate Nash equilibrium after a number of iterations that is at most:*

$$O\left(\log \frac{1}{\epsilon}\right) \text{poly}\left(\frac{1}{\mu_x}, \frac{1}{\mu_y}, \frac{1}{\min_s \varrho(s)}, \gamma, \frac{1}{1-\gamma}, |\mathcal{S}|, |\mathcal{A}| + |\mathcal{B}|\right).$$

We deem noteworthy the fact that Theorem 4.6 guarantees (expected) last-iterate convergence for a class of nonconvex games that take place over a constrained domain.

## 5. Numerical Results

We demonstrate Algorithms 1 and 2 on an iterated version of rock-paper-scissors-*dummy* where each player remembers the actions selected in the previous round. Hence, the previous joint action constitutes the state in the Markov game. The *dummy* action is dominated by all other actions such that the Nash equilibrium of the stage game is uniform across rock, paper, and scissors with zero mass on *dummy*. We set the step-sizes $\tau_x = \tau_y = 0.1$ and vary the regularization coefficient $\mu$ to demonstrate its effect in biasing convergence.

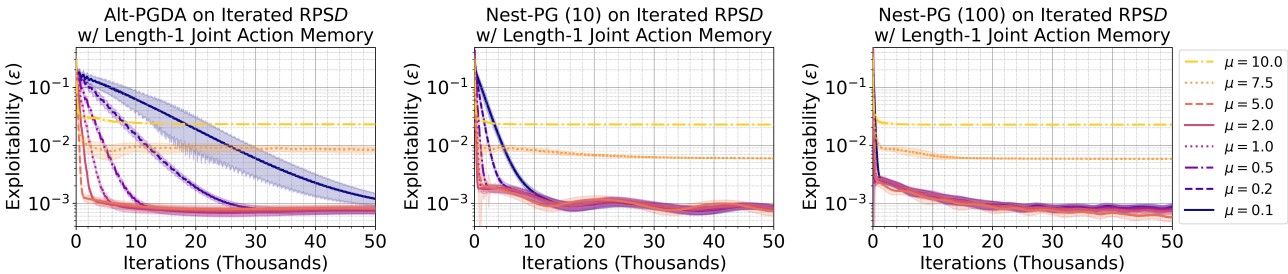

*Figure 1.* Exploitability decays towards a small, but positive value corresponding to the bias introduced by the regularization coefficient $\mu$. Results are averaged over 100 trials, each running the algorithm with a different randomly initialized policy profile. The leftmost plot reports results for Algorithm 2; the right two report results for Algorithm 1 with $T_{\text{in}} = 10$ and 100 respectively.

Our experiments suggest clearly that (i) there is a pronounced trade-off between speed of convergence and the exploitability of the solution of the perturbed problem (ii) more inner-loop iterations account for more stable exploitability decrease across consecutive iterates.

## 6. Conclusion and Future Work

Convex Markov games (cMGs) unify the domains of convex MDPs and Markov games. Even the rudimentary form of a cMG, a two-player pure competition, fosters a rich mosaic of applications spanning language model alignment, self-driving cars, and creative chess playing, to name a few.

In this work, we present the first algorithmic solution for computing Nash equilibria. To achieve this, we develop the first guarantees of convergence of alternating descent-ascent to a saddle-point for nonconvex functions that satisfy a one-sided or two-sided proximal Polyak-Łojasiewicz condition over constrained domains. We utilized these results to design a number of *independent policy gradient algorithms* for convex Markov games that provably *converge* to an approximate Nash equilibrium. In a nutshell, we develop *simple-to-use learning* dynamics that converge to the optimal game solutions even under realistic and challenging conditions where batch learning only allows noisy gradient estimation.

In terms of a message to practitioners, our work highlights that the regularized (`Alt-GDA`) algorithm is exceptionally effective and easy to deploy to tackle min-max problems. Its efficacy is achieved by mere (i) *regularization*, (ii) *alternating updates*, and (iii) *step-size magnitude separation*. Moreover, the generality of Theorems 3.3 and 3.4, coupled with the prevalence of the PL condition in modern machine learning objectives, makes a strong case in favor of (`Alt-GDA`) as the default algorithm for min-max optimization. Furthermore, we believe that it can seamlessly accommodate more elaborate update schemes on top of (i-iii), like adaptive learning rates, preconditioning, and, if necessary, higher-order information.

From a theoretical standpoint, we anticipate fascinating explorations of multi-player cMG interactions and a deep investigation of an array of game-theoretic solution concepts. We hope to see the experience of the rich multi-agent cMG applications inform theory and give rise to meaningful and challenging problems like equilibrium learning, equilibrium selection, and notions of equilibrium performance. We reiterate the fact that cMGs render the value-iteration subroutine useless. The latter lies at the heart of the corresponding algorithmic solutions for equilibrium learning or computation in conventional MGs; its inefficacy in cMGs, more than just posing an additional challenge, can stimulate the search for new algorithmic tools.

## Acknowledgements

FK gratefully acknowledges Panayotis Mertikopoulos for his mentoring and insightful discussions during the former's 2024 summer research fellowship at Archimedes, Athena Research Center.

## Impact Statement

This paper presents work whose goal is to advance the field of Machine Learning. There are many potential societal consequences of our work, none which we feel must be specifically highlighted here.

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

# Appendix

## Table of Contents

## Roadmap of the Appendix

Helping the reader navigate, we list a short summary of the topics that are covered in each section.

- In Appendix A, we attempt to present an overview of key research directions, theoretical landmarks, and applications. Interested readers can find further details in the cited literature therein although an exhaustive survey is infeasible.

- In Appendix B, we establish a series of lemmata concerning gradient descent for smooth functions: (i) Lemmata B.2 to B.4 are standard results regarding the iterates of projected gradient descent (with an exact or inexact and stochastic first-order oracle). (ii) We introduce $\mathcal{D}_X$ as the primary proxy of stationarity and derive a descent inequality for inexact gradient estimates based on this proxy (Lemmata B.7 to B.9). (iii) The section concludes with an analysis of the Lipschitz continuity of this proxy in min-max optimization. We also examine how one player's deviation affects the proxy of the other player.

- Appendix C, we study relationship of Hidden Strong Convexity (HSC) and the equivalent conditions of the proximal Polyak-Łojasiewicz (pPL) and the Kurdyka-Łojasiewicz (KL).

- In Appendix D, we present key results, including: (i) The Lipschitz continuity of maximizers, (ii) Convergence rates for nested and alternating schemes under the general framework of hidden-convex hidden-concave min-max optimization.

- Finally, in Appendix E, we extend our analysis to convex multi-agent reinforcement learning, combining the results from Appendix D with policy gradient estimators to compute parameter updates.

## A. Further Related Work

### A.1. Hidden Convex Optimization

Hidden convexity has emerged as an important structural property in nonconvex optimization, enabling global convergence results in settings where traditional convex analysis fails—a survey can be found here (Xia, 2020). Fatkhullin et al. (2023) have decisively proven the convergence of first-order methods even for the nonsmooth and stochastic settings. Moreover, hidden convexity has been extensively studied across diverse applications, including policy optimization in reinforcement learning (RL) and optimal control (Hazan et al., 2019; Zhang et al., 2020; Ying et al., 2023), generative models (Kobyzev et al., 2020), supply chain and revenue management (Feng & Shanthikumar, 2018; Chen et al., 2024), and neural network training (Wang et al., 2021). Even earlier, instances of an implicit convex structure have appeared in a number of works in the past (Ben-Tal & Teboulle, 1996; Li et al., 2005; Wu et al., 2007). In a certain sense, hidden convexity falls into the general category of metric regularity conditions (Karimi et al., 2016; Li & Pong, 2018; Drusvyatskiy & Paquette, 2019; Drusvyatskiy & Lewis, 2018; Liao et al., 2024; Rebjock & Boumal, 2024; Luo & Tseng, 1993; Oikonomidis et al., 2025) that have been used to prove convergence of iterative gradient-based methods.

Particularly, hidden convexity ensures convergence in nonconvex-nonconcave games. Namely, Vlatakis-Gkaragkounis et al. (2021) introduced the notion of hidden-convex hidden-concave games and proved global convergence to a NE. Further extending these ideas, (Sakos et al., 2024) investigates the impact of hidden structure and show that such properties can be leveraged to enhance the stability of first-order methods. The generalized notion of hidden *monotonicity* (Mladenovic et al., 2021) guarantees global convergence for multi-player nonconvex games.

### A.2. Min-Max Optimization

The literature of min-max optimization is long-standing and intimately connected to game theory (v. Neumann, 1928). In recent years, the exploration of nonconvex-nonconcave min-max problems gained prominence in machine learning, particularly due to developments like generative adversarial networks (GANs) (Goodfellow et al., 2014) and adversarial learning (Madry et al., 2017). Research has focused on defining appropriate solution concepts and developing methods to mitigate oscillatory behaviors in optimization algorithms. A min-max optimization problem is often formulated as a *variational inequality* problem (VIP) (Facchinei & Pang, 2003) and the literature has managed to guarantee provable convergence to solutions of the corresponding VIP only under certain assumptions. Namely, convergence to an approximate solution of the VIP point is guaranteed under (i) monotonicity, (or, a convex-concave objective) (ii) a gradient domination, and (iii) other regularity conditions.

In particular, *under monotonicity*, the proximal point methods (Martinet, 1970; Rockafellar, 1976) for VIP guarantee convergence. When the objective function is Lipschitz and strictly convex-concave, simple forward-backward schemes are known to converge. Moreover, when coupled with Polyak–Ruppert averaging (Ruppert, 1988; Polyak & Juditsky, 1992; Nemirovski et al., 2009), these methods achieve an $O(1/\epsilon^2)$ complexity without requiring strict convex-concavity (Bauschke & Combettes, 2011). If additionally the objective function has Lipschitz continuous gradients, the extragradient algorithm (Korpelevich, 1976) ensures trajectory convergence without strict monotonicity assumptions, while the time-averaged iterates converge at a rate of $O(1/\epsilon)$ (Nemirovski, 2004). Furthermore, in strongly convex-concave settings, forward-backward methods compute an $\epsilon$-saddle point in $O(1/\epsilon)$ steps. If the operator is also Lipschitz continuous, classical results in operator theory establish that simple forward-backward methods are sufficient to achieve linear convergence (Facchinei & Pang, 2007; Bauschke & Combettes, 2011).

Under a single-sided gradient domination condition, convergence to a stationary point of the min-max objective has been proven in (Lin et al., 2020; Nouiehed et al., 2019; Yang et al., 2022). Under a two-sided gradient domination condition, (Daskalakis et al., 2020; Yang et al., 2020; Zheng et al., 2023) have proven the convergence to a solution of the corresponding

VIP (and equivalently to a min-max point). Further, under the Minty condition, (Mertikopoulos et al., 2018; Liu et al., 2021; Diakonikolas et al., 2021) show convergence to solution of the VIP even under nonconvexity. Evenmore, (Azizian et al., 2024) shows convergence of gradient descent under various conditions of regularity and local optimization geometry.

### A.3. Convex Markov Decision Processes

The study of convex reinforcement learning emerged as a natural extension of standard RL to handle more expressive, non-linear utility functions. Hazan et al. (2019) introduced the problem of reward-free exploration of an MDP through state-occupancy entropy maximization. To tackle policy optimization, they proposed a provably efficient algorithm based on the Frank-Wolfe method. Further advances were made by Zhang et al. (2020; 2021), who studied convex RL under the framework of RL with general utilities. Their key contribution was a policy gradient estimator for general utilities and the identification of a hidden convexity property within the convex RL objective, enabling statistically efficient policy optimization in the infinite-trials setting. More recently, in (Zahavy et al., 2021; Geist et al., 2022) convex RL is reinterpreted through a game-theoretic perspective. The former work views convex RL as a min-max game between a policy player and a cost player, while the latter positioned convex RL as a subclass of mean-field games.

A related strand of research focuses on the expressivity of scalar (Markovian) rewards. Abel et al. (2021) demonstrated that scalar rewards cannot naturally encode all tasks, such as policy ordering or trajectory ranking. While convex RL extends the expressivity of scalar RL in these aspects, it still has inherent limitations. Specifically, infinite-trial convex RL excels at defining policy orderings but lacks full trajectory ordering capabilities, as it only considers the stationary state distribution. In contrast, the finite-trials convex RL formulation presented in this paper naturally captures trajectory orderings at the cost of reduced expressivity in policy ordering.

Another relevant direction concerns RL with trajectory feedback, where learning occurs through entire sequences rather than scalar rewards. Most prior works in this area assume an underlying scalar reward model, which merely delays feedback until the episode's end (Efroni et al., 2021). A notable exception is the once-per-episode feedback model studied by Chatterji et al. (2021). Lastly, related research in multi-objective RL has explored the use of vectorial rewards to encode convex objectives. The works of Cheung (2019a;b) demonstrated that stationary policies are often suboptimal in such settings, necessitating non-stationary strategies. They provided principled procedures to optimize policies with sub-linear regret, complementing our analysis in infinite-horizon convex RL, where distinctions between finite and infinite trials diminish.

### A.4. Markov Games and Multi-Agent Reinforcement Learning

Markov, or *stochastic*, games, were introduced by Lloyd S. Shapley (Shapley, 1953). Interestingly enough, their introduction coincides with that of single-agent of Markov decision processes (Schneider & Wagner, 1957; Bellman, 1958). In an MG, agents interact with both the environment and each other. Each agent must balance immediate rewards against the potential future benefits of guiding the system to more advantageous states. Since their inception, MGs have served as the canonical model of MARL (Littman, 1994) which in turn encompasses numerous applications like autonomous vehicles (Shalev-Shwartz et al., 2016), multi-agent robotics (Singh et al., 2022; Gronauer & Diepold, 2022), and general recreational game-playing (Mnih et al., 2013; Silver et al., 2017; Berner et al., 2019).

In recent years, theoretical MG research has focused on developing computationally and statistically efficient algorithms. The literature has experienced a rapid development making an exhaustive review infeasible in the context of this small note. We outline results regarding (i) the computational complexity of equilibrium computation in MGs, (ii) "no-regret" learning approaches with convergence to the corresponding equilibrium notion, (iii) convergence to a NE when the game's structure permits it, and (iv) offline equilibrium learning from data.

A number of works (Deng et al., 2023; Jin et al., 2022; Daskalakis et al., 2023) concurrently proved that (coarse) correlated equilibria are intractable in infinite-horizon MG when the policies are required to be stationary and Markovian; computing them is PPAD-complete. This is a pronounced contrast to normal-form games where they are computable in strongly polynomial time. In terms of "no-regret" approaches, the literature has been quite fruitful, as it considers finite-horizon games and policies that are stationary and both Markovian and non-Markovian. The solutions (Jin et al., 2021; Yang & Ma, 2022; Zhang et al., 2022a; Erez et al., 2023; Cai et al., 2024) are diverse and build upon the *follow-the-regularized-leader* and *online-mirror-descent* framework. Nash equilibrium convergence has mainly considered fully competitive two-player zero-sum MGs of infinite horizon, the Markovian counterpart of potential games, or their unification (adversarial team Markov games) and games with certain structural properties on the rewards and the dynamics. In two-player zero-sum games there have been results with a mostly stochastic-optimization approach (Daskalakis et al., 2020; Wei et al., 2021;

Sayin et al., 2021; Cen et al., 2022; Zeng et al., 2022). In addition, a notable work guarantees convergence to a NE in infinite-horizon games using only bandit feedback (Cai et al., 2023). Then, for Markov potential games, global convergence of policy gradient methods has been presented in (Zhang et al., 2022b; Leonardos et al., 2021; Ding et al., 2022). Moreover, (Kalogiannis et al., 2022; 2024) consider an MG of a "team" versus an "adversary" similar to (Von Stengel & Koller, 1997) and guarantee provable convergence to NE. Lastly, (Kalogiannis & Panageas, 2023; Park et al., 2023) generalize zero-sum polymatrix game to their Markovian counterpart with extra assumptions on the dynamics, and prove the tractability of NEs. In terms of learning equilibria from samples, (Bai & Jin, 2020; Cui & Du, 2022) guaranteed sample-efficient learning of NE in zero-sum games and (Song et al., 2021) went further to consider general-sum games. Finally, (Wang et al., 2023; Cui et al., 2023) consider the sample complexity of learning an equilibrium beyond the tabular setting.

# B. Optimization Preliminaries

In this section we go over some rudimentary optimization lemmata as well as some novel exploration of the min-max optimization landscape for pPL functions. Namely, Appendix B.2 compliments the study of (Nouiehed et al., 2019) for the constrained domain. To our knowledge, we offer the first such investigation of the constrained pPL landscape.

## B.1. Optimization Definitions & Lemmata

**Lemma B.1** (Smoothness inequality). *Assume that $f : \mathcal{X} \to \mathbb{R}$ is an $\ell$-smooth function. Then, for $x, y \in \mathcal{X}$ it holds that:*

$$f(y) + \langle \nabla f(y), y - x \rangle - \frac{\ell}{2} \|y - x\|^2 \leq f(x) \leq f(y) + \langle \nabla f(y), y - x \rangle + \frac{\ell}{2} \|y - x\|^2.$$

**Definition 7.** A function $f$ is said to be $\ell$-*weakly convex* if $f(x) + \frac{\ell}{2} \|x\|^2$ is convex.

### B.1.1. CONCERNING THE GRADIENT MAPPING

**Lemma B.2.** *Let $\mathcal{X} \subseteq \mathbb{R}^d$ be a non-empty, closed, and convex set. Denote by $\Pi_{\mathcal{X}} : \mathbb{R}^d \to \mathcal{X}$ the Euclidean projection operator onto $\mathcal{X}$. The following inequalities hold,*

• *for projected gradient descent:*

$$\langle v, x - \Pi_{\mathcal{X}} (x - \eta v) \rangle \geq \frac{1}{\eta} \|x - \Pi_{\mathcal{X}} (x - \eta v)\|^2;$$

• *for projected gradient ascent:*

$$\langle v, \Pi_{\mathcal{X}} (x + \eta v) - x \rangle \geq \frac{1}{\eta} \|x - \Pi_{\mathcal{X}} (x + \eta v)\|^2.$$

*Proof.* After writing the Euclidean projection of gradient descent with feedback $v$ as an optimization problem:

$$\min_{x' \in \mathcal{X}} \frac{1}{2} \|x' - (x - \eta v)\|^2,$$

we get the first-order optimality conditions,

$$\langle \eta v + \Pi_{\mathcal{X}} (x - \eta v) - x, z - \Pi_{\mathcal{X}} (x - \eta v) \rangle \geq 0, \quad \forall z \in \mathcal{X}.$$

Setting $z = x$ and re-arranging,

$$\langle v, z - \Pi_{\mathcal{X}} (x - \eta v) \rangle \geq \frac{1}{\eta} \|\Pi_{\mathcal{X}} (x - \eta v) - x\|^2.$$

Similarly for gradient ascent with feedback $v$ we write,

$$\min_{x' \in \mathcal{X}} \frac{1}{2} \|x' - (x + \eta v)\|^2$$

The first-order optimality condition for the Euclidean projection, reads,

$$\langle x' - x - \eta v, z - x' \rangle \geq 0, \quad \forall z \in \mathcal{X}.$$

Plugging-in $x' = \Pi_{\mathcal{X}}(x + \eta v)$,

$$\langle \Pi_{\mathcal{X}}(x + \eta v) - x - \eta v, x - \Pi_{\mathcal{X}}(x + \eta v) \rangle \geq 0.$$

Re-arranging we conclude,

$$\langle v, \Pi_{\mathcal{X}}(x + \eta v) - x \rangle \geq \frac{1}{\eta} \|\Pi_{\mathcal{X}}(x + \eta v) - x\|^2.$$

$\square$

**Lemma B.3** ((Ghadimi et al., 2016, Lemma 2)). *Let $v_1, v_2$ be vectors in $\mathbb{R}^d$ and $\mathcal{X} \subseteq \mathbb{R}^d$ be a compact convex set and a scalar $\eta > 0$. Also, let points $x_1^+, x_2^+ \in \mathcal{X}$ such that:*

$$x_1^+ := \Pi_{\mathcal{X}}(x - \eta v_1);$$
$$x_2^+ := \Pi_{\mathcal{X}}(x - \eta v_2).$$

*Then, it holds true that:*

$$\left\| x_1^+ - x_2^+ \right\| \leq \eta \|v_1 - v_2\|.$$

.

**Lemma B.4** (Stoch. vs. Det. Grad. Mapping). *Assume a stochastic gradient oracle $\hat{g}_x$ for a differentiable function $f$. For the stochastic gradient oracle, it holds that $\mathbb{E}[\hat{g}_x(x)] = g_x(x)$, $\mathbb{E}[\|\hat{g}_x\|^2] \leq \sigma_x^2$, and $\|g_x(x) - \nabla f(x)\| < \delta_x$ for all $x \in \mathcal{X}$. Also, let $x, x^+, \hat{x}^+$ be points in $\mathcal{X}$ such that:*

$$x^+ := \Pi_{\mathcal{X}}(x - \eta \nabla_x f(x));$$
$$\hat{x}^+ := \Pi_{\mathcal{X}}(x - \eta \hat{g}_x(x)).$$

*Then, the following inequalities hold:*

$$\left\| x - \hat{x}^+ \right\|^2 \leq 2 \left\| x - x^+ \right\|^2 + 4\eta^2 \sigma_x^2 + 4\eta^2 \delta_x^2;$$
$$\left\| x - x^+ \right\|^2 \leq 2 \left\| x - \overline{x}^+ \right\|^2 + 4\eta^2 \sigma_x^2 + 4\eta^2 \delta_x^2.$$

*Proof.* We will prove the first inequality; the second follows from the same arguments.

$$\begin{aligned}
\mathbb{E} \left\| x - \hat{x}^+ \right\|^2 &\leq 2\mathbb{E} \left\| x - x^+ \right\|^2 + 2\mathbb{E} \left\| \hat{x}^+ - x^+ \right\|^2 \\
&\leq 2\mathbb{E} \left\| x - x^+ \right\|^2 + 2\eta^2 \mathbb{E} \|\hat{g}_x(x) - \nabla_x f(x)\|^2 \\
&\leq 2\mathbb{E} \left\| x - x^+ \right\|^2 + 4\eta^2 \mathbb{E} \|g_x(x) - \hat{g}_x(x)\|^2 + 4\eta^2 \mathbb{E} \|g_x(x) - \nabla_x f(x)\|^2 \\
&\leq 2\mathbb{E} \left\| x - x^+ \right\|^2 + 4\eta^2 \sigma_x^2 + 4\eta^2 \delta_x^2
\end{aligned}$$

The first and second inequalities hold from the fact $|a + b|^2 \leq 2|a|^2 + 2|b|^2$. The last inequality comes from the properties of the stochastic gradient oracle. $\square$

### B.1.2. $\mathcal{D}_\mathcal{X}$: AN ALTERNATIVE PROXY OF STATIONARITY

In unconstrained optimization of differentiable functions, the conventional bounds of optimization algorithms directly guarantee the minimization of $\|\nabla f(x)\|$, *i.e.*, the norm of the gradient of $f$ at point $x$. In constrained optimization of differentiable functions, guarantees for algorithms like projected gradient descent ensure the minimization of the norm of the *gradient mapping*, *i.e.*, $\frac{1}{\eta}\|x - \Pi_\mathcal{X}(x - \eta\nabla f(x))\|$. It can be shown that the gradient mapping is indeed a good proxy of stationarity. For our work, we will consider the quantity $\mathcal{D}_\mathcal{X}$ to be defined shortly. This quantity is greater than the squared norm of the gradient mapping and turns out to be particularly favorable when handling the constrained optimization of proximal-PL functions. In what follows, we will define $\mathcal{D}_\mathcal{X}$ and discuss some of its useful properties.

**Definition 8.** Let $\mathcal{X} \subseteq \mathbb{R}^d$ be a compact convex set and $f : \mathcal{X} \to \mathbb{R}$ be an $\ell$-smooth function. We define $\mathcal{D}_\mathcal{X}(x, \ell)$ to be:

$$\mathcal{D}_\mathcal{X}(x, \alpha) = -2\alpha \min_{y \in \mathbb{R}^d}\left\{\langle \nabla f(x), y - x\rangle + \frac{\alpha}{2}\|y - x\|^2 + I_\mathcal{X}(y) - I_\mathcal{X}(x)\right\}.$$

Where $I_\mathcal{X}$ is the indicator function of the set $\mathcal{X}$ with $I_\mathcal{X}(x) = 0$ if $x \in \mathcal{X}$ and $I_\mathcal{X}(x) = +\infty$ otherwise.

Remarkably, this expression can take a closed-form. As we will show, the minimizer of the display inside the brackets is exactly the point returned by one step of projected gradient descent on the argument $x$ with a stepsize equal to $\frac{1}{\alpha}$.

**Claim B.5** (Closed form of $\mathcal{D}_\mathcal{X}$). *Let $f$ be an $\ell$-smooth function defined on a compact convex set $\mathcal{X} \subseteq \mathbb{R}^d$, a point $x \in \mathcal{X}$, and a scalar $\alpha \geq \ell$. Then the following equation holds for $\mathcal{D}_\mathcal{X}(x, a)$:*

$$\mathcal{D}_\mathcal{X}(x, \alpha) = 2\alpha\left\langle \nabla f(x), x - \Pi_\mathcal{X}\left(x - \frac{1}{\alpha}\nabla f(x)\right)\right\rangle - \alpha^2\left\|x - \Pi_\mathcal{X}\left(x - \frac{1}{\alpha}\nabla f(x)\right)\right\|^2.$$

*Proof.* By first-order optimality conditions, we see that the objective is equivalent to the objective of the Euclidean projection of the point $\left(x - \frac{1}{\alpha}\nabla f(x)\right) \in \mathbb{R}^d$ to $\mathcal{X}$. Then, we just plug the minimizer into the display. $\square$

Next, we see that $\mathcal{D}(\cdot, \alpha)$ is non-decreasing in the scalar $\alpha$.

**Lemma B.6** ((Karimi et al., 2016, Lemma 1)). *Let a differentiable functions $f : \mathcal{X} \to \mathbb{R}$ and positive scalars $0 < \alpha_1 \leq \alpha_2$. Then, the following inequality holds true:*

$$\mathcal{D}_\mathcal{X}(x, \alpha_1) \leq \mathcal{D}_\mathcal{X}(x, \alpha_2).$$

The following Lemma demonstrates the relationship between $\mathcal{D}_\mathcal{X}$ and the norm of the gradient mapping.

**Lemma B.7** ($\mathcal{D}_\mathcal{X}$ vs. Gradient Mapping Norm). *Let $\mathcal{X} \subseteq \mathbb{R}^d$ be a non-empty, closed, and convex set. Consider a differentiable function $f : \mathcal{X} \to \mathbb{R}$ that has an $\ell$-Lipschitz continuous gradient. Define the* gradient mapping *at point $x \in \mathcal{X}$ with step size $1/\ell$ as,*

$$\ell^2(x - x^+)$$

*where,*

$$x^+ := \Pi_\mathcal{X}\left(x - \frac{1}{\ell}\nabla f(x)\right).$$

*Then, the following inequality holds:*

$$\mathcal{D}_\mathcal{X}(x, \ell) \geq \ell^2\|x^+ - x\|^2.$$

*Proof.* By observing the definition of $\mathcal{D}_\mathcal{X}$, we observe that first-order optimality for its inner minimization problem read,

$$\langle \nabla f(x) + \ell(y - x), z - y\rangle \geq 0, \quad \forall z \in \mathcal{X}.$$

On closer inspection, we recognize the first-order optimality of the Euclidean projection of one gradient descent step, *i.e.*,

$$x^+ = \arg\min_{y \in \mathcal{Y}}\left\{\langle \nabla f(x), y - x\rangle + \frac{\ell}{2}\|y - x\|^2\right\}.$$

Plugging-in $x^+$:

$$\mathcal{D}_{\mathcal{X}}(x, \ell) = -2\ell \left\langle \nabla f(x), x^+ - x \right\rangle - \ell^2 \|x^+ - x\|^2$$
$$= 2\ell \left\langle \nabla f(x), x - x^+ \right\rangle - \ell^2 \|x^+ - x\|^2$$
$$\geq \ell^2 \left\| x^+ - x \right\|^2,$$

where the inequality follows from Lemma B.2. $\qquad \square$

**Lemma B.8** ((J Reddi et al., 2016, Lemma 6)). *Let $f : \mathcal{X} \to \mathbb{R}$ be an $\ell$-smooth function and a point $x \in \mathcal{X} \subseteq \mathbb{R}^d$. Also, define the vector $v \in \mathbb{R}^d$ and $y \in \mathcal{X}$ to be*

$$y := \Pi_{\mathcal{X}}\left(x - \eta v\right).$$

*Then, the following inequality is true:*

$$f(y) \leq f(z) + \langle \nabla f(x) - v, y - z \rangle$$
$$+ \left(\frac{\ell}{2} - \frac{1}{2\eta}\right) \|y - x\|^2 + \left(\frac{\ell}{2} + \frac{1}{2\eta}\right) \|z - x\|^2 - \frac{1}{2} \|y - z\|^2 .$$

### B.1.3. A DESCENT LEMMA INVOLVING $\mathcal{D}_{\mathcal{X}}$

**Lemma B.9.** *Let $\mathcal{X} \subseteq \mathbb{R}^d$ be a closed convex set, and let $f : \mathcal{X} \to \mathbb{R}$ be an $\ell$-smooth function for some $\ell > 0$. Suppose $\eta > 0$ with $\eta \leq \frac{1}{5\ell}$. For any $x \in \mathcal{X}$ and any vector $v \in \mathbb{R}^d$, define $x^+ = \Pi_{\mathcal{X}}\left(x - \eta v\right)$. Then the following inequality holds:*

$$f(x^+) \leq f(x) - \frac{\eta}{6} \mathcal{D}_{\mathcal{X}}(x, 1/\eta) + \frac{\eta}{2} \|\nabla f(x) - v\|^2$$

*Proof.* First, we define $\overline{x}^+ := \Pi_{\mathcal{X}}\left(x - \frac{1}{\alpha}\nabla f(x)\right)$.

- Invoking $\ell$-smoothness of $f$ for $x, \overline{x}_+$ and assuming $\alpha > 0$ with $\alpha \geq \ell$,

$$f(\overline{x}_+) \leq f(x) + \langle \nabla f(x), \overline{x}_+ - x \rangle + \frac{\ell}{2} \|x_+ - x\|^2$$
$$\leq f(x) + \langle \nabla f(x), \overline{x}_+ - x \rangle + \frac{\alpha}{2} \|x_+ - x\|^2$$
$$= f(x) - \left( \langle \nabla f(x), x - \overline{x}_+ \rangle - \frac{\alpha}{2} \|x_+ - x\|^2 \right)$$
$$= f(x) - \frac{1}{2\alpha} \mathcal{D}_{\mathcal{X}}(x, \alpha). \tag{1}$$

- Invoking Lemma B.8 with $x = x$, $y = \overline{x}_+$, $z = x$, $v = \nabla f(x)$

$$f(\overline{x}_+) \leq f(x) + \left(\frac{\ell}{2} - \frac{1}{\alpha}\right) \|\overline{x}_+ - x\|^2 . \tag{2}$$

- Again, invoking Lemma B.8 but with $x = x$, $y = x_+$, $z = \overline{x}_+$, $v$,

$$f(x_+) \leq f(\overline{x}_+) + \langle \nabla f(x) - v, x_+ - \overline{x}_+ \rangle$$
$$+ \left(\frac{\ell}{2} - \frac{1}{2\eta}\right) \|x_+ - x\|^2 + \left(\frac{\ell}{2} + \frac{1}{2\eta}\right) \|\overline{x}_+ - x\|^2 - \frac{1}{2\eta} \|x_+ - \overline{x}_+\|^2 . \tag{3}$$

Adding $1/3\times$(1) and $2/3\times$(2) and letting $1/\alpha = \eta \leq \frac{1}{\ell}$

$$f(\overline{x}_+) \leq f(x) - \frac{1}{6\eta} \mathcal{D}_{\mathcal{X}}(x, 1/\eta) + \left(\frac{\ell}{3} - \frac{2}{3\eta}\right) \|\overline{x}_+ - x\|^2$$

Adding (3),

$$
\begin{aligned}
f(x_+) \leq{} & f(x) - \frac{\eta}{6}\mathcal{D}_{\mathcal{X}}(x, 1/\eta) + \left(\frac{\ell}{3} - \frac{2}{3\eta}\right)\|\overline{x}_+ - x\|^2 \\
& + \langle \nabla f(x) - v, x_+ - \overline{x}_+ \rangle \\
& + \left(\frac{\ell}{2} - \frac{1}{2\eta}\right)\|x_+ - x\|^2 + \left(\frac{\ell}{2} + \frac{1}{2\eta}\right)\|\overline{x}_+ - x\|^2 - \frac{1}{2\eta}\|x_+ - \overline{x}_+\|^2 \\
\leq{} & f(x) - \frac{\eta}{6}\mathcal{D}_{\mathcal{X}}(x, 1/\eta) + \left(\frac{5\ell}{6} - \frac{1}{6\eta}\right)\|\overline{x}_+ - x\|^2 \\
& + \frac{\rho}{2}\|\nabla f(x) - v\|^2 + \frac{1}{2\rho}\|x_+ - \overline{x}_+\|^2 \\
& + \left(\frac{\ell}{2} - \frac{1}{2\eta}\right)\|x_+ - x\|^2 - \frac{1}{2\eta}\|x_+ - \overline{x}_+\|^2 \\
={} & f(x) - \frac{\eta}{6}\mathcal{D}_{\mathcal{X}}(x, 1/\eta) + \left(\frac{5\ell}{6} - \frac{1}{6\eta}\right)\|\overline{x}_+ - x\|^2 \\
& + \frac{\eta}{2}\|\nabla f(x) - v\|^2 \\
& + \left(\frac{\ell}{2} - \frac{1}{2\eta}\right)\|x_+ - x\|^2 \\
\leq{} & f(x) - \frac{\eta}{6}\mathcal{D}_{\mathcal{X}}(x, 1/\eta) + \frac{\eta}{2}\|\nabla f(x) - v\|^2
\end{aligned}
$$

$$(4)$$

$$(5)$$

- (4) follows from the application of Young's inequality;
- (5) follows by dropping the non-positive terms; non-positivity follows from the choice of the step-size, $\eta \leq \frac{1}{5\ell}$.

$\square$

## B.2. Min-Max Optimization Lemmata

The following claims are novel to our knowledge. They justify the intuition that the constrained min-max pPL landscape should resemble its unconstrained PL counterpart. For example, the trivial bound $\|\nabla_x f(x,y) - \nabla_x f(x, y')\|^2 \leq \ell^2 \|y - y'\|$ of the unconstrained setting takes the form Lemma B.12.

**Claim B.10.** *Let $\mathcal{X}$ and $\mathcal{Y}$ be convex and compact sets. Suppose that the function $f : \mathcal{X} \times \mathcal{Y} \to \mathbb{R}$ is $\ell$-smooth. For given points $y_1, y_2 \in \mathcal{Y}$, define $x_1$ and $x_1^+$ as follows:*

- $x_1 := \arg\min_{x \in \mathcal{X}} f(x, y_1)$, *and*
- $x_1^+ := \Pi_{\mathcal{X}}(x_1 - \eta\nabla_x f(x_1, y_2))$.

*Then, then it holds true that:*

$$
\frac{1}{\eta}\left\|x_1 - x_1^+\right\| \leq \ell\left\|y_1 - y_2\right\|.
$$

*Symmetrically, for given points $x_1, x_2 \in \mathcal{X}$, if $y_1 := \arg\max_{y \in \mathcal{Y}} f(x_1, y_1)$, $y_1^+ := \Pi_{\mathcal{Y}}(y_1 + \eta\nabla_y f(x_2, y_1))$,*

$$
\frac{1}{\eta}\left\|y_1 - y_1^+\right\| \leq \ell\left\|x_1 - x_2\right\|.
$$

*Proof.* By optimality of $x_1, x_1^+$ for their corresponding optimization problems:

$$
\left\langle \nabla f(x_1, y_1), x_1^+ - x_1 \right\rangle \geq 0 \tag{6}
$$

$$
\left\langle \nabla f(x_1, y_2), x_1 - x_1^+ \right\rangle \geq \frac{1}{\eta}\left\|x_1 - x_1^+\right\|^2 \tag{7}
$$

Adding-and-subtracting $\nabla_x f(x_1, y_2)$ to Equation (6),

$$
\begin{aligned}
0 &\leq \left\langle \nabla f(x_1, y_1), x_1^+ - x_1 \right\rangle \\
&= \left\langle \nabla f(x_1, y_1) - \nabla f(x_1, y_2) + \nabla f(x_1, y_2), x_1^+ - x_1 \right\rangle \\
&= \left\langle \nabla f(x_1, y_1) - \nabla f(x_1, y_2), x_1^+ - x_1 \right\rangle + \left\langle \nabla f(x_1, y_2), x_1^+ - x_1 \right\rangle.
\end{aligned} \tag{8}
$$

Hence, combining Equation (8) with Equation (7),

$$
\begin{aligned}
\left\langle \nabla f(x_1, y_1) - \nabla f(x_1, y_2), x_1^+ - x_1 \right\rangle &\geq \left\langle \nabla f(x_1, y_2), x_1 - x_1^+ \right\rangle \\
&\geq \frac{1}{\eta} \left\| x_1 - x_1^+ \right\|^2
\end{aligned}
$$

By Cauchy-Schwarz,

$$
\|\nabla f(x_1, y_1) - \nabla f(x_1, y_2)\| \left\| x_1^+ - x_1 \right\| \geq \frac{1}{\eta} \left\| x_1 - x_1^+ \right\|^2
$$

Finally by Lispchitz continuity of $\nabla f(x, y)$,

$$
\ell \left\| y_1 - y_2 \right\| \geq \frac{1}{\eta} \left\| x_1 - x_1^+ \right\|.
$$

This completes the proof for a step of projected gradient descent. The arguments for the projected gradient ascent claim are identical. $\qquad \square$

**Lemma B.11.** *Assume $f : \mathcal{X} \times \mathcal{Y}$ an $\ell$-smooth function, and a scalar $a > 0$, two points $y, y' \in \mathcal{Y}$ and $x = \arg\min_{z \in \mathcal{X}} f(z, y)$. It is true that:*

$$
2\ell^2 \left\| y - y' \right\|^2 \geq \mathcal{D}_{\mathcal{X}}(x, \alpha; y').
$$

*Proof.* Let $x^+ := \arg\max_{\overline{x} \in \mathcal{X}} \left\{ \langle \nabla f(x, y'), x - \overline{x} \rangle - \frac{\alpha}{2} \|x - \overline{x}\|^2 \right\} = \Pi_{\mathcal{X}} \left( x - \frac{1}{\alpha} \nabla f(x, y') \right).$

$$
\begin{aligned}
\frac{1}{2\alpha} \mathcal{D}_{\mathcal{X}}(x, y', \alpha) &= \left\langle \nabla f(x, y'), x - x^+ \right\rangle - \frac{\alpha}{2} \left\| x - x^+ \right\|^2 \\
&= \left\langle \nabla f(x, y') - \nabla f(x, y), x - x^+ \right\rangle - \frac{\alpha}{2} \left\| x - x^+ \right\|^2 + \left\langle \nabla f(x, y), x - x^+ \right\rangle
\end{aligned} \tag{9}
$$

We have assumed that $x = \arg\min_{\overline{x} \in \mathcal{X}} f(\overline{x}, y)$, therefore,

$$
\langle \nabla f(x, y), z - x \rangle \geq 0 \quad \forall z \in \mathcal{X}.
$$

As such, when $z = x^+$,

$$
\left\langle \nabla f(x, y), x - x^+ \right\rangle \leq 0.
$$

Hence, since $\frac{-1}{\eta} \left\| x - x^+ \right\|^2, \left\langle \nabla f(x, y), x - x^+ \right\rangle \leq 0$ plugging in (9),

$$
\left\langle \nabla f(x, y') - \nabla f(x, y), x - x^+ \right\rangle \geq \frac{1}{2\alpha} \mathcal{D}_{\mathcal{X}}(x, \alpha; y').
$$

Using Cauchy-Schwarz and $\ell$-Lipschitz continuity of the gradient,

$$
\ell \left\| y - y' \right\| \left\| x - x^+ \right\| \geq \frac{1}{2\alpha} \mathcal{D}_{\mathcal{X}}(x, \alpha; y').
$$

Finally, using Claim B.10 with $\eta = \frac{1}{\alpha}$,

$$
\frac{\ell^2}{\alpha} \left\| y - y' \right\|^2 \geq \frac{1}{2\alpha} \mathcal{D}_{\mathcal{X}}(x, \alpha; y').
$$

$\qquad \square$

**Lemma B.12.** *Let $f : \mathcal{X} \times \mathcal{Y}$ be an $\ell$-smooth function, $a > 0$, two points $y, y' \in \mathcal{Y}$, and a point $x \in \mathcal{X}$. Then, the following inequality holds:*

$$|\mathcal{D}_{\mathcal{X}}(x, a; y) - \mathcal{D}_{\mathcal{X}}(x, a; y')| \le 3\ell^2 \|y - y'\|^2.$$

*Proof.* We define $\overline{x}, \overline{x}' \in \mathcal{X}$ to be:

$$\overline{x} := \Pi_{\mathcal{X}} \left( x - \frac{1}{\alpha} \nabla_x f(x, y) \right);$$

$$\overline{x}' := \Pi_{\mathcal{X}} \left( x - \frac{1}{\alpha} \nabla_x f(x, y') \right).$$

By the definition of $\mathcal{D}_{\mathcal{X}}(x, \alpha; y')$ we write:

$$\frac{1}{2\alpha} \mathcal{D}_{\mathcal{X}}(x, \alpha; y) = \langle \nabla f(x, y), x - \overline{x} \rangle - \frac{\alpha}{2} \|x - \overline{x}\|^2$$

$$\frac{1}{2\alpha} \mathcal{D}_{\mathcal{X}}(x, \alpha; y') = \langle \nabla f(x, y'), x - \overline{x}' \rangle - \frac{\alpha}{2} \|x - \overline{x}'\|^2.$$

Considering the difference $\mathcal{D}_{\mathcal{X}}(x, \alpha; y) - \mathcal{D}_{\mathcal{X}}(x, \alpha; y')$ we see that:

$$\begin{aligned}
\frac{1}{2\alpha} |\mathcal{D}_{\mathcal{X}}(x, \alpha; y) - \mathcal{D}_{\mathcal{X}}(x, \alpha; y')| &= \left| \langle \nabla_x f(x, y) - \nabla_x f(x, y'), \overline{x}' - \overline{x} \rangle - \frac{\alpha}{2} \left( \|x - \overline{x}\|^2 - \|x - \overline{x}'\|^2 \right) \right| \\
&\le |\langle \nabla_x f(x, y) - \nabla_x f(x, y'), \overline{x}' - \overline{x} \rangle| + \frac{\alpha}{2} \left| \left( \|x - \overline{x}\|^2 - \|x - \overline{x}'\|^2 \right) \right| \\
&\le |\langle \nabla_x f(x, y) - \nabla_x f(x, y'), \overline{x}' - \overline{x} \rangle| + \frac{\alpha}{2} \|\overline{x} - \overline{x}'\|^2 \\
&\le \|\nabla_x f(x, y) - \nabla_x f(x, y')\| \|\overline{x}' - \overline{x}\| + \frac{\alpha}{2} \|\overline{x} - \overline{x}'\|^2 \\
&\le \frac{1}{\alpha} \|\nabla_x f(x, y) - \nabla_x f(x, y')\|^2 + \frac{1}{2\alpha} \|\nabla_x f(x, y) - \nabla_x f(x, y')\|^2 \\
&\le \frac{3\ell^2}{2\alpha} \|y - y'\|^2.
\end{aligned}$$

We note that:

- The first inequality follows from the triangle inequality.
- In the second, we apply the reverse triangle inequality.
- The third inequality uses the Cauchy-Schwarz inequality.
- The penultimate uses Lemma B.3 and the final $\ell$-Lipschitz continuity of the gradient.

$\square$

**Claim B.13.** *Assume a function $f : \mathcal{X} \times \mathcal{Y}$ satisfying the pPL condition with modulus $\mu$ over $f(x, \cdot)$ for any $x \in \mathcal{X}$. Also, assume that $\Phi(x) := \max_{y \in \mathcal{Y}} f(x, y)$. Additionally, define: $\mathcal{D}_{\mathcal{X}}^{\Phi}(x, \alpha) = -2\alpha \min_{z \in \mathcal{X}} \left\{ \langle \nabla \Phi(x), z - x \rangle + \frac{\alpha}{2} \|z - x\|^2 \right\}$. Then, it holds true that,*

$$\left| \mathcal{D}_{\mathcal{X}}^{\Phi}(x, \alpha_1) - \mathcal{D}_{\mathcal{X}}(x, \alpha_2; y) \right| \le 3\ell\kappa^2 \mathcal{D}_{\mathcal{Y}}(y, \alpha_2; x)$$

*where $\kappa := \frac{\ell}{\mu}$.*

*Proof.* We define $\overline{x}, \overline{x}'$ as,

$$\overline{x} := \Pi_{\mathcal{X}} \left( x - \frac{1}{\alpha_1} \nabla \Phi(x) \right);$$

$$\overline{x}' := \Pi_{\mathcal{X}} \left( x - \frac{1}{\alpha_1} \nabla f(x, y) \right).$$

Then, we see that:

$$\frac{1}{2\alpha_1}\left|\mathcal{D}_{\mathcal{X}}^{\Phi}(x,\alpha_1) - \mathcal{D}_{\mathcal{X}}(x,\alpha_2;y)\right| \leq \left|\left(\langle\nabla\Phi(x), x-\overline{x}\rangle - \frac{\alpha_1}{2}\|x-\overline{x}\|^2\right) - \left(\langle\nabla f(x,y), x-\overline{x}'\rangle - \frac{\alpha_1}{2}\|x-\overline{x}'\|^2\right)\right|$$

$$\leq \left|\langle\nabla\Phi(x) - \nabla f(x,y), \overline{x}'-\overline{x}\rangle\right| + \frac{\alpha_1}{2}\left|\|x-\overline{x}\|^2 - \|x-\overline{x}'\|^2\right|$$

$$\leq \|\nabla\Phi(x) - \nabla f(x,y)\|\|\overline{x}'-\overline{x}\| + \frac{\alpha_1}{2}\|\overline{x}-\overline{x}'\|^2$$

$$\leq \frac{3}{2\alpha_1}\|\nabla\Phi(x) - \nabla f(x,y)\|^2$$

$$\leq \frac{3\ell}{2\alpha_1}\|y - y^{\star}(x)\|^2$$

$$\leq \frac{3\ell\kappa^2}{2\alpha_1}\mathcal{D}_{\mathcal{Y}}(y,\alpha_2;x).$$

Where, the first inequality is due to the definition. The second and the third, follow from the triangle inequality and the reverse triangle inequality respectively. The fourth follows from the fact that the projection operator is contractive (1-Lipschitz). The last one is due to the quadratic growth condition and the pPL inequality. $\square$

## C. Hidden Convexity, PL, KL, and EB

In this section, we offer a brief exposition to the notions of hidden convexity and other regularity conditions it is related to. We refert the interested reader to (Karimi et al., 2016; Li & Pong, 2018; Drusvyatskiy & Paquette, 2019; Drusvyatskiy & Lewis, 2018; Liao et al., 2024; Rebjock & Boumal, 2024) and references therein. We commence by defining hidden convexity in the manner that (Fatkhullin et al., 2023) do it.

**Definition 9** (Hidden Convex Problem). Consider the optimization problem

$$\min_{x\in\mathcal{X}} f(x) := H(c(x)).$$

This problem is called hidden convex with moduli $L_{c^{-1}} > 0$ and $\mu_H \geq 0$ (or the function $f$ is hidden convex on $\mathcal{X}$) if the following conditions are satisfied:

(HC.1) Convexity of $H$ and Domain:

- The domain $\mathcal{U} = c(\mathcal{X})$ is convex.
- The function $H : \mathcal{U} \to \mathbb{R}$ satisfies for all $u, v \in \mathcal{U}$ and $0 \leq \alpha \leq 1$,

$$H((1-\alpha)u + \alpha v) \leq (1-\alpha)H(u) + \alpha H(v) - \frac{(1-\alpha)\alpha\mu_H}{2}\|u-v\|^2.$$

- The problem admits a solution $u^{\star} \in \mathcal{U}$.

(HC.2) Invertibility and Lipschitz Continuity of $c^{-1}$:

- The mapping $c : \mathcal{X} \to \mathcal{U}$ is invertible.
- There exists $\mu_c > 0$ such that for all $x, y \in \mathcal{X}$,

$$\|x-y\| \leq \mu_c\|c(x) - c(y)\|.$$

Furthermore, if $\mu_H > 0$, the problem is referred to as $(\mu_c, \mu_H)$-*hidden strongly convex*.

**Assumption 2** (Lipschitzness and smoothnes of $c$). Let $c : \mathcal{X} \times \mathcal{Y} \to \mathcal{U}$ be a mapping such that for all $(x,y), (x',y') \in \mathcal{X} \times \mathcal{Y}$ and all $u, u' \in \mathcal{U}$, the following conditions hold:

$$\|c(x,y) - c(x',y')\| \leq L_c\|(x,y) - (x',y')\|;$$
$$\|c^{-1}(u) - c^{-1}(u')\| \leq L_{c^{-1}}\|u-u'\|;$$

and also,

$$\|\nabla c(x,y) - \nabla c(x',y')\| \leq \ell_c\|(x,y) - (x',y')\|;$$
$$\|\nabla c^{-1}(u) - \nabla c^{-1}(u')\| \leq \ell_{c^{-1}}\|u-u'\|.$$

**Fact C.1.** *A hidden strongly-convex function has a unique maximizer.*

### C.1. Equivalence between the three conditions

Throught this subsection, we will denote $\mathcal{X}^\star$ to be $\mathcal{X}^\star = \arg\min_{x \in \mathcal{X}} f(x)$, $x_p^\star$ w.r.t. to a point $x \in \mathcal{X}$ is defined as some element of $\arg\min_{x' \in \mathcal{X}^\star} \|x - x'\|$. Moreover, we define $F(x) := f(x) + I_\mathcal{X}(x)$ with $I_\mathcal{X}(x) = 0$, if $x \in \mathcal{X}$, and $I_\mathcal{X}(x) = +\infty$, else.

**Definition 10** (pPL, KL, pEB). Let $f : \mathcal{X} \to \mathbb{R}$ be an $L$-Lipschitz continuous function with $\ell$-Lipschitz continuous gradient. Then,

- *Proximal Error-Bound (pEB):* $f$ is said to satisfy the proximal Error-Bound if $\exists c > 0$ s.t.

$$\left\| x - x_p^\star \right\| \leq c \left\| x - \Pi_\mathcal{X}\left( x - \frac{1}{\ell}\nabla f(x) \right) \right\|, \quad \forall x \in \mathcal{X}$$

- *Proximal Polyak-Łojasiewicz (pPL):* $f$ is said to satisfy the proximal Polyak-Łojasiewicz condition if $\exists \mu > 0$ s.t.

$$\frac{1}{2}\mathcal{D}_\mathcal{X}(x, \ell) \geq \mu\left( f(x) - f(x^\star) \right)$$

- *Kurdyaka-Łojasiewicz (KL):* f is said to satisfy if $\exists \overline{\mu}$ s.t.

$$\min_{s_x \in \partial(f + I_\mathcal{X})(x)} \|s_x\|^2 \geq 2\overline{\mu}\left( f(x) - f(x^\star) \right), \quad \forall x \in \mathcal{X}.$$

**Lemma C.1** ((Karimi et al., 2016, Appendix G)).

$$(pPL) \equiv (pEB) \equiv (KL).$$

*Remark* 1. As we will see, (KL) with a modulus $\overline{\mu}$ implies (pEB) with a modulus $1 + \frac{2\ell}{\overline{\mu}}$. In turn, (pEB) with a modulus $c$, implies (pPL) with a modulus $\frac{\ell}{1 + 4c^2}$.

Following, we repeat the calculations of (Karimi et al., 2016) by explicitly tracking the dependence between the constants. We will denote $\mathcal{X}^\star := \arg\min_{x \in \mathcal{X}} f(x)$ and $N_\mathcal{X}(x)$ to be the normal cone of $\mathcal{X}$ at $x$.

**KL→pEB**   First, let $F$ be $F(x) := f(x) + I_\mathcal{X}(x)$. By the $\ell$-smoothness of $f$ and the convexity of $I_\mathcal{X}$, we observe that $F$ is weakly-convex. By (Bolte et al., 2010, Theorem 13), for any $x \in \mathrm{dom}F$ there exists a subgradient curve $\chi_x : [0, \infty) \to \mathrm{dom}F$ that satisfies the following:

$$\dot{\chi}_x(t) \in -\partial F(\chi_x(t)),$$
$$\chi_x(0) = x,$$
$$\frac{d}{dt}F(\chi_x(t)) = -\|\dot{\chi}_x(t)\|^2,$$

where $t \mapsto F(\chi_x(t))$ is a non-increasing and Lipschitz continuous function for $t \in [\eta, \infty]$ for any $\eta > 0$. Further, we define the function $r(t) = \sqrt{F(\chi_x(t)) - F^\star}$. By differentiating $r(t)$, we observe that:

$$\frac{dr(t)}{dt} = \frac{\dot{F}(\chi_x(t))}{2\sqrt{F(\chi_x(t)) - F^\star}}$$
$$= -\frac{\|\dot{\chi}_x(t)\|^2}{2\sqrt{F(\chi_x(t)) - F^\star}}$$
$$\leq -\sqrt{\overline{\mu}/2}\|\dot{\chi}_x(t)\|,$$

- In the second line, we plug in the definition of the subgradient curve.

- In the third line, we use the KL inequality and the fact that $\dot{\chi}_x(t) \in -\partial F(\chi_x(t))$.

Taking the integral, we write:

$$r(T) - r(0) = \int_0^T \frac{dr(t)}{dt} dt$$

$$\leq -\sqrt{\overline{\mu}/2} \int_0^T \|\dot{\chi}_x(t)\| dt \tag{10}$$

$$\leq -\sqrt{\overline{\mu}/2} \operatorname{dist}(\chi_x(T), \chi_x(0)), \tag{11}$$

where:

- (10) uses the bound on $r(t)$

- (11) uses the fact that any curve connecting two points has a length greater than their Euclidean distance.

Now, we will show that $\lim_{T\to\infty} r(T) = 0$. By (11), we get

$$r(T) - r(0) = \int_0^T \frac{dr(t)}{dt} dt$$

$$= -\int_0^T \frac{\|\dot{\chi}_x(t)\|^2}{2\sqrt{F(\chi_x(t)) - F^\star}} dt$$

$$\leq -\frac{\overline{\mu}}{2} \int_0^T \sqrt{F(\chi_x(t)) - F^\star} dt$$

$$\leq -\frac{\overline{\mu}}{2} Tr(T),$$

where the first inequality follows from the KL condition, and second inequality uses the fact that $F(\chi_x(t))$ is non-increasing in $t$. Hence, we get a bound on $r(T)$,

$$0 \leq r(T) \leq \frac{2r(0)}{2 + \overline{\mu}T},$$

By taking the limit of $T \to \infty$, we get $\lim_{T\to\infty} r(T) \to 0$. Now we can focus on the term $r(0) = \sqrt{F(x) - F^\star}$. Proceeding, by (11) we write,

$$\sqrt{F(x) - F^\star} \geq \sqrt{\overline{\mu}/2} \operatorname{dist}(x, \mathcal{X}^\star), \tag{12}$$

From (12) and the KL condition, we see that:

$$\operatorname{dist}(0, \partial F(x)) \geq \overline{\mu} \operatorname{dist}(x, \mathcal{X}^\star). \tag{13}$$

Now, let define $x^+ = \operatorname{prox}_{\frac{1}{\ell} g}\left(x - \frac{1}{\ell}\nabla f(x)\right)$. By the optimality of $x^+$, we have $-\nabla f(x^+) - \ell(x^+ - x) \in N_\mathcal{X}(x^+)$. As such,

$$\nabla f(x^+) - \nabla f(x) - \ell(x^+ - x) \in \partial I_\mathcal{X}(x^+).$$

Letting the particular subgradient of $I_\mathcal{X}(x^+)$ that achieves this be $\xi$, we write,

$$\operatorname{dist}(0, \partial F(x^+)) = \|0 - \xi\|$$

$$= \|\nabla f(x^+) - \nabla f(x) - \ell(x^+ - x)\|$$

$$\leq \ell\|x^+ - x\| + \|\nabla f(x^+) - \nabla f(x)\|$$

$$\leq 2\ell\|x^+ - x\|, \tag{14}$$

where we used the triangle inequality and the $\ell$-Lispchitz continuity of $\nabla f$. Finally, we can derive the proximal-EB condition by

$$\text{dist}(x, \mathcal{X}^\star) \leq \|x - x^+\| + \text{dist}(x^+, \mathcal{X}^\star)$$
$$\leq \|x - x^+\| + \frac{1}{\mu} \text{dist}(0, \partial F(x^+))$$
$$\leq \|x - x^+\| + \frac{2\ell}{\mu} \|x^+ - x\|$$
$$= \left(1 + \frac{2\ell}{\mu}\right) \|x - x^+\|.$$

We note that:

- the first inequality follows by the triangle inequality;
- the second inequality uses the (13);
- the third inequality uses (14).

Re-iterating:

$$\min_{s_x \in \partial(f + I_\mathcal{X})(x)} \|s_x\|^2 \geq 2\overline{\mu} \left(f(x) - f(x^\star)\right), \quad \forall x \in \mathcal{X} \quad \implies \quad \text{dist}(x, \mathcal{X}^\star) \leq \left(1 + \frac{2\ell}{\overline{\mu}}\right) \|x - x^+\|.$$

**pEB→pPL**   Before proceeding, we define the forward-backward envelope, $F_{\frac{1}{\ell}}$, of $F$ (Stella et al., 2017, Definition 2.1), as:

$$F_{\frac{1}{\ell}} := \min_{y \in \mathbb{R}^d} \left\{ f(x) + \langle \nabla f(x), y - x \rangle + \frac{\ell}{2} \|y - x\|^2 + I_\mathcal{X}(y) - I_\mathcal{X}(x) \right\}.$$

Additionally, we observe that: $\mathcal{D}_\mathcal{X}(x, \ell) = 2\ell \left( F(x) - F_{\frac{1}{\ell}}(x) \right)$. Moving forward we note that by assumption, it holds that:

$$\text{dist}(x, \mathcal{X}^\star) \leq c \left\| x - \Pi_\mathcal{X} \left( x - \frac{1}{\ell} \nabla f(x) \right) \right\|.$$

From the latter we write,

$$F(x) - F^\star = F(x) - F_{\frac{1}{\ell}}(x) + F_{\frac{1}{\ell}}(x) - F^\star$$
$$\leq F(x) - F_{\frac{1}{\ell}}(x) + 2\ell \|x^\star - x\|^2$$
$$\leq F(x) - F_{\frac{1}{\ell}}(x) + 2\ell c^2 \left\| x - \Pi_\mathcal{X} \left( x - \frac{1}{\ell} \nabla f(x) \right) \right\|^2$$
$$\leq \left(1 + 4c^2\right) \left( F(x) - F_{\frac{1}{\ell}}(x) \right)$$
$$= \left(1 + 4c^2\right) \frac{1}{2\ell} \mathcal{D}_\mathcal{X}(x, \ell).$$

Where the second inequality follows from the pEB condition and the two last steps follow from (Stella et al., 2017, Proposition 2.2(i)): $\frac{\ell}{2} \left\| x - \Pi_\mathcal{X} \left( x - \frac{1}{\ell} \nabla f(x) \right) \right\|^2 \leq F(x) - F_{\frac{1}{\ell}}(x)$ and $F(x) - F_{\frac{1}{\ell}}(x) = \frac{1}{2\ell} \mathcal{D}_\mathcal{X}(x, \ell)$. In short,

$$\frac{\ell}{1 + 4c^2} \left(F(x) - F^\star\right) \leq \frac{1}{2} \mathcal{D}_\mathcal{X}(x, \ell).$$

Concluding, we see that KL with modulus $\overline{\mu}$ translates to pEB with constant $c = \left(1 + \frac{2\ell}{\overline{\mu}}\right)$. In turn, pEB with constant $c$ translates to pPL with modulus $\frac{\ell}{1+4c^2}$. This means that KL with $\overline{\mu}$ is equivalent to pPL with modulus $\mu$,

$$\mu := \frac{\ell}{1 + 4\left(1 + \frac{2\ell}{\overline{\mu}}\right)^2} \geq \frac{\ell \overline{\mu}^2}{9\overline{\mu}^2 + 32\ell^2}.$$

## C.2. pPL from HSC

**Proposition C.2.** *Let $f : \mathcal{X} \to \mathbb{R}$ be an $\ell$-smooth function that is $(\mu_c, \mu_H)$-hidden strongly convex with $\mu_H > 0$. Then, $f$ satisfies the proximal Polyak-Łojasiewciz condition with a modulus $\mu$,*

$$\mu := \frac{\ell}{1 + 4\left(1 + \frac{2\ell}{2\mu_c^2 \mu_H}\right)^2}$$

*Proof.* By (Fatkhullin et al., 2023, Proposition 2(ii)), $(\mu_c, \mu_H)$-hidden strong convexity implies,

$$\min_{s_x \in \partial(f + I_{\mathcal{X}})(x)} \|s_x\|^2 \geq 2\mu_c^2 \mu_H \left(f(x) - f(x^\star)\right).$$

While, (Karimi et al., 2016, Appendix G) and our precise quantification of the constants,

$$\mu := \frac{\ell}{1 + 4\left(1 + \frac{2\ell}{2\mu_c^2 \mu_H}\right)^2}.$$

$\square$

## C.3. QG directly from HSC, KL, pPL

**Proposition C.3** (QG from HSC). *Let $f : \mathcal{X} \to \mathbb{R}$ be an $\ell$-smooth function that is $(\mu_c, \mu_H)$-hidden strongly convex. Then, for all $x \in \mathcal{X}$,*

$$f(x) - f(x^\star) \geq \frac{\mu_{\mathrm{QG}}}{2} \|x - x^\star\|^2,$$

*with $\mu_{\mathrm{QG}} := \mu_c^2 \mu_H$ where $x^\star$ is the unique minimizer of $f$ over $\mathcal{X}$.*

*Proof.* We start by leveraging the strong convexity property of the function $H$. By the definition of strong convexity with modulus $\mu_H$, for any $u, v \in \mathcal{U}$:

$$H(u) - H(u^\star) - \langle \nabla_u H(u^\star), u - u^\star \rangle \geq \frac{\mu_H}{2} \|u - u^\star\|^2. \tag{15}$$

When $u^\star$ is a minimizer of $H$, the gradient at $u^\star$ satisfies:

$$\langle \nabla_u H(u), u - u^\star \rangle \geq 0$$

Leveraging the latter, (15) simplifies into:

$$H(u) - H(u^\star) \geq \frac{\mu_H}{2} \|u - u^\star\|^2. \tag{16}$$

Because $c^{-1}$ is Lipschitz continuous with modulus $1/\mu_c$, we obtain:

$$\|x - x^\star\| = \left\|c^{-1}(u) - c^{-1}(u^\star)\right\| \leq \frac{1}{\mu_c} \|u - u^\star\|.$$

Substituting this bound into equation (16), we obtain:

$$H(u) - H(u^\star) \geq \frac{\mu_H}{2} \|u - u^\star\|^2$$
$$= \frac{\mu_c^2 \mu_H}{2} \|x - x^\star\|^2.$$

Recognizing that $H(u) = f(x)$ and $H(u^\star) = f(x^\star)$, we can rewrite the above inequality as:

$$f(x) - f(x^\star) \geq \frac{\mu_c^2 \mu_H}{2} \|x - x^\star\|^2.$$

This concludes the proof. $\square$

**Proposition C.4** (QG from KL). *Let an $\ell$-smooth function $f : \mathcal{X} \to \mathbb{R}$ satisfy the Kurdyka-Łojasiewicz (KL) condition with modulus $\mu > 0$, i.e.:*

$$\mu \left( f(x) - f^\star \right) \leq \frac{1}{2} \inf_{s_x \in \partial F(x)} \|s_x\|^2, \quad \forall x \in \mathcal{X}.$$

*Then, it satisfies the quadratic growth (QG) condition with a modulus $\frac{\mu}{2}$, i.e.,*

$$\frac{\mu}{4} \left\| x - x_p^\star \right\|^2 \leq f(x) - f^\star, \quad \forall x \in \mathcal{X},$$

*where $x_p^\star \in \arg\min_{x^\star \in \mathcal{X}^\star} \|x - x^\star\|^2$ and in turn $\mathcal{X}^\star := \arg\min_{x \in \mathcal{X}} f(x)$.*

*Proof.* Before proceeding, we note that the PL condition in (Liao et al., 2024) is what we call the KL condition in this manuscript. Let us define $F(\cdot) := f(\cdot) + I_\mathcal{X}(\cdot)$ where $I_\mathcal{X}$ is the indicator function of the compact convex set $\mathcal{X}$. We note that, $I_\mathcal{X}$ is convex and $f$ is $\ell$-smooth and as such $\ell$-weakly convex. Hence, $F(\cdot) = f(\cdot) + I_\mathcal{X}(\cdot)$ is also $\ell$-weakly convex. By (Liao et al., 2024, Proof of Theorem 3.1 (PL→EB))

$$\left\| x - x_p^\star \right\| \leq \frac{1}{\mu} \inf_{s \in \partial F(x)} \|s_x\|,$$

where $x_p^\star \in \arg\min_{x^\star \in \mathcal{X}^\star} \|x - x^\star\|^2$ and in turn $\mathcal{X}^\star := \arg\min_{x \in \mathcal{X}} f(x)$. Then, by (Liao et al., 2024, Proof of Theorem 3.1 (EB→QG)), we know that,

$$\frac{1}{4/\mu} \left\| x - x_p^\star \right\|^2 \leq f(x) - f^*.$$

$\square$

**Proposition C.5** (QG from pPL; (Mulvaney-Kemp et al., 2022, Theorem 2)). *Let an $\ell$-smooth function $f : \mathcal{X} \to \mathbb{R}$ satisfy the proximal Polyak-Łojasiewicz (PL) condition with modulus $\mu > 0$. Then, it satisfies the quadratic growth (QG) condition with the same modulus $\mu$,*

$$\frac{\mu}{4} \left\| x - x_p^\star \right\|^2 \leq f(x) - f^\star, \quad \forall x \in \mathcal{X},$$

*where $x_p^\star \in \arg\min_{x^\star \in \mathcal{X}^\star} \|x - x^\star\|^2$ and in turn $\mathcal{X}^\star := \arg\min_{x \in \mathcal{X}} f(x)$.*

### C.4. Warm-up: Stochastic Projected Gradient Descent on a proximal-PL Function

Now consider stochastic projected gradient on a nonconvex pPL $\ell$-smooth function $f$. The analysis closely follows (J Reddi et al., 2016).

Assume $\mathbb{E}\hat{g}_x(x) = g_x(x)$, $\|\hat{g}_x(x) - g_x(x)\| \leq \delta$ and $\mathbb{E} \|\hat{g}_x(x)\|^2 \leq \sigma^2$. Further, define points $x_{t+1}, \overline{x}_{t+1}$ to be:

$$x_{t+1} = \Pi_\mathcal{X} \left( x_t - \eta\hat{g}_x(x_t) \right);$$
$$\overline{x}_{t+1} = \Pi_\mathcal{X} \left( x_t - \eta\nabla_x f(x_t) \right).$$

**Theorem C.6.** *Assume an $\ell$-smooth function $f : \mathcal{X} \to \mathbb{R}$ with an inexact stochastic gradient oracle $\hat{g}_x$ such that $\mathbb{E}\hat{g}_x(x) = g_x(x)$, $\|\hat{g}_x(x) - g_x\| \leq \delta$ and $\mathbb{E} \|\hat{g}_x\|^2 \leq \sigma^2$. Then,*

- *running stochastic gradient descent with a $\delta_x$-inexact gradient for $T > 0$ iterations, yields the following inequality:*

$$\frac{1}{T} \sum_{t=0}^{T-1} \frac{1}{2} \mathcal{D}_\mathcal{X}(x_{t+1}, \ell) \leq \frac{3\ell \left( \mathbb{E}f(x_0) - f(x^\star) \right)}{T} + 3\delta_x^2 + 3\sigma_x^2.$$

- *Further, if $f$ is pPL with modulus $\mu$, the followinng inequality holds true:*

$$\mathbb{E}f(x_{T+1}) - f(x^\star) \leq \exp\left( -\frac{\mu}{3\ell}T \right) (f(x_0) - f^\star) + \frac{3\ell\delta_x^2}{\mu} + \frac{3\ell\sigma_x^2}{\mu}.$$

*Proof.* $\qquad\qquad\qquad\qquad\qquad\qquad\qquad\qquad\qquad\qquad\qquad\qquad\qquad\qquad\qquad\qquad\qquad\qquad\qquad\qquad$ $\square$

- We let $\overline{x}_{t+1} := \Pi_{\mathcal{X}}\left(x_t - \eta \nabla f(x_t)\right)$. Invoking $\ell$-smoothness of $f$ for $x_t, \overline{x}_{t+1}$ and requiring that $\eta \le \frac{1}{\ell}$,

$$
\begin{aligned}
f(\overline{x}_{t+1}) &\le f(x_t) + \langle \nabla f(x_t), \overline{x}_{t+1} - x_t \rangle + \frac{1}{2\eta}\|\overline{x}_{t+1} - x_t\|^2 \\
&= f(x_t) - \left(\langle \nabla f(x_t), x_t - \overline{x}_{t+1} \rangle - \frac{1}{2\eta}\|\overline{x}_{t+1} - x_t\|^2\right) \\
&= f(x_t) - \frac{\eta}{2}\mathcal{D}_{\mathcal{X}}(x_t, 1/\eta) \tag{17}
\end{aligned}
$$

- Invoking Lemma B.8 with $x = x_t$, $y = \overline{x}_{t+1}$, $z = x_t$, $v = \nabla f(x_t)$

$$
f(\overline{x}_{t+1}) \le f(x_t) + \left(\frac{\ell}{2} - \frac{1}{\eta}\right)\|\overline{x}_{t+1} - x_t\|^2. \tag{18}
$$

- Again, invoking Lemma B.8 but with $x = x_t$, $y = x_{t+1}$, $z = \overline{x}_{t+1}$, $v = \hat{g}(x_t)$,

$$
\begin{aligned}
f(x_{t+1}) \le{}& f(\overline{x}_{t+1}) + \langle \nabla f(x_t) - \hat{g}_x(x_t), x_{t+1} - \overline{x}_{t+1} \rangle \\
&+ \left(\frac{\ell}{2} - \frac{1}{2\eta}\right)\|x_{t+1} - x_t\|^2 + \left(\frac{\ell}{2} + \frac{1}{2\eta}\right)\|\overline{x}_{t+1} - x_t\|^2 - \frac{1}{2\eta}\|x_{t+1} - \overline{x}_{t+1}\|^2. \tag{19}
\end{aligned}
$$

Adding $1/3\times$(17) and $2/3\times$(18)

$$
f(\overline{x}_{t+1}) \le f(x_t) - \frac{\eta}{6}\mathcal{D}_{\mathcal{X}}(x_t, 1/\eta) + \left(\frac{\ell}{3} - \frac{2}{3\eta}\right)\|\overline{x}_{t+1} - x_t\|^2
$$

Adding (19),

$$
\begin{aligned}
f(x_{t+1}) \le{}& f(x_t) - \frac{\eta}{6}\mathcal{D}_{\mathcal{X}}(x_t, 1/\eta) + \left(\frac{\ell}{3} - \frac{2}{3\eta}\right)\|\overline{x}_{t+1} - x_t\|^2 \\
&+ \langle \nabla f(x_t) - \hat{g}_x(x_t), x_{t+1} - \overline{x}_{t+1} \rangle \\
&+ \left(\frac{\ell}{2} - \frac{1}{2\eta}\right)\|x_{t+1} - x_t\|^2 + \left(\frac{\ell}{2} + \frac{1}{2\eta}\right)\|\overline{x}_{t+1} - x_t\|^2 - \frac{1}{2\eta}\|x_{t+1} - \overline{x}_{t+1}\|^2 \\
\le{}& f(x_t) - \frac{\eta}{6}\mathcal{D}_{\mathcal{X}}(x_t, 1/\eta) + \left(\frac{5\ell}{6} - \frac{1}{6\eta}\right)\|\overline{x}_{t+1} - x_t\|^2 \\
&+ \frac{\eta}{2}\|\nabla f(x_t) - \hat{g}_x(x_t)\|^2 \tag{20} \\
&+ \left(\frac{\ell}{2} - \frac{1}{2\eta}\right)\|x_{t+1} - x_t\|^2 \\
\le{}& f(x_t) - \frac{\eta}{6}\mathcal{D}_{\mathcal{X}}(x_t, 1/\eta) + \frac{\eta}{2}\|\nabla f(x_t) - \hat{g}_x(x_t)\|^2 \tag{21} \\
\le{}& f(x_t) - \frac{\eta}{6}\mathcal{D}_{\mathcal{X}}(x_t, 1/\eta) + \eta\|\nabla f(x_t) - g_x(x_t)\|^2 + \eta\|g_x(x_t) - \hat{g}_x(x_t)\|^2 \tag{22}
\end{aligned}
$$

We note that we have picked $\eta \le \frac{1}{5\ell}$.

- (20) follows from the application of Young's inequality $\langle a, b \rangle \le \frac{\rho}{2}\|a\|^2 + \frac{1}{2\rho}\|b\|^2$ with $\rho = \eta$;

- (21) follows by dropping the non-positive terms; non-positivity follows from the inequalities:
  1. $0 \ge 5\ell - \frac{1}{\eta}$
  2. $\ell \le 5\ell \le \frac{1}{\eta}$ then $\ell - \frac{1}{\eta} \le 0$

- (22) follows from $\|a + b\|^2 \leq 2\|a\|^2 + 2\|b\|^2$.

Taking expectations:

$$\mathbb{E}f(x_{t+1}) \leq \mathbb{E}f(x_t) - \frac{\eta}{6}\mathcal{D}_{\mathcal{X}}(x_t, 1/\eta) + \eta\delta_x^2 + \eta\sigma_x^2 \tag{23}$$

where the last inequality follows from the choice of the step-size. Rearranging (23) and summing over $T$, we get:

$$\frac{1}{T}\sum_{t=0}^{T-1}\frac{1}{2}\mathcal{D}_{\mathcal{X}}(x_t, 1/\eta) \leq \frac{3}{\eta T}\sum_{t=0}^{T-1}\left(\mathbb{E}f(x_t) - \mathbb{E}f(x_{t+1}) + 3\delta_x^2 + 3\sigma_x^2\right)$$

$$\leq \frac{3\left(\mathbb{E}f(x_0) - f(x^\star)\right)}{\eta T} + 3\delta_x^2 + 3\sigma_x^2.$$

When the pPL condition holds and $\eta = \frac{1}{5\ell}$, we get from (23),

$$\mathbb{E}f(x_{t+1}) - f^\star \leq \left(1 - \frac{\mu}{3\ell}\right)\left(\mathbb{E}f(x_t) - f^\star\right) + \delta_x^2 + \sigma_x^2,$$

and as such,

$$\mathbb{E}f(x_T) - f^\star \leq \left(1 - \frac{\mu}{3\ell}\right)^T\left(f(x_0) - f^\star\right) + \left(\delta_x^2 + \sigma_x^2\right)\sum_{t=1}^{T}\left(1 - \frac{\mu}{3\ell}\right)^t$$

$$= \left(1 - \frac{\mu}{3\ell}\right)^T\left(f(x_0) - f^\star\right) + \left(\delta_x^2 + \sigma_x^2\right)\frac{1 - \left(1 - \frac{\mu}{3\ell}\right)^{T+1}}{1 - \left(1 - \frac{\mu}{3\ell}\right)}$$

$$\leq \left(1 - \frac{\mu}{3\ell}\right)^T\left(f(x_0) - f^\star\right) + \frac{3\ell\delta_x^2}{\mu} + \frac{3\ell\sigma_x^2}{\mu}$$

$$\leq \exp\left(-\frac{\mu}{3\ell}T\right)\left(f(x_0) - f^\star\right) + \frac{3\ell\delta_x^2}{\mu} + \frac{3\ell\sigma_x^2}{\mu}.$$

# D. Constrained Min-Max Optimization Under the pPL Condition

## D.1. Key Lemmata

**Theorem D.1** (NC-pPL and cont. of maximizers). *Let function $f : \mathcal{X} \times \mathcal{Y} \to \mathbb{R}$ with $f(x, \cdot)$ satisfying the proximal-PL condition with parameter $\mu$ for all $x \in \mathcal{X}$. Then, consider points $x_1, x_2 \in \mathcal{X}$ and $y^\star(x_1), y^\star(x_2) \in \mathcal{Y}$ with $y^\star(x_1) := \arg\max_{y \in \mathcal{Y}} f(x_1, y)$ and $y^\star(x_2) := \arg\max_{y \in \mathcal{Y}} f(x_2, y)$, it holds true that:*

$$\|y^\star(x_1) - y^\star(x_2)\| \leq L_\star \|x_1 - x_2\|,$$

*where $L_\star := \frac{\ell}{\mu}$.*

*Remark* 2. One might compare the last statement to the Robust Berge Maximum Theorem (Papadimitriou et al., 2023, Th. 3.20) which concerns (non)convex–strongly-concave functions with coupled feasibility sets. Essentially, the former statement illustrates that hidden-strong-concavity is in some aspect a stronger assumption than strong-concavity; in hidden-strong-concavity the feasibility sets are only "hiddenly" coupled. This allows us to decouple the constraint sets and view the problem as a constrained nonconvex-pPL problem. Then, it is quite intuitive that Lispchitz continuity of the maximizers holds in light of the analogous result (Nouiehed et al., 2019, Lemma A.3) which concerns unconstrained min-max optimization over nonconvex-PL functions. Ultimately, this decoupling principle is also the reason why Kalogiannis et al. (2024) can compute the gradient of the maximum function by only invoking Danskin's Theorem and not the more elaborate Envelope Theorem (Afriat, 1971).

*Proof.* Since we have defined the pPL and QG growth conditions for a minimization problem, let us assume $g(x, y) = -f(x, y)$. Consequentially, $\arg\min_{y \in \mathcal{Y}} g(x, y) = \arg\max_{y \in \mathcal{Y}} f(x, y) \ni y^\star(x)$. Finally, we define:

$$\mathcal{D}_{\mathcal{Y}}(y, \alpha; x) := -2\alpha \min_{z \in \mathcal{Y}}\left\{\langle\nabla_y g(x, y), z - y\rangle + \frac{\alpha}{2}\|z - y\|^2\right\}.$$

By the proximal-PL condition, it holds that,

$$\frac{1}{2}\mathcal{D}_{\mathcal{Y}}\left(y^{\star}(x_1), 1/\eta; x_2\right) \geq \mu\left(g\left(x_2, y^{\star}(x_1)\right) - g\left(x_2, y^{\star}(x_2)\right)\right)$$

By the QG condition:

$$g\left(x_2, y^{\star}(x_1)\right) - g\left(x_2, y^{\star}(x_2)\right) \geq \frac{\mu_{\mathrm{QG}}}{2}\left\|y^{\star}(x_1) - y_p^{\star}(x_2)\right\|^2$$

Where, $y_p^{\star}(x_2) := \arg\min_{y' \in \mathcal{Y}^{\star} x}\|y^{\star}(x_1) - y'\|$ and $\mathcal{Y}^{\star}(x) := \arg\min_{y \in \mathcal{Y}} g(x, y)$. Finally, by Lemma B.11,

$$\mathcal{D}_{\mathcal{Y}}\left(y^{\star}(x_1), 1/\eta; x_2\right) \leq \ell^2\left\|x_1 - x_2\right\|^2.$$

Putting these pieces together,

$$\frac{\mu_{\mathrm{QG}}}{2}\left\|y^{\star}(x_1) - y_p^{\star}(x_2)\right\|^2 \leq \frac{\ell^2}{2\mu}\left\|x_1 - x_2\right\|^2.$$

Further, we know that pPL with modulus $\mu$ implies QG with the same modulus (Proposition C.5). This concludes the proof.

$\square$

**Corollary D.2.** *Let function $f : \mathcal{X} \times \mathcal{Y} \to \mathbb{R}$ with $f(x, \cdot)$ satisfying the proximal-PL condition with modulus $\mu$ for all $x \in \mathcal{X}$. Then, let $\Phi(x) := \max_{y \in \mathcal{Y}} f(x, y)$. For any two points $x_1, x_2 \in \mathcal{X}$ it holds true that:*

$$\left\|\nabla_x\Phi(x_1) - \nabla_x\Phi(x_2)\right\| \leq \ell_{\Phi}\left\|x_1 - x_2\right\|,$$

*where $\ell_{\Phi} := \ell(1 + L_{\star}) = \ell + \frac{\ell^2}{\mu}$.*

*Proof.* We write,

$$\begin{aligned}
\left\|\nabla_x\Phi(x_1) - \nabla_x\Phi(x_2)\right\| &= \left\|\nabla_x f(x_1, y^{\star}(x_1)) - \nabla_x f(x_2, y^{\star}(x_2))\right\| \\
&\leq \ell\left\|(x_1, y^{\star}(x_1)) - (x_2, y^{\star}(x_2))\right\| \\
&\leq \ell\left\|x_1 - x_2\right\| + \ell L_{\star}\left\|x_1 - x_2\right\|.
\end{aligned}$$

The first equation holds due to Danskin's lemma, and the first inequality is due to $\ell$-Lipschitz continuity of the gradient. The second inequality is due to the triangle inequality and the $L_{\star}$-Lipschitz continuity of the maximizers. $\square$

**Lemma D.3.** *Let $f : \mathcal{X} \times \mathcal{Y} \to \mathbb{R}$ be an L-Lipschitz continuous and $\ell$-smooth function that satisfies the two-sided pPL condition for both $f(\cdot, y)$ and $f(x, \cdot)$, then:*

$$\min_{x \in \mathcal{X}} \max_{y \in \mathcal{Y}} f(x, y) = \max_{y \in \mathcal{Y}} \min_{x \in \mathcal{X}} f(x, y) =: \Phi^{\star}.$$

*Proof.* We can invoke (Yang et al., 2020, Lemma 2.1) which holds for two-sided pPL functions with minor modifications.

$\square$

**Lemma D.4.** *Let the function $f : \mathcal{X} \times \mathcal{Y} \to \mathbb{R}$ satisfy the pPL condition with moduli $\mu_1, \mu_2 > 0$ respectively for $f(\cdot, y), \forall y \in \mathcal{Y}$ and $f(x, \cdot), \forall x \in \mathcal{X}$. Then, the function $\Phi : \mathcal{X} \to \mathbb{R}$ with $\Phi(x) = \max_{y' \in \mathcal{Y}} f(x, y')$ satisfies the pPL condition with modulus $\mu_1$.*

*Proof.* For the purposes of this proof, we will enhance the notation of $\mathcal{D}_{\mathcal{X}}$ as follows:

$$\begin{aligned}
\mathcal{D}_{\mathcal{X}}^f(x, \alpha; y) &:= -2\alpha \min_{z \in \mathcal{X}}\left\{\langle\nabla f(x, y), z - x\rangle + \frac{\alpha}{2}\|z - x\|^2\right\}; \\
\mathcal{D}_{\mathcal{X}}^{\Phi}(x, \alpha) &:= -2\alpha \min_{z \in \mathcal{X}}\left\{\langle\nabla\Phi(x), z - x\rangle + \frac{\alpha}{2}\|z - x\|^2\right\}.
\end{aligned}$$

$$\mathcal{D}_{\mathcal{X}}^{\Phi}(x, \ell_{\Phi}) = \mathcal{D}_{\mathcal{X}}^{f}(x, \ell_{\Phi}; y^{\star}(x))$$

$$\geq 2\mu_1 \left( f(x, y^{\star}(x)) - \min_{x' \in \mathcal{X}} f(x', y^{\star}(x)) \right).$$

Further,

$$f(x', y^{\star}(x)) \leq \max_{y \in \mathcal{Y}} f(x', y),$$

and minimizing on both sides yields,

$$\min_{x' \in \mathcal{X}} f(x', y^{\star}(x)) \leq \min_{x' \in \mathcal{X}} \max_{y \in \mathcal{Y}} f(x', y) = \Phi^{\star}.$$

Hence,

$$\mathcal{D}_{\mathcal{X}}^{\Phi}(x, \ell_{\Phi}) \geq 2\mu_1 \left( \Phi(x) - \Phi^{\star} \right).$$

$\square$

### D.2. Stationarity Proxies and the Gradient Oracle

**Definition 11.** We define $\mathcal{D}_{\mathcal{Y}}$ and $\mathcal{D}_{\mathcal{X}}^{\Phi}$ to be the following quantities:

$$\mathcal{D}_{\mathcal{Y}}(y, \alpha; x) = -2\alpha \min_{z \in \mathcal{Y}} \left\{ \langle -\nabla f(x, y), z - y \rangle + \frac{\alpha}{2} \|z - y\|^2 \right\},$$

and correspondigly,

$$\mathcal{D}_{\mathcal{X}}^{\Phi}(x, \alpha) = -2\alpha \min_{z \in \mathcal{X}} \left\{ \langle \nabla \Phi(x), z - x \rangle + \frac{\alpha}{2} \|z - x\|^2 \right\}.$$

**Definition 12.** We define the deterministic and stochastic gradient mapping at point $x_t$ and $y_t$ to be:

- $\mathcal{G}_{x, \tau_x}(x) := \frac{1}{\tau_x} \left( x - \Pi_{\mathcal{X}} \left( x - \tau_x v \right) \right)$ and $\mathcal{G}_{x, \tau_x}^{t} = \mathcal{G}_{x, \tau_x}(x_t)$;

- $\hat{\mathcal{G}}_{x, \tau_x}^{t}(x_t) := \frac{1}{\tau_x} \left( x - \Pi_{\mathcal{X}} \left( x - \tau_x \hat{g}_x(x_t, y_t) \right) \right)$ and $\hat{\mathcal{G}}_{x, \tau_x}^{t} = \hat{\mathcal{G}}_{x, \tau_x}(x_t)$,

and respectively:

- $\mathcal{G}_{y, \tau_y}(y) := \frac{1}{\tau_y} \left( \Pi_{\mathcal{Y}} \left( y + \tau_y v \right) - y \right)$ and $\mathcal{G}_{y, \tau_y}^{t} = \mathcal{G}_{y, \tau_y}^{t}(y)$;

- $\hat{\mathcal{G}}_{y, \tau_y}(y) := \frac{1}{\tau_y} \left( \Pi_{\mathcal{Y}} \left( y + \tau_y \hat{v} \right) - y \right)$ and $\hat{\mathcal{G}}_{y, \tau_y}^{t} = \hat{\mathcal{G}}_{y, \tau_y}^{t}(y)$.

**Assumption 3** (Unbiased Inexact Gradient Estimators and Bounded Second Moments). For all iterations $t$, the gradient estimators $\hat{g}_x(x_t, y_t)$ and $\hat{g}_y(x_t, y_t)$ satisfy

$$\mathbb{E}\left[\hat{g}_x(x_t, y_t)\right] = g(x_t, y_t),$$

$$\mathbb{E}\left[\hat{g}_y(x_t, y_t)\right] = g(x_t, y_t),$$

and

$$\mathbb{E}\left[\|\hat{g}_x(x_t, y_t)\|^2\right] \leq \sigma_x^2,$$

$$\mathbb{E}\left[\|\hat{g}_y(x_t, y_t)\|^2\right] \leq \sigma_y^2.$$

In turn, $\|g_x(x_t, y_t) - \nabla_x f(x_t, y_t)\| \leq \delta_x$, $\|g_y(x_t, y_t) - \nabla_y f(x_t, y_t)\| \leq \delta_y$.

*Remark* 3 (Bound on Second Moment instead of Variance). At first, it might appear a slightly stronger assumption to place a bound on the second moment of the gradient estimator. Yet, in most relevant works the variance of the relevant estimators is bounded only after bounding the second moment (Daskalakis et al., 2020; Zhang et al., 2020; 2021). As such, this assumption is reasonable and rather standard to satisfy for the aforementioned applications.

### D.3. Convergence of Nested Gradient Iterations

We can formulate the nested gradient iterations algorithm using the following template,

$$
\begin{cases}
y_{t+1} \leftarrow \texttt{ARGMAX}(f(x_t, \cdot), \epsilon_y); \\
x_{t+1} \leftarrow \Pi_{\mathcal{X}}\left(x_t - \eta \nabla f(x_t, y_{t+1})\right)
\end{cases}
\tag{24}
$$

where, $\texttt{ARGMAX}(h, \epsilon)$ returns an $\epsilon$-approximate maximize of function $h$. As a function, it can be implemented efficiently by projected gradient ascent.

Finally, the outer loop of the process implements projected gradient descent with a stochastic and inexact gradient feedback on $\Phi(\cdot)$.

**Theorem D.5** (NC-pPL). *Let $f : \mathcal{X} \times \mathcal{Y} \to \mathbb{R}$ be an $\ell$-smooth function satisfying the pPL condition with a modulus $\mu$ for $f(x, \cdot)$. Then, after $T$ iterations of* (24) *it holds true that:*

$$
\frac{1}{T} \sum_{t=0}^{T-1} \left\| \hat{\mathcal{G}}_{x,\tau_x}^{\Phi,t} \right\|^2 \leq \frac{6 L \mathrm{D}_{\mathcal{X}}}{\tau_x T} + 6\delta_x^2 + 6\sigma_x^2.
$$

*Proof.* By Theorem C.6 and a tuning of step-size $\tau_x \leq \frac{1}{5\ell_\Phi}$, we get that,

$$
\frac{1}{T} \sum_{t=0}^{T-1} \frac{1}{2} \mathcal{D}_{\mathcal{X}}^{\Phi}(x_{t+1}, 1/\tau_x) \leq \frac{3\ell \left(\mathbb{E}\Phi(x_0) - \Phi(x^\star)\right)}{\tau_x T} + 3\delta_x^2 + 3\sigma_x^2.
$$

In this context, the inexactness of the gradient oracle $\delta_x$ is:

$$
\delta_x = \max_{0 \leq t \leq T-1} \mathbb{E}\left\| \nabla\Phi(x_t) - \nabla f(x_t, y_t) \right\| = \max_{0 \leq t \leq T-1} \ell \mathbb{E}\left\| y_t - y^\star(x_t) \right\|,
$$

while by the quadratic growth condition of $f(x, \cdot)$ we know that,

$$
\frac{\mu_y}{2} \left\| y_t - y^\star(x_t) \right\| \leq \mathbb{E}\Phi(x_t) - \mathbb{E}f(x_t, y_t)
$$

$$
\leq \epsilon_y
$$

where $\epsilon_y$ is the accuracy of the inner loop which we will defer tuning. We, invoke Lemma B.7 to see that the sum of $\mathcal{D}_{\mathcal{X}}^{\Phi}(x_t, 1/\tau_x)$ upper bounds the sum of $\left\| \hat{\mathcal{G}}_{x,\tau_x}^{\Phi,t} \right\|^2$:

$$
\frac{1}{T} \sum_{t=0}^{T-1} \left\| \hat{\mathcal{G}}_{x,\tau_x}^{\Phi,t} \right\|^2 \leq \frac{6 \left(\mathbb{E}\Phi(x_0) - \Phi(x^\star)\right)}{\tau_x T} + 6\delta_x^2 + 6\sigma_x^2.
$$

Now, we see that $\Phi(x_0) - \Phi(x^\star)$ is bounded by $L\mathrm{D}_{\mathcal{X}}$ due to the $L$-Lipschitz continuity of $\Phi$ and the bounded diameter of $\mathcal{X}$. Finally, we can tune $\tau_x$ and $\epsilon_y$.

- $\tau_x = \frac{1}{5\ell_\Phi}$

- $T = \frac{18 L \mathrm{D}_{\mathcal{X}}}{\epsilon^2}$

- $M_x = \frac{\sigma_x^2}{18\epsilon^2}$

- $\epsilon_y = \frac{\epsilon}{\sqrt{18}}$.

$\square$

**Theorem D.6** (pPL-pPL). *Let $f : \mathcal{X} \times \mathcal{Y} \to \mathbb{R}$ be an $L$-Lipschitz, $\ell$-smooth function and $\mathcal{X}, \mathcal{Y}$ be two compact convex sets with Euclidean diameters $\mathrm{D}_{\mathcal{X}}, \mathrm{D}_{\mathcal{Y}}$ respectively. Further, assume that $f(\cdot, y)$ satisfies the pPL condition with modulus $\mu_x$ for all $y \in \mathcal{Y}$ while $f(x, \cdot)$ satisfies the pPL condition with a modulus $\mu_y$ for any $x \in \mathcal{X}$. Additionally, let $\hat{g}_x$ be an*

*inexact stochastic gradient oracle such that* $\mathbb{E}\hat{g}_x(x) = g_x(x)$, $\mathbb{E}\|\hat{g}_x(x) - g_x(x)\| \leq \sigma_x^2$, *and* $\|g_x(x)\| \leq \delta_x$. *Then, after* $T$ *iterations of* (24) *with a tuning of step-sizes* $\tau_x = \frac{1}{5\ell_\Phi}$:

$$\mathbb{E}\Phi(x_{T+1}) - \Phi(x^\star) \leq \exp\left(-\frac{\mu}{3\ell_\Phi}T\right)LD_\mathcal{X} + \frac{3\ell_\Phi\delta_x^2}{\mu_x} + \frac{3\ell_\Phi\sigma_x^2}{\mu_x}.$$

*Proof.* By Theorem C.6 with $\tau_x = \frac{1}{5\ell_\Phi}$,

$$\mathbb{E}\Phi(x_{T+1}) - \Phi(x^\star) \leq \exp\left(-\frac{\mu}{3\ell_\Phi}T\right)(\Phi(x_0) - \Phi^\star) + \frac{3\ell_\Phi\delta_x^2}{\mu_x} + \frac{3\ell_\Phi\sigma_x^2}{\mu_x}.$$

First, we repeat the fact that $LD_\mathcal{X} \geq \Phi(x_0) - \Phi^\star$ due to Lipschitz continuity. Then, we note that $\delta_x$ has to be tuned as $\sqrt{\frac{\epsilon\mu_x}{9\ell_\Phi}}$ and the batch-size needs to be $M_x = \left\lceil\frac{9\ell_\Phi\sigma_x^2}{\epsilon\mu_x}\right\rceil$ where $\epsilon > 0$ is the desired accuracy. Finally, $T = \left\lceil\frac{3\ell_\Phi}{\mu}\log\left(\frac{3LD_\mathcal{X}}{\epsilon}\right)\right\rceil$. $\qquad\square$

### D.4. Convergence of Stochastic Alternating Gradient Descent-Ascent

In what follows, we will analyze the convergence of projected alternating gradient descent-ascent for nonconvex-pPL and two-sided pPL functions. The convergence proofs closely follow those of (Yang et al., 2022) and (Yang et al., 2020) respectively after carefully modifying the arguments to make them work for the constrained setting. Convergence is proven by showing that an appropriate Lyapunov function diminshes along the trajectories of the algorithm's iterates (Bof et al., 2018).

In both scenarios we face, proving that the corresponding Lyapunov function diminishes is proven by first:

- lower bounding the descent on the maximum function $\Phi(\cdot)$ for every update on $x$;
- lower bounding the ascent on $f(x, \cdot)$ for every update on $y$;
- *upper* bounding the descent on $f(\cdot, y)$ for every update on $x$.

As a reminder, the iteration scheme of alternating gradient descent-ascent is the following,

$$x_{t+1} = \Pi_\mathcal{X}\left(x_t - \tau_x\hat{g}_x(x_t, y_t)\right);$$
$$y_{t+1} = \Pi_\mathcal{Y}\left(y_t + \tau_y\hat{g}_y(x_{t+1}, y_t)\right).$$

#### D.4.1. NC-PPL

**Theorem D.7.** *Let* $f : \mathcal{X} \times \mathcal{Y} \to \mathbb{R}$ *be an L-Lipschitz, $\ell$-smooth function and $\mathcal{X}, \mathcal{Y}$ be two compact convex sets with Euclidean diameters* $D_\mathcal{X}, D_\mathcal{Y}$ *respectively. Further, assume that $f(x, \cdot)$ satisfies the pPL condition with modulus $\mu$. Additionally, let $(\hat{g}_x, \hat{g}_y)$ be an inexact stochastic gradient oracle satisfying assumption 3. Then, after $T$ iterations of* (`Alt-GDA`) *with a tuning of step-sizes* $\tau_x = \frac{1}{500\ell\kappa^2}$ *and* $\tau_y = \frac{1}{5\ell}$, *it holds true that*

$$\frac{1}{T}\sum_{t=1}^{T-1}\left(\mathbb{E}\left\|\mathcal{G}_{x,\tau_x}^t\right\|^2 + \kappa^2\mathbb{E}\mathcal{D}_\mathcal{Y}(y_t, \ell; x_t)\right) \leq \frac{\kappa^2\ell L(D_\mathcal{X} + D_\mathcal{Y})}{T} + \frac{c_2\sigma_x^2}{M_x} + c_2\delta_x^2 + \frac{c_3\kappa^2\sigma_y^2}{M_y} + c_3\kappa^2\delta_y^2,$$

*where,* $c_1, c_2, c_3 \in O(1)$.

*Proof.*

**Descent bound on $\Phi$.**

$$\mathcal{G}_{x,\tau_x}^{\Phi,t} := \frac{1}{\tau_x}\left(x_t - \Pi_x\left(x_t - \tau_x\nabla_x\Phi(x_t)\right)\right)$$

Due to $\ell$-smoothness and the fact that $\frac{1}{\tau_x} \geq 5\ell_\Phi$ we can use Equation (23) to get:

$$\mathbb{E}\Phi(x_{t+1}) \leq \mathbb{E}\Phi(x_t) - \frac{\tau_x}{6}\mathbb{E}\mathcal{D}_\mathcal{X}^\Phi(x_t, 1/\tau_x) + \tau_x\mathbb{E}\|\nabla_x\Phi(x_t) - \nabla_x f(x_t, y_t)\|^2 + \tau_x\mathbb{E}\|\nabla_x f(x_t, y_t) - \hat{g}_x(x_t, y_t)\|^2$$

$$\leq \mathbb{E}\Phi(x_t) - \frac{\tau_x}{6}\mathbb{E}\left\|\mathcal{G}_{x,\tau_x}^{\Phi,t}\right\|^2 + \tau_x\mathbb{E}\left\|\nabla_x\Phi(x_t) - \nabla_x f(x_t, y_t)\right\|^2 + \tau_x\mathbb{E}\left\|\nabla_x f(x_t, y_t) - \hat{g}_x(x_t, y_t)\right\|^2$$

$$\leq \mathbb{E}\Phi(x_t) - \frac{\tau_x}{6}\mathbb{E}\left\|\mathcal{G}_{x,\tau_x}^{\Phi,t}\right\|^2 + \tau_x\mathbb{E}\left\|\nabla_x\Phi(x_t) - \nabla_x f(x_t, y_t)\right\|^2 + 2\tau_x\sigma_x^2 + 2\tau_x\delta_x^2$$

**Ascent bound on $f(x_t, \cdot)$**   For a choice of $\tau_y \leq \frac{1}{5\ell} \leq \frac{1}{\ell}$,

$$\mathbb{E}f(x_{t+1}, y_{t+1}) \geq \mathbb{E}f(x_{t+1}, y_t) + \frac{\tau_y}{6}\mathbb{E}\mathcal{D}_{\mathcal{Y}}(y_t, 1/\tau_y; x_{t+1}) - \tau_y\delta_y^2 - \tau_y\sigma_y^2.$$

**Upper Bound on the descent:** $f(x_t, y_t) - f(x_{t+1}, y_t)$**.**

$$
\begin{aligned}
f(x_{t+1}, y_t) &\geq f(x_t, y_t) + \langle \nabla_x f(x_t, y_t), x_{t+1} - x_t \rangle - \frac{\ell}{2}\left\|x_{t+1} - x_t\right\|^2 \\
&= f(x_t, y_t) - \tau_x\left\langle g_x(x_t, y_t), \hat{\mathcal{G}}_{x,\tau_x}^t \right\rangle - \tau_x\left\langle \nabla_x f(x_t, y_t) - g_x(x_t, y_t), \hat{\mathcal{G}}_{x,\tau_x}^t \right\rangle \\
&\quad - \frac{\ell\tau_x^2}{2}\left\|\hat{\mathcal{G}}_{x,\tau_x}^t\right\|^2 \\
&= f(x_t, y_t) - \tau_x\left\langle \hat{g}_x(x_t, y_t), \hat{\mathcal{G}}_{x,\tau_x}^t \right\rangle - \tau_x\left\langle g_x(x_t, y_t) - \hat{g}_x(x_t, y_t), \hat{\mathcal{G}}_{x,\tau_x}^t \right\rangle \\
&\quad - \tau_x\left\langle \nabla_x f(x_t, y_t) - g_x(x_t, y_t), \hat{\mathcal{G}}_{x,\tau_x}^t \right\rangle \\
&\quad - \frac{\ell\tau_x^2}{2}\left\|\hat{\mathcal{G}}_{x,\tau_x}^t\right\|^2 \\
&= f(x_t, y_t) - \tau_x\left\langle \hat{g}_x(x_t, y_t), \hat{\mathcal{G}}_{x,\tau_x}^t \right\rangle \\
&\quad - \tau_x\left\langle g_x(x_t, y_t) - \hat{g}_x(x_t, y_t), \mathcal{G}_{x,\tau_x}^t \right\rangle - \tau_x\left\langle g_x(x_t, y_t) - \hat{g}_x(x_t, y_t), \hat{\mathcal{G}}_{x,\tau_x}^t - \mathcal{G}_{x,\tau_x}^t \right\rangle \\
&\quad - \tau_x\left\langle \nabla_x f(x_t, y_t) - g_x(x_t, y_t), \hat{\mathcal{G}}_{x,\tau_x}^t \right\rangle \\
&\quad - \frac{\ell\tau_x^2}{2}\left\|\hat{\mathcal{G}}_{x,\tau_x}^t\right\|^2 \\
&\geq f(x_t, y_t) - \frac{\tau_x}{2}\left\|\hat{g}_x(x_t, y_t)\right\|^2 - \frac{\tau_x}{2}\left\|\hat{\mathcal{G}}_{x,\tau_x}^t\right\|^2 \\
&\quad - \tau_x\left\langle g_x(x_t, y_t) - \hat{g}_x(x_t, y_t), \mathcal{G}_{x,\tau_x}^t \right\rangle - \tau_x\left\langle g_x(x_t, y_t) - \hat{g}_x(x_t, y_t), \hat{\mathcal{G}}_{x,\tau_x}^t - \mathcal{G}_{x,\tau_x}^t \right\rangle \\
&\quad - \frac{\tau_x}{2}\left\|\nabla_x f(x_t, y_t) - g_x(x_t, y_t)\right\|^2 - \frac{\tau_x}{2}\left\|\hat{\mathcal{G}}_{x,\tau_x}^t\right\| \\
&\quad - \frac{\ell\tau_x^2}{2}\left\|\hat{\mathcal{G}}_{x,\tau_x}^t\right\|^2 \\
&\geq f(x_t, y_t) - \left(\tau_x + \frac{\ell\tau_x^2}{2}\right)\left\|\hat{\mathcal{G}}_{x,\tau_x}^t\right\|^2 - \frac{\tau_x}{2}\left\|\hat{g}_x(x_t, y_t)\right\|^2 \\
&\quad - \tau_x\left\langle g_x(x_t, y_t) - \hat{g}_x(x_t, y_t), \mathcal{G}_{x,\tau_x}^t \right\rangle - \tau_x\left\langle g_x(x_t, y_t) - \hat{g}_x(x_t, y_t), \hat{\mathcal{G}}_{x,\tau_x}^t - \mathcal{G}_{x,\tau_x}^t \right\rangle \\
&\quad - \frac{\tau_x\delta_x^2}{2} \\
&\geq f(x_t, y_t) - \left(\tau_x + \frac{\ell\tau_x^2}{2}\right)\left\|\hat{\mathcal{G}}_{x,\tau_x}^t\right\|^2 - \frac{\tau_x}{2}\left\|\hat{g}_x(x_t, y_t)\right\|^2 \\
&\quad - \tau_x\left\langle g_x(x_t, y_t) - \hat{g}_x(x_t, y_t), \mathcal{G}_{x,\tau_x}^t \right\rangle - \tau_x\left\|g_x(x_t, y_t) - \hat{g}_x(x_t, y_t)\right\|^2 - \frac{\tau_x\delta_x^2}{2}.
\end{aligned}
$$

$$(25)$$

$$(26)$$

$$(27)$$

• the initial equations are mere additions-subtractions of terms and plugging-in of definitions;

- (25) follows from Young's inequality;
- (26) follows from gathering terms and using the bound on the gradient inexactness error;
- (27) follows from the Cauchy-Schwarz inequality and the non-expansiveness of the projection.

Taking expectations again:

$$
\begin{aligned}
\mathbb{E}f(x_{t+1}, y_t) &\geq \mathbb{E}f(x_t, y_t) - \left(\tau_x + \frac{\ell\tau_x^2}{2}\right)\mathbb{E}\left\|\hat{\mathcal{G}}_{x,\tau_x}^t\right\|^2 - \frac{\tau_x\sigma_x^2}{2} \\
&\quad - \tau_x\mathbb{E}\left\langle g_x(x_t, y_t) - \hat{g}_x(x_t, y_t), \mathcal{G}_{x,\tau_x}^t\right\rangle - \tau_x\sigma_x^2 - \frac{\tau_x\delta_x^2}{2} \\
&= \mathbb{E}f(x_t, y_t) - \left(\tau_x + \frac{\ell\tau_x^2}{2}\right)\mathbb{E}\left\|\hat{\mathcal{G}}_{x,\tau_x}^t\right\|^2 - \frac{3\tau_x\sigma_x^2}{2} - \frac{\tau_x\delta_x^2}{2} \\
&\geq \mathbb{E}f(x_t, y_t) - \frac{3\tau_x}{4}\mathbb{E}\left\|\hat{\mathcal{G}}_{x,\tau_x}^t\right\|^2 - \frac{3\tau_x\sigma_x^2}{2} - \frac{\tau_x\delta_x^2}{2} \\
&\geq \mathbb{E}f(x_t, y_t) - \frac{3\tau_x}{2}\mathbb{E}\left\|\mathcal{G}_{x,\tau_x}^t\right\|^2 - \frac{9\tau_x\sigma_x^2}{2}\sigma_x^2 - \frac{7\tau_x\delta_x^2}{2}
\end{aligned} \tag{28}
$$

Where, we used that $\tau_x \leq \frac{1}{2\ell}$, which means that $\frac{1}{2}\left(\tau_x + \ell\tau_x^2\right) \leq \frac{3\tau_x}{4}$.

**Bounding AGDA iterate difference** $\mathbb{E}f(x_{t+1}, y_{t+1}) - \mathbb{E}f(x_t, y_t)$**.**

$$
\begin{aligned}
\mathbb{E}f(x_{t+1}, y_{t+1}) - \mathbb{E}f(x_t, y_t) &\geq -\frac{3\tau_x}{2}\mathbb{E}\left\|\mathcal{G}_{x,\tau_x}^t\right\| - \frac{9\tau_x\sigma_x^2}{2} - \frac{7\tau_x}{2}\delta_x^2 \\
&\quad + \frac{\tau_y}{6}\mathbb{E}\mathcal{D}_{\mathcal{Y}}(y_t, 1/\tau_y; x_{t+1}) - \tau_y\delta_y^2 - \tau_y\sigma_y^2
\end{aligned}
$$

**The Lyapunov function.** We will define the Lyapunov function,

$$
V(x_t, y_t) := \Phi(x_t) + \alpha\Big(\Phi(x_t) - f(x_t, y_t)\Big) = (1+\alpha)\Phi(x_t) - \alpha f(x_t, y_t),
$$

with $\alpha > 0$ to be tuned at the end. Also, we will denote $V_t := V(x_t, y_t)$.

$$
\mathbb{E}V_t - \mathbb{E}V_{t+1} = (1+\alpha)(\mathbb{E}\Phi(x_t) - \mathbb{E}\Phi(x_{t+1})) + \alpha(\mathbb{E}f(x_{t+1}, y_{t+1}) - f(x_t, y_t))
$$

$$
\begin{aligned}
\mathbb{E}V_t - \mathbb{E}V_{t+1} &\geq (1+\alpha)\left(\frac{\tau_x}{6}\mathbb{E}\left\|\mathcal{G}_{x,\tau_x}^{\Phi,t}\right\|^2 - \tau_x\mathbb{E}\left\|\nabla_x\Phi(x_t) - \nabla_x f(x_t, y_t)\right\|^2 - 2\tau_x\sigma_x^2 - 2\tau_x\delta_x^2\right) \\
&\quad + \alpha\left(-\frac{3\tau_x}{2}\mathbb{E}\left\|\mathcal{G}_{x,\tau_x}^t\right\|^2 - \frac{9\tau_x\sigma_x^2}{2} - \frac{7\tau_x}{2}\delta_x^2\right) \\
&\quad + \alpha\left(\frac{\tau_y}{6}\mathbb{E}\mathcal{D}_{\mathcal{Y}}(y_t, 1/\tau_y; x_{t+1}) - \tau_y\delta_y^2 - \tau_y\sigma_y^2\right) \\
&\geq \frac{(1+\alpha)\tau_x}{6}\mathbb{E}\left\|\mathcal{G}_{x,\tau_x}^{\Phi,t}\right\|^2 - (1+\alpha)\tau_x\mathbb{E}\left\|\nabla_x\Phi(x_t) - \nabla_x f(x_t, y_t)\right\|^2 \\
&\quad - \frac{3\tau_x\alpha}{2}\mathbb{E}\left\|\mathcal{G}_{x,\tau_x}^t\right\|^2 + \frac{\alpha\tau_y}{6}\mathbb{E}\mathcal{D}_{\mathcal{Y}}(y_t, 1/\tau_y; x_{t+1}) \\
&\quad + \left(-2\tau_x(1+\alpha) - \frac{9\tau_x}{2}\alpha\right)\sigma_x^2 + (-\alpha\tau_y)\sigma_y^2 \\
&\quad + \left(-2\tau_x(1+\alpha) - \frac{7\tau_x}{2}\alpha\right)\delta_x^2 + (-\alpha\tau_y)\delta_y^2
\end{aligned}
$$

$$\geq \frac{(1+\alpha)\tau_x}{6} \mathbb{E} \left\| \mathcal{G}_{x,\tau_x}^{\Phi,t} \right\|^2 - (1+\alpha)\tau_x \mathbb{E} \left\| \nabla_x \Phi(x_t) - \nabla_x f(x_t, y_t) \right\|^2$$

$$- \frac{3\tau_x \alpha}{2} \mathbb{E} \left\| \mathcal{G}_{x,\tau_x}^t \right\|^2 + \frac{\alpha\tau_y}{6} \mathbb{E} \mathcal{D}_\mathcal{Y}(y_t, 1/\tau_y; x_t) - \frac{3\alpha\tau_y \ell^2}{6} \mathbb{E} \left\| x_t - x_{t+1} \right\|^2$$

$$+ \left( -2\tau_x(1+\alpha) - \frac{9\tau_x}{2}\alpha \right) \sigma_x^2 + (-\alpha\tau_y) \sigma_y^2$$

$$+ \left( -2\tau_x(1+\alpha) - \frac{7\tau_x}{2}\alpha \right) \delta_x^2 + (-\alpha\tau_y) \delta_y^2 \tag{29}$$

$$\geq \frac{(1+\alpha)\tau_x}{6} \mathbb{E} \left\| \mathcal{G}_{x,\tau_x}^{\Phi,t} \right\|^2 - (1+\alpha)\tau_x \mathbb{E} \left\| \nabla_x \Phi(x_t) - \nabla_x f(x_t, y_t) \right\|^2$$

$$- \frac{3\tau_x \alpha}{2} \mathbb{E} \left\| \mathcal{G}_{x,\tau_x}^t \right\|^2 + \frac{\alpha\tau_y}{6} \mathbb{E} \mathcal{D}_\mathcal{Y}(y_t, \ell; x_t)$$

$$- \frac{\alpha\tau_x^2 \tau_y \ell^2}{2} \mathbb{E} \left\| \mathcal{G}_{x,\tau_x}^t \right\|^2 - 2\alpha\tau_x^2 \tau_y \ell^2 \sigma_x^2 - 2\alpha\tau_x^2 \tau_y \ell^2 \delta_x^2$$

$$+ \left( -2\tau_x(1+\alpha) - \frac{9\tau_x}{2}\alpha \right) \sigma_x^2 + (-\alpha\tau_y) \sigma_y^2$$

$$+ \left( -2\tau_x(1+\alpha) - \frac{7\tau_x}{2}\alpha \right) \delta_x^2 + (-\alpha\tau_y) \delta_y^2 \tag{30}$$

$$= \frac{(1+\alpha)\tau_x}{6} \mathbb{E} \left\| \mathcal{G}_{x,\tau_x}^{\Phi,t} \right\|^2 - (1+\alpha)\tau_x \mathbb{E} \left\| \nabla_x \Phi(x_t) - \nabla_x f(x_t, y_t) \right\|^2$$

$$+ \left( -\frac{3\tau_x \alpha}{2} - \alpha\tau_x^2 \tau_y \ell^2 \right) \mathbb{E} \left\| \mathcal{G}_{x,\tau_x}^t \right\|^2 + \frac{\alpha\tau_y}{6} \mathbb{E} \mathcal{D}_\mathcal{Y}(y_t, \ell; x_t)$$

$$+ \left( -2\tau_x(1+\alpha) - \frac{9\tau_x}{2}\alpha - 2\alpha\tau_x^2 \tau_y \ell^2 \right) \sigma_x^2 + (-\alpha\tau_y) \sigma_y^2$$

$$+ \left( -2\tau_x(1+\alpha) - \frac{7\tau_x}{2}\alpha - 2\alpha\tau_x^2 \tau_y \ell^2 \right) \delta_x^2 + (-\alpha\tau_y) \delta_y^2$$

$$\geq \left( \frac{(1+\alpha)\tau_x}{6} - 3\tau_x \alpha - 2\alpha\tau_x^2 \tau_y \ell^2 \right) \mathbb{E} \left\| \mathcal{G}_{x,\tau_x}^{\Phi,t} \right\|^2 - (1+\alpha)\tau_x \mathbb{E} \left\| \nabla_x \Phi(x_t) - \nabla_x f(x_t, y_t) \right\|^2$$

$$+ \left( \frac{\alpha\tau_y}{6} - 3\tau_x \alpha\kappa^2 - 2\alpha\tau_x^2 \tau_y \ell^2 \kappa^2 \right) \mathbb{E} \mathcal{D}_\mathcal{Y}(y_t, \ell; x_t)$$

$$+ \left( -2\tau_x(1+\alpha) - \frac{9\tau_x}{2}\alpha - 2\alpha\tau_x^2 \tau_y \ell^2 \right) \sigma_x^2 + (-\alpha\tau_y) \sigma_y^2$$

$$+ \left( -2\tau_x(1+\alpha) - \frac{7\tau_x}{2}\alpha - 2\alpha\tau_x^2 \tau_y \ell^2 \right) \delta_x^2 + (-\alpha\tau_y) \delta_y^2$$

$$\geq \left( \frac{(1+\alpha)\tau_x}{6} - 3\tau_x \alpha - 2\alpha\tau_x^2 \tau_y \ell^2 \right) \mathbb{E} \left\| \mathcal{G}_{x,\tau_x}^{\Phi,t} \right\|^2 - (1+\alpha)\tau_x \kappa^2 \mathcal{D}_\mathcal{Y}(y_t, \ell; x_t)$$

$$+ \left( \frac{\alpha\tau_y}{6} - 3\tau_x \alpha\kappa^2 - 2\alpha\tau_x^2 \tau_y \ell^2 \kappa^2 \right) \mathbb{E} \mathcal{D}_\mathcal{Y}(y_t, \ell; x_t)$$

$$+ \left( -2\tau_x(1+\alpha) - \frac{9\tau_x}{2}\alpha - 2\alpha\tau_x^2 \tau_y \ell^2 \right) \sigma_x^2 + (-\alpha\tau_y) \sigma_y^2$$

$$+ \left( -2\tau_x(1+\alpha) - \frac{7\tau_x}{2}\alpha - 2\alpha\tau_x^2 \tau_y \ell^2 \right) \delta_x^2 + (-\alpha\tau_y) \delta_y^2 \tag{31}$$

$$= \left( \frac{(1+\alpha)\tau_x}{6} - 3\tau_x \alpha - 2\alpha\tau_x^2 \tau_y \ell^2 \right) \mathbb{E} \left\| \mathcal{G}_{x,\tau_x}^{\Phi,t} \right\|^2$$

$$+ \left( \frac{\alpha\tau_y}{6} - 3\tau_x \alpha\kappa^2 - 2\alpha\tau_x^2 \tau_y \ell^2 \kappa^2 - (1+\alpha)\tau_x \kappa^2 \right) \mathbb{E} \mathcal{D}_\mathcal{Y}(y_t, \ell; x_t)$$

$$+ \left( -2\tau_x(1+\alpha) - \frac{9\tau_x}{2}\alpha - 2\alpha\tau_x^2 \tau_y \ell^2 \right) \sigma_x^2 + (-\alpha\tau_y) \sigma_y^2$$

$$+ \left( -2\tau_x(1+\alpha) - \frac{7\tau_x}{2}\alpha - 2\alpha\tau_x^2\tau_y\ell^2 \right) \delta_x^2 + (-\alpha\tau_y)\,\delta_y^2$$

- (29) uses Lemma B.12 and the fact that $|a-b| < c$ implies $a > b - c$, *i.e.*,

$$\mathcal{D}_{\mathcal{Y}}(y_{t+1}, \alpha; x_t) \geq \mathcal{D}_{\mathcal{Y}}(y_t, \alpha; x_t) - 3\ell^2\,\|x_t - x_{t+1}\|^2,$$

- (30) uses the definition of $\hat{\mathcal{G}}_{x,\tau_x}^t, \mathcal{G}_{x,\tau_x}^t$ and Lemma B.4 to replace the term: $\|x_t - x_{t+1}\|^2$, *i.e.*:

$$\begin{aligned}
\|x_t - x_{t+1}\|^2 &\leq 2\,\|x_t - \overline{x}_{t+1}\|^2 + 2\,\|x_{t+1} - \overline{x}_{t+1}\|^2 \\
&\leq 2\tau_x^2\,\left\|\mathcal{G}_{x,\tau_x}^t\right\|^2 + 2\tau_x^2\,\|\hat{g}_x(x_t,y_t) - \nabla_x f(x_t,y_t)\|^2 \\
&\leq 2\tau_x^2\,\left\|\mathcal{G}_{x,\tau_x}^t\right\|^2 + 4\tau_x^2\sigma_x^2 + 4\tau_x^2\delta_x^2.
\end{aligned}$$

- By the fact that $\|c-d\|^2 = \|c\|^2 + \|d\|^2 - 2\langle c,d\rangle$ and Young's inequality, we can write $\|a\|^2 = \|(a-b)+b\|^2 \leq (1+1/\rho)\|a\|^2 + (1+\rho)\|b\|^2$. Then we plug in $a \leftarrow \mathcal{G}_{x,\tau_x}^t, b \leftarrow \mathcal{G}_{x,\tau_x}^{\Phi,t}$,

$$\begin{aligned}
\left\|\mathcal{G}_{x,\tau_x}^t\right\|^2 &\leq (1+\rho)\,\left\|\mathcal{G}_{x,\tau_x}^{\Phi,t}\right\|^2 + \left(1+\frac{1}{\rho}\right)\left\|\frac{1}{\tau_x}\left(\Pi_{\mathcal{X}}\left(x - \tau_x\nabla f(x_t,y_t)\right) - \Pi_{\mathcal{X}}\left(x - \tau_x\nabla\Phi(x_t)\right)\right)\right\|^2 \\
&\leq (1+\rho)\,\left\|\mathcal{G}_{x,\tau_x}^{\Phi,t}\right\|^2 + \left(1+\frac{1}{\rho}\right)\|\nabla f(x_t,y_t) - \nabla\Phi(x_t)\|^2 \\
&\leq (1+\rho)\,\left\|\mathcal{G}_{x,\tau_x}^{\Phi,t}\right\|^2 + \left(1+\frac{1}{\rho}\right)\ell^2\,\|y^\star(x_t) - y_t\|^2 \\
&\leq (1+\rho)\,\left\|\mathcal{G}_{x,\tau_x}^{\Phi,t}\right\|^2 + \left(1+\frac{1}{\rho}\right)\frac{\ell^2}{\mu_{\mathrm{QG}}}\left(\Phi(x_t) - f(x_t,y_t)\right) \\
&\leq (1+\rho)\,\left\|\mathcal{G}_{x,\tau_x}^{\Phi,t}\right\|^2 + \left(1+\frac{1}{\rho}\right)\kappa^2\mathcal{D}_{\mathcal{Y}}(y_t,\ell;x_t)
\end{aligned}$$

Where the third to last inequality follows from Danskin's lemma, the penultimate is due to the quadratic growth property, and the last is due to the proximal-PL property. We have set $\kappa := \frac{\ell}{\sqrt{\mu_{\mathrm{QG}}\mu}}$.

- (31) follows from the pPL property.

$$\begin{aligned}
\left\|\mathcal{G}_{x,\tau_x}^t\right\|^2 &\leq 2\,\left\|\mathcal{G}_{x,\tau_x}^{\Phi,t}\right\|^2 + 2\,\left\|\mathcal{G}_{x,\tau_x}^t - \mathcal{G}_{x,\tau_x}^{\Phi,t}\right\|^2 \\
&\leq 2\,\left\|\mathcal{G}_{x,\tau_x}^{\Phi,t}\right\|^2 + 2\,\|\nabla f(x_t,y_t) - \nabla\Phi(x_t)\|^2 \\
&\leq 2\,\left\|\mathcal{G}_{x,\tau_x}^{\Phi,t}\right\|^2 + 2\ell\,\|y_t - y^\star(x_t)\|^2 \\
&\leq 2\,\left\|\mathcal{G}_{x,\tau_x}^{\Phi,t}\right\|^2 + 2\ell\kappa^2\mathcal{D}_{\mathcal{Y}}(y_t,\ell;x_t)
\end{aligned}$$

Re-iterating,

$$\begin{aligned}
\mathbb{E}V_t - \mathbb{E}V_{t+1} \geq\ & \left(\frac{(1+\alpha)\tau_x}{6} - 3\tau_x\alpha - 2\alpha\tau_x^2\tau_y\ell^2\right)\mathbb{E}\left\|\mathcal{G}_{x,\tau_x}^{\Phi,t}\right\|^2 \\
&+ \left(\frac{\alpha\tau_y}{6} - 3\tau_x\alpha\kappa^2 - 2\alpha\tau_x^2\tau_y\ell^2\kappa^2 - (1+\alpha)\tau_x\kappa^2\right)\mathbb{E}\mathcal{D}_{\mathcal{Y}}(y_t,\ell;x_t) \\
&+ \left(-2\tau_x(1+\alpha) - \frac{9\tau_x}{2}\alpha - 2\alpha\tau_y\ell^2\right)\sigma_x^2 + \left(-\alpha\tau_x^2\tau_y\right)\sigma_y^2 \\
&+ \left(-2\tau_x(1+\alpha) - \frac{7\tau_x}{2}\alpha - 2\alpha\tau_x^2\tau_y\ell^2\right)\delta_x^2 + (-\alpha\tau_y)\,\delta_y^2
\end{aligned}$$

What is left to do is to ensure that the coefficients of $\left\|\mathcal{G}_{x,\tau_x}^{\Phi,t}\right\|^2, \mathcal{D}_{\mathcal{Y}}(y_t,\ell;x_t)$ in the last display are positive. We will show that this is indeed the case for a correct tuning of $\tau_x, \tau_y$ and $\alpha$.

- **Coefficient of $\left\|\mathcal{G}_{x,\tau_x}^{\Phi,t}\right\|^2$.** We assume *a priori* that, $\tau_x, \tau_y \leq \frac{1}{\ell}$. Hence we can write:

$$\frac{(1+\alpha)\tau_x}{6} - 3\tau_x\alpha - 2\alpha\tau_x^2\tau_y\ell^2 \geq \frac{1+\alpha - 18\alpha - 12\alpha}{6}\tau_x \geq \frac{1 - 29\alpha}{6}\tau_x \geq \frac{\tau_x}{180}$$

Where we require that $\alpha \leq 1/30 < 1/29$.

- **Coefficient of $\mathcal{D}_{\mathcal{Y}}$.** We require that $\tau_x \leq \frac{\tau_y}{c\kappa^2}$ for some $c > 1$.

$$\begin{aligned}
\frac{\alpha\tau_y}{6} - 3\tau_x\alpha\kappa^2 - 2\alpha\tau_x^2\tau_y\ell^2\kappa^2 - (1+\alpha)\tau_x\kappa^2 &\geq \frac{\alpha - 18\alpha/c - 12\alpha/c^2 - 6(1+\alpha)/c}{6}\tau_y \\
&= \frac{(c^2 - 24c - 12)\alpha - 6c}{6c^2}\tau_y \\
&\geq \frac{\tau_y}{10}
\end{aligned}$$

Where in the inequality we assume $c = 100, \alpha = 1/30$.

- **Coefficients of $\sigma_x^2, \sigma_y^2, \delta_x, \delta_y$.**

  – For $\sigma_x^2, \delta_x^2$

$$\begin{aligned}
2\tau_x(1+\alpha) + \frac{7\tau_x}{2}\alpha + 2\alpha\tau_x^2\tau_y\ell^2 &\leq 2\tau_x(1+\alpha) + \frac{9\tau_x}{2}\alpha + 2\alpha\tau_x^2\tau_y\ell^2 \\
&= \tau_x\frac{60 + 2 + 135}{30} + \frac{200\tau_x^3\kappa^2\ell^2}{30} && \left(\tau_x = \frac{\tau_y}{100\kappa^2}\right) \\
&= \tau_x\frac{60 + 2 + 135}{30} + \frac{200\tau_x^2\ell}{30 \cdot 500} && \left(\tau_x = \frac{1}{500\kappa^2\ell}\right) \\
&\leq \tau_x\frac{60 + 2 + 135}{30} + \frac{200\tau_x}{30 \cdot 500} \leq 8\tau_x && (\tau_x \leq 1/\ell \implies \ell \leq 1/\tau_x).
\end{aligned}$$

  – For $\sigma_y^2, \delta_y^2$:

$$\alpha\tau_y \leq \frac{100\kappa^2}{30}\tau_x \leq 4\kappa^2\tau_x$$

$$\left\|\mathcal{G}_{x,\tau_x}^t\right\|^2 \leq 2\left\|\mathcal{G}_{x,\tau_x}^{\Phi,t}\right\|^2 + 2\kappa^2\mathcal{D}_{\mathcal{Y}}(y_t, \ell; x_t)$$

Summarizing:

$$\begin{aligned}
\frac{\tau_x}{360}\left\|\mathcal{G}_{x,\tau_x}^t\right\|^2 + 9\kappa^2\tau_x\mathbb{E}\mathcal{D}_{\mathcal{Y}}(y_t, \ell; x_t) &\leq \frac{\tau_x}{360}\left\|\mathcal{G}_{x,\tau_x}^t\right\|^2 + \left(\frac{\tau_y}{10} - \frac{2\kappa^2\tau_x}{360}\right)\mathbb{E}\mathcal{D}_{\mathcal{Y}}(y_t, \ell; x_t) \\
&\leq \mathbb{E}V_t - \mathbb{E}V_{t+1} + 8\tau_x\sigma_x^2 + 8\tau_x\delta_x^2 + 4\kappa^2\tau_x\sigma_y^2 + 4\kappa^2\tau_x\delta_y^2.
\end{aligned}$$

Summing over $t$ and dividing by $T$:

$$\frac{1}{T}\sum_{t=1}^{T-1}\left(\left\|\mathcal{G}_{x,\tau_x}^t\right\|^2 + \kappa^2\mathbb{E}\mathcal{D}_{\mathcal{Y}}(y_t, \ell; x_t)\right) \leq \frac{720L(\mathrm{D}_{\mathcal{X}} + \mathrm{D}_{\mathcal{Y}})}{\tau_x T} + 2880(\sigma_x^2 + \delta_x^2) + 1440\kappa^2(\sigma_y^2 + \delta_y^2).$$

$\square$

D.4.2. TWO-SIDED pPL MIN-MAX OPTIMIZATION

**Theorem D.8.** *Let $f : \mathcal{X} \times \mathcal{Y} \to \mathbb{R}$ be an $L$-Lipschitz, $\ell$-smooth function and $\mathcal{X}, \mathcal{Y}$ be two compact convex sets with Euclidean diameters $\mathrm{D}_\mathcal{X}, \mathrm{D}_\mathcal{Y}$ respectively. Further, assume that $f(\cdot, y)$ satisfies the pPL condition with modulus $\mu_x$ for all $y \in \mathcal{Y}$ while $f(x, \cdot)$ satisfies the pPL condition with a modulus $\mu_y$ for any $x \in \mathcal{X}$. Additionally, let $(\hat{g}_x, \hat{g}_y)$ be an inexact stochastic gradient oracle satisfying assumption 3. Then, after $T$ iterations of (`Alt-GDA`) with a tuning of step-sizes $\tau_x = \frac{\mu_y^2}{160\ell^3}$ and $\tau_y = \frac{1}{5\ell}$, it holds true that:*

$$\mathbb{E}\Phi(x_T) - \Phi^\star + 1/10\left(\mathbb{E}\Phi(x_T) - \mathbb{E}f(x_T, y_T)\right) \leq \exp\left(-\frac{\mu_x \mu_y^2}{160\ell^3}T\right)L(\mathrm{D}_\mathcal{X} + \mathrm{D}_\mathcal{Y}) + \frac{c_1\sigma_x^2}{\mu_x} + \frac{c_1\delta_x^2}{\mu_x} + \frac{c_2\ell^2\sigma_y^2}{\mu_x\mu_y^2} + \frac{c_2\ell^2\delta_y^2}{\mu_x\mu_y^2},$$

*where, $c_1, c_2 \in O(1)$.*

*Proof.* Our goal is to ultimately demonstrate that there exists a Lyapunov function, $V_t$ whose value decreases along any trajectory of the algorithm's iterates. Not only so, but its value contracts, *i.e.*, there exists $0 < \varpi < 1$ such that $V_{t+1} \leq \varpi V_t$, $\forall t$. To demonstrate this, we first need to lower bound the descent on $\Phi(\cdot)$, lower bound the ascent on $f(x_t, \cdot)$, and finally *upper bound* the descent on $f(\cdot, y_t)$.

**Descent Bound on $\Phi(\cdot)$.** By Lemma B.9 and Lemma D.4 we write,

$$\mathbb{E}\Phi(x_{t+1}) \leq \mathbb{E}\Phi(x_t) - \frac{\tau_x}{6}\mathbb{E}\mathcal{D}_\mathcal{X}^\Phi(x_t, 1/\tau_x) + \tau_x\mathbb{E}\left\|\nabla_x\Phi(x_t) - \nabla_x f(x_t, y_t)\right\|^2$$
$$+ 2\tau_x\sigma_x^2 + 2\tau_x\delta_x^2$$
$$\leq \mathbb{E}\Phi(x_t) - \frac{\tau_x\mu_x}{3}\left(\Phi(x_t) - \Phi^\star\right) + \tau_x\mathbb{E}\left\|\nabla_x\Phi(x_t) - \nabla_x f(x_t, y_t)\right\|^2$$
$$+ 2\tau_x\sigma_x^2 + 2\tau_x\delta_x^2$$

Hence,

$$\mathbb{E}\Phi(x_{t+1}) - \Phi^\star \leq \left(1 - \frac{\tau_x\mu_x}{3}\right)\left(\Phi(x_t) - \Phi^\star\right) + \tau_x\ell^2\left\|y^\star - y_t\right\|^2$$
$$+ 2\tau_x\sigma_x^2 + 2\tau_x\delta_x^2$$
$$\leq \left(1 - \frac{\tau_x\mu_x}{3}\right)\left(\Phi(x_t) - \Phi^\star\right) + \tau_x\ell^2\mu_{\mathrm{QG}}/2\left(\Phi(x_t) - f(x_t, y_t)\right)$$
$$+ 2\tau_x\sigma_x^2 + 2\tau_x\delta_x^2, \tag{32}$$

or, we can also write,

$$\mathbb{E}\Phi(x_{t+1}) - \mathbb{E}\Phi(x_t) \leq -\frac{\tau_x\mu_x}{3}\left(\Phi(x_t) - \Phi^\star\right) + \tau_x\ell^2\mu_{\mathrm{QG}}/2\left(\Phi(x_t) - f(x_t, y_t)\right)$$
$$+ 2\tau_x\sigma_x^2 + 2\tau_x\delta_x^2 \tag{33}$$

**Ascent Bound on $f(x_t, \cdot)$.** A simple application of Lemma B.9 yields,

$$\mathbb{E}f(x_{t+1}, y_{t+1}) \geq \mathbb{E}f(x_{t+1}, y_t) + \frac{\tau_y}{6}\mathbb{E}\mathcal{D}_\mathcal{Y}(y_t, 1/\tau_y; x_{t+1}) - \tau_y\delta_y^2 - \tau_y\sigma_y^2.$$

From which, we can also write,

$$\mathbb{E}\Phi(x_{t+1}) - \mathbb{E}f(x_{t+1}, y_{t+1})$$
$$\leq \mathbb{E}\Phi(x_{t+1}) - \mathbb{E}f(x_{t+1}, y_t) - \frac{\tau_y}{6}\mathbb{E}\mathcal{D}_\mathcal{Y}(y_t, 1/\tau_y; x_{t+1}) + \tau_y\delta_y^2 + \tau_y\sigma_y^2$$

$$\leq \left(1 - \frac{\mu_y \tau_y}{6}\right)(\mathbb{E}\Phi(x_{t+1}) - \mathbb{E}f(x_{t+1}, y_t)) + \tau_y \delta_y^2 + \tau_y \sigma_y^2$$

$$= \left(1 - \frac{\mu_y \tau_y}{6}\right)(\mathbb{E}\Phi(x_t) - \mathbb{E}f(x_t, y_t) + \mathbb{E}f(x_t, y_t) - \mathbb{E}f(x_{t+1}, y_t) + \mathbb{E}\Phi(x_{t+1}) - \mathbb{E}\Phi(x_t))$$
$$+ \tau_y \delta_y^2 + \tau_y \sigma_y^2 \tag{34}$$

**Upper bound on the descent on $f(\cdot, y_t)$**    From (28) we write,

$$\mathbb{E}f(x_{t+1}, y_t) \geq \mathbb{E}f(x_t, y_t) - \frac{3\tau_x}{2}\mathbb{E}\left\|\mathcal{G}_{x,\tau_x}^t\right\|^2 - \frac{9\tau_x}{2}\sigma_x^2 - \frac{7\tau_x}{2}\delta_x^2$$

$$\geq \mathbb{E}f(x_t, y_t) - \frac{3\tau_x}{2}\mathcal{D}_{\mathcal{X}}(x_t, 1/\tau_x; y_t) - \frac{9\tau_x}{2}\sigma_x^2 - \frac{7\tau_x}{2}\delta_x^2 \tag{35}$$

The second inequality follows Lemma B.7. Now, (34) by plugging-in (33) and (35) reads,

$$\mathbb{E}\Phi(x_{t+1}) - \mathbb{E}f(x_{t+1}, y_{t+1}) \leq \left(1 - \frac{\mu_y \tau_y}{6}\right)(\mathbb{E}\Phi(x_t) - \mathbb{E}f(x_t, y_t))$$
$$+ \left(1 - \frac{\mu_y \tau_y}{6}\right)(\mathbb{E}f(x_t, y_t) - \mathbb{E}f(x_{t+1}, y_t))$$
$$+ \left(1 - \frac{\mu_y \tau_y}{6}\right)(\mathbb{E}\Phi(x_{t+1}) - \mathbb{E}\Phi(x_t))$$
$$+ \tau_y \delta_y^2 + \tau_y \sigma_y^2$$
$$\leq \left(1 - \frac{\mu_y \tau_y}{6}\right)(\mathbb{E}\Phi(x_t) - \mathbb{E}f(x_t, y_t))$$
$$+ \left(1 - \frac{\mu_y \tau_y}{6}\right)\left(\frac{3\tau_x}{2}\mathcal{D}_{\mathcal{X}}(x_t, 1/\tau_x; y_t) + \frac{9\tau_x}{2}\sigma_x^2 + \frac{7\tau_x}{2}\delta_x^2\right)$$
$$+ \left(1 - \frac{\mu_y \tau_y}{6}\right)\left(-\frac{\tau_x}{6}\mathbb{E}\mathcal{D}_{\mathcal{X}}^{\Phi}(x_t, 1/\tau_x) + \tau_x \mathbb{E}\left\|\nabla_x \Phi(x_t) - \nabla_x f(x_t, y_t)\right\|^2 + 2\tau_x \sigma_x^2 + 2\tau_x \delta_x^2\right)$$
$$+ \tau_y \sigma_y^2 + \tau_y \delta_y^2$$
$$\leq \left(1 - \frac{\mu_y \tau_y}{6}\right)(\mathbb{E}\Phi(x_t) - \mathbb{E}f(x_t, y_t))$$
$$+ \left(1 - \frac{\mu_y \tau_y}{6}\right)\frac{3\tau_x}{2}\mathcal{D}_{\mathcal{X}}(x_t, 1/\tau_x; y_t)$$
$$+ \left(1 - \frac{\mu_y \tau_y}{6}\right)\left(-\frac{\tau_x}{6}\mathbb{E}\mathcal{D}_{\mathcal{X}}^{\Phi}(x_t, 1/\tau_x) + \tau_x \mathbb{E}\left\|\nabla_x \Phi(x_t) - \nabla_x f(x_t, y_t)\right\|^2\right)$$
$$+ 7\left(1 - \frac{\mu_y \tau_y}{6}\right)\tau_x \sigma_x^2 + 7\left(1 - \frac{\mu_y \tau_y}{6}\right)\tau_x \delta_x^2 + \tau_y \sigma_y^2 + \tau_y \delta_y^2 \tag{36}$$

**The Lyapunov Function.**    We will consider the following Lyapunov function where $\alpha > 0$ is to be defined later along the proof:

$$V(x_t, y_t) := (\Phi(x_t) - \Phi^\star) + \alpha(\Phi(x_t) - f(x_t, y_t)).$$

For convenience, with $U_t, W_t$ we will denote the quantities:

$$U_t := \Phi(x_t) - \Phi^\star$$
$$W_t := \Phi(x_t) - f(x_t, y_t),$$

and we can then write $V_t = U_t + \alpha W_t$. Piecing together (32), and (36)

$$U_{t+1} + \alpha W_{t+1} \leq U_t - \frac{\tau_x}{6}\mathbb{E}\mathcal{D}_{\mathcal{X}}^{\Phi}(x_t, 1/\tau_x) + \tau_x \mathbb{E}\left\|\nabla_x \Phi(x_t) - \nabla_x f(x_t, y_t)\right\|^2$$
$$+ 2\tau_x \sigma_x^2 + 2\tau_x \delta_x^2$$

$$+ \alpha \left( 1 - \frac{\mu_y \tau_y}{6} \right) W_t$$

$$+ \alpha \left( 1 - \frac{\mu_y \tau_y}{6} \right) \frac{3\tau_x}{2} \mathcal{D}_{\mathcal{X}}(x_t, 1/\tau_x; y_t)$$

$$+ \alpha \left( 1 - \frac{\mu_y \tau_y}{6} \right) \left( -\frac{\tau_x}{6} \mathbb{E} \mathcal{D}_{\mathcal{X}}^{\Phi}(x_t, 1/\tau_x) + \tau_x \mathbb{E} \left\| \nabla_x \Phi(x_t) - \nabla_x f(x_t, y_t) \right\|^2 \right)$$

$$+ \alpha \left( 7 \left( 1 - \frac{\mu_y \tau_y}{6} \right) \tau_x \sigma_x^2 + 7 \left( 1 - \frac{\mu_y \tau_y}{6} \right) \tau_x \delta_x^2 + \tau_y \sigma_y^2 + \tau_y \delta_y^2 \right)$$

$$\leq U_t + \left( -\frac{\tau_x}{6} - \frac{\alpha \tau_x}{6} \left( 1 - \frac{\mu_y \tau_y}{6} \right) + \alpha \left( 1 - \frac{\mu_y \tau_y}{6} \right) \frac{3\tau_x}{2} \right) \mathbb{E} \mathcal{D}_{\mathcal{X}}^{\Phi}(x_t, 1/\tau_x)$$

$$+ \left( \tau_x + \alpha \tau_x \left( 1 - \frac{\mu_y \tau_y}{6} \right) \right) \mathbb{E} \left\| \nabla_x \Phi(x_t) - \nabla_x f(x_t, y_t) \right\|^2$$

$$+ \alpha \left( 1 - \frac{\mu_y \tau_y}{6} \right) W_t$$

$$+ \alpha \left( 1 - \frac{\mu_y \tau_y}{6} \right) \frac{3\tau_x}{2} \left| \mathcal{D}_{\mathcal{X}}(x_t, 1/\tau_x; y_t) - \mathcal{D}_{\mathcal{X}}^{\Phi}(x_t, 1/\tau_x) \right|$$

$$+ \left( 2 + 7\alpha \left( 1 - \frac{\mu_y \tau_y}{6} \right) \right) \tau_x \sigma_x^2 + \left( 2 + 7\alpha \left( 1 - \frac{\mu_y \tau_y}{6} \right) \right) \tau_x \delta_x^2 + \alpha \tau_y \sigma_y^2 + \alpha \tau_y \delta_y^2 \qquad (37)$$

$$\leq U_t + \left( -\frac{\tau_x}{6} - \frac{\alpha \tau_x}{6} \left( 1 - \frac{\mu_y \tau_y}{6} \right) + \alpha \left( 1 - \frac{\mu_y \tau_y}{6} \right) \frac{3\tau_x}{2} \right) \mathbb{E} \mathcal{D}_{\mathcal{X}}^{\Phi}(x_t, 1/\tau_x)$$

$$+ \left( \tau_x + \alpha \tau_x \left( 1 - \frac{\mu_y \tau_y}{6} \right) \right) \frac{2\ell^2}{\mu_{\text{QG}}} \left( \mathbb{E} \Phi(x_t) - \mathbb{E} f(x_t, y_t) \right)$$

$$+ \alpha \left( 1 - \frac{\mu_y \tau_y}{6} \right) W_t$$

$$+ \alpha \left( 1 - \frac{\mu_y \tau_y}{6} \right) \frac{9\tau_x \ell^2}{\mu_{\text{QG}}} \left( \mathbb{E} \Phi(x_t) - \mathbb{E} f(x_t, y_t) \right)$$

$$+ \left( 2 + 7\alpha \left( 1 - \frac{\mu_y \tau_y}{6} \right) \right) \tau_x \sigma_x^2 + \left( 2 + 7\alpha \left( 1 - \frac{\mu_y \tau_y}{6} \right) \right) \tau_x \delta_x^2 + \alpha \tau_y \sigma_y^2 + \alpha \tau_y \delta_y^2 \qquad (38)$$

$$\leq \left( 1 - \frac{\mu_x}{2} \left( \frac{\tau_x}{6} - \frac{\alpha \tau_x}{6} \left( 1 - \frac{\mu_y \tau_y}{6} \right) + \alpha \left( 1 - \frac{\mu_y \tau_y}{6} \right) \frac{3\tau_x}{2} \right) \right) U_t$$

$$+ \left( \tau_x + \alpha \tau_x \left( 1 - \frac{\mu_y \tau_y}{6} \right) \right) \frac{2\ell^2}{\mu_{\text{QG}}} W_t$$

$$+ \alpha \left( 1 - \frac{\mu_y \tau_y}{6} \right) W_t$$

$$+ \alpha \left( 1 - \frac{\mu_y \tau_y}{6} \right) \frac{9\tau_x \ell^2}{\mu_{\text{QG}}} W_t$$

$$+ \left( 2 + 7\alpha \left( 1 - \frac{\mu_y \tau_y}{6} \right) \right) \tau_x \sigma_x^2 + \left( 2 + 7\alpha \left( 1 - \frac{\mu_y \tau_y}{6} \right) \right) \tau_x \delta_x^2 + \alpha \tau_y \sigma_y^2 + \alpha \tau_y \delta_y^2 \qquad (39)$$

- In (37) we use the fact $a \leq |a - b| + b$ with $a = \mathcal{D}_{\mathcal{X}}(x_t, 1/\tau_x; y_t), b = \mathcal{D}_{\mathcal{X}}^{\Phi}(x_t, 1/\tau_x)$ in order to remove the term $\mathcal{D}_{\mathcal{X}}(x_t, 1/\tau_x; y_t)$ and introduce the terms $\left| \mathcal{D}_{\mathcal{X}}(x_t, 1/\tau_x; y_t) - \mathcal{D}_{\mathcal{X}}^{\Phi}(x_t, 1/\tau_x) \right|$ and $\mathcal{D}_{\mathcal{X}}^{\Phi}(x_t, 1/\tau_x)$;

- In (38), we use the following facts:

  1. $\left| \mathcal{D}_{\mathcal{X}}(x_t, 1/\tau_x; y_t) - \mathcal{D}_{\mathcal{X}}^{\Phi}(x_t, 1/\tau_x) \right| \leq 3\ell^2 \left\| y_t - y^{\star}(x_t) \right\|^2$, Lemma B.12;

  2. $\left\| \nabla_x \Phi(x_t) - \nabla_x f(x_t, y_t) \right\|^2 \leq \ell^2 \left\| y_t - y^{\star}(x_t) \right\|^2$;

  3. $\frac{\mu_{\text{QG}}}{2} \left\| y_t - y^{\star}(x_t) \right\|^2 \leq \Phi(x_t) - f(x_t, y_t)$, quadratic growth condition of pPL functions Proposition C.5.

- In (39) we use the fact that $\frac{1}{2} \mathcal{D}_{\mathcal{X}}^{\Phi}(x, 1/\tau_x) \geq \mu_x U_t$ (Lemma D.4) and the negativity of the coefficient of $\mathbb{E} \mathcal{D}_{\mathcal{X}}^{\Phi}(x_t, 1/\tau_x)$, *i.e.*, the display $\left( -\frac{\tau_x}{6} - \frac{\alpha \tau_x}{6} \left( 1 - \frac{\mu_y \tau_y}{6} \right) + \alpha \left( 1 - \frac{\mu_y \tau_y}{6} \right) \frac{3\tau_x}{2} \right)$. To ensure negativity, we require that $\alpha < 1$ and utilize the fact that $\mu_x \leq \ell$ and hence $\tau_x \mu_x \leq 1$ by the choice of step-size.

Summarizing:

$$U_{t+1} + \alpha W_{t+1} \le \underbrace{\left(1 - \frac{\mu_x \tau_x}{3} + \frac{\alpha \mu_x \tau_x}{3}\left(1 - \frac{\mu_y \tau_y}{6}\right)\right)}_{\varpi_1} U_t$$

$$+ \alpha \underbrace{\left(1 + \frac{11\ell^2 \tau_x}{\mu_{\mathrm{QG}}} - \frac{11\mu_y \ell^2 \tau_x \tau_y}{6\mu_{\mathrm{QG}}} - \frac{\mu_y \tau_y}{6} + \frac{2\ell^2 \tau_x}{\alpha \mu_{\mathrm{QG}}}\right)}_{\varpi_2} W_t$$

$$+ \underbrace{\left(2 + 7\alpha\left(1 - \frac{\mu_y \tau_y}{6}\right)\right)\tau_x \sigma_x^2 + \left(2 + 7\alpha\left(1 - \frac{\mu_y \tau_y}{6}\right)\right)\tau_x \delta_x^2 + \alpha \tau_y \sigma_y^2 + \alpha \tau_y \delta_y^2}_{\xi}$$

**Tuning the parameters:** We need to ensure that $0 < \varpi_1, \varpi_2 < 1$ and then, we can show that the value of the Lyapunov function is contracting, *i.e.* $U_{t+1} + \alpha W_{t+1} \le \max\{\varpi_1, \varpi_2\}(U_t + \alpha W_t) + \xi$. We wiil now upper-bound $\varpi_1, \varpi_2$. For $\varpi_1$ we see that:

$$1 - \frac{\mu_x \tau_x}{3} + \frac{\alpha \mu_x \tau_x}{3}\left(1 - \frac{\mu_y \tau_y}{6}\right) \le 1 - \frac{\tau_x \mu_x}{3} + \frac{\alpha \mu_x \tau_x}{3} = 1 - \frac{29\mu_x \tau_x}{30} \le 1 - \mu_x \tau_x.$$

For $\varpi_2$,

$$1 - \frac{\mu_y \tau_y}{6} + \frac{2\tau_x \ell^2}{\alpha \mu_{\mathrm{QG}}} + \tau_x\left(1 - \frac{\mu_y \tau_y}{6}\right)\frac{2\ell^2}{\mu_{\mathrm{QG}}} + \left(1 - \frac{\mu_y \tau_y}{6}\right)\frac{9\tau_x \ell^2}{\mu_{\mathrm{QG}}}$$

$$= 1 - \frac{\ell^2 \tau_x}{\mu_y}\left(\frac{\mu_y^2 \tau_y}{\tau_x \ell^2} - \frac{2\mu_y}{\alpha} - \left(1 - \frac{\mu_y \tau_y}{6}\right)11\right)$$

$$\le 1 - \frac{\ell^2 \tau_x}{\mu_y}\left(\frac{\mu_y^2 \tau_y}{\tau_x \ell^2} - \frac{2}{\alpha} - 11\right)$$

$$\le 1 - \frac{\ell^2 \tau_x}{\mu_y}\left(\frac{\mu_y^2 \tau_y}{\tau_x \ell^2} - 31\right)$$

$$\le 1 - \frac{\ell^2 \tau_x}{\mu_y},$$

where we have used the fact that $\mu_{\mathrm{QG}}$ is equal to $\mu_y$ by Proposition C.5 and we require that $\frac{\mu_y^2 \tau_y}{\tau_x \ell^2} = 32$. As such, we $\tau_x = \frac{\mu_y^2 \tau_y}{32\ell^2}$ and $\tau_y = \frac{1}{5\ell}$. Further, We observe that $1 - \mu_x \tau_x = \max\left\{1 - \frac{\ell^2 \tau_x}{\mu_y}, 1 - \mu_x \tau_x\right\}$ since $\ell \ge \mu_x, \mu_y$. Combining the pieces:

$$\mathbb{E} V_{t+1} \le (1 - \mu_x \tau_x)\mathbb{E} V_t + 3\tau_x \sigma_x^2 + 3\tau_x \delta_x^2 + \tau_y \sigma_y^2 + \tau_y \delta_y^2$$

Applying the inequality recursively and using the formula for the sum of the geometric sequence, we get,

$$\mathbb{E} V_T \le (1 - \mu_x \tau_x)^T V_0 + \frac{3\sigma_x^2}{\mu_x} + \frac{3\delta_x^2}{\mu_x} + + \frac{32\ell^2 \sigma_y^2}{\mu_x \mu_y^2} + \frac{32\ell^2 \delta_y^2}{\mu_x \mu_y^2}$$

$$\le \exp\left(-\mu_x \tau_x T\right) V_0 + \frac{3\sigma_x^2}{\mu_x} + \frac{3\delta_x^2}{\mu_x} + \frac{32\ell^2 \sigma_y^2}{\mu_x \mu_y^2} + \frac{32\ell^2 \delta_y^2}{\mu_x \mu_y^2}$$

Further, we note that since $f, \Phi$ are $L$-Lipschitz continuous and their domains have bounded diameters, we can bound $V_0$ as $V_0 \le L(\mathrm{D}_{\mathcal{X}} + \mathrm{D}_{\mathcal{Y}})$. Finally, we see that by the choice of step-sizes $\mu_x \tau_x = \frac{\mu_x \mu_y^2}{160\ell^3}$.

$\square$

# E. Convex Markov Games

**Lemma E.1** (Continuity of the occupancy measure). *Let $\lambda \in \Delta(\mathcal{S} \times \mathcal{A} \times \mathcal{B})$ be the occupancy measure in a Markov game. Then $\lambda$ is $L_\lambda$-Lipschitz continuous and $\ell_\lambda$-smooth with respect to the policy pair $(x, y) \in \mathcal{X} \times \mathcal{Y}$. Specifically, for all $(x, y)$ and $(x', y')$,*

$$\|\lambda(x, y) - \lambda(x', y')\| \leq L_\lambda \|(x, y) - (x', y')\| \, ;$$
$$\|\nabla\lambda(x, y) - \nabla\lambda(x', y')\| \leq \ell_\lambda \|(x, y) - (x', y')\| \, ,$$

*where $L_\lambda := \frac{|\mathcal{S}|^{\frac{1}{2}}(|\mathcal{A}| + |\mathcal{B}|)}{(1-\gamma)^2}$, and $\ell_\lambda := \frac{2\gamma|\mathcal{S}|^{\frac{1}{2}}(|\mathcal{A}| + |\mathcal{B}|)^{\frac{3}{2}}}{(1-\gamma)^3}$.*

*Moreover, consider the functions $\lambda_1^{-1} : \Delta(\mathcal{S} \times \mathcal{A}) \to \mathcal{X}$ and $\lambda_2^{-1} : \Delta(\mathcal{S} \times \mathcal{B}) \to \mathcal{Y}$, such that:*

$$\lambda_1^{-1}\left(\lambda_1(x, y)\right) = x;$$
$$\lambda_2^{-1}\left(\lambda_2(x, y)\right) = y.$$

*For any fixed $y$ (respectively, $x$), $\lambda^{-1}$ is $L_{\lambda^{-1}}$-Lipschitz continuous with respect to $\lambda_1$ (respectively, $\lambda_2$), i.e., for all $\lambda_1(x, y), \lambda_1(x', y)$—respectively, $\lambda_2(x, y), \lambda_2(x, y')$,*

$$\|x - x'\| \leq L_{\lambda^{-1}} \left\|\lambda_1^{-1}\left(\lambda_1(x, y) - \lambda_1(x', y)\right)\right\| \, ;$$
$$\|y - y'\| \leq L_{\lambda^{-1}} \left\|\lambda_2^{-1}\left(\lambda_2(x, y) - \lambda_2(x, y')\right)\right\| \, ,$$

*with $L_{\lambda^{-1}} := \frac{2}{\min_s \varrho(s)(1-\gamma)}$.*

*Proof.* (Kalogiannis et al., 2024, Lemmata C.2 & C.3). $\qquad\qquad\qquad\qquad\qquad\qquad\qquad\qquad\qquad\qquad$ $\square$

Throughout, we will assume $\varepsilon$-greedy parametrization of the policies, *i.e.,* the players play according to policies:

$$\pi_x = (1 - \varepsilon)x + \frac{\varepsilon\mathbf{1}}{|\mathcal{A}|}, \quad \text{and} \quad \pi_y = (1 - \varepsilon)y + \frac{\varepsilon\mathbf{1}}{|\mathcal{B}|},$$

where $\mathbf{1}$ is the all-ones vector of appropriate dimension.

### E.1. Properties of the cMG Utility Fucntions

We reinstate that the utility function $L_U$-Lipschitz continuous and $\ell_U$-smooth with,

$$L_U = \frac{L_F|\mathcal{S}|^{\frac{1}{2}}(|\mathcal{A}| + |\mathcal{B}|)}{(1 - \gamma)^2}; \quad \ell_U = \frac{2\ell_F\gamma|\mathcal{S}|^{\frac{1}{2}}(|\mathcal{A}| + |\mathcal{B}|)^{\frac{3}{2}}}{(1 - \gamma)^3}.$$

#### E.1.1. PROPERTIES OF CMGS WITH CONVEX UTILITIES

For the two-player zero-sum cMG with merely concave utilites, we consider the regularized utility function $U^\mu(x, y) := U(x, y) + \frac{\mu}{2}\|\lambda_2(x, y)\|^2$. We note that that an upper bound to the Lipschitz smoothness moduli of $U^\mu, \nabla U^\mu$ is given by $L_U^\mu, \ell_U^\mu$:

$$L_U^\mu = 2L_U L_\lambda^2 = \frac{L_F|\mathcal{S}|^{\frac{3}{2}}(|\mathcal{A}| + |\mathcal{B}|)^3}{(1 - \gamma)^6}; \quad \ell_U^\mu = \frac{4\ell_F\gamma^2|\mathcal{S}|^{\frac{3}{2}}(|\mathcal{A}| + |\mathcal{B}|)^4}{(1 - \gamma)^8}.$$

Let us now compute the moduli of the quadratic growth and pPL condition,

$$\mu_{\mathrm{QG}} = \frac{(\min_s \varrho(s))^2(1 - \gamma)^2\mu}{4}; \quad \mu_{\mathrm{PL}} = \frac{(\min_s \varrho(s))^4(1 - \gamma)^{12}\mu^2}{4\ell_F\gamma^2|\mathcal{S}|^{\frac{3}{2}}(|\mathcal{A}| + |\mathcal{B}|)^4}$$

Since the quantity $\mu_{\mathrm{QG}}\mu_{\mathrm{PL}}$ will frequently appear in our calculations, we write:

$$\mu_{\mathrm{QG}}\mu_{\mathrm{PL}} = \frac{(\min_s \varrho(s))^6(1 - \gamma)^{14}\mu^3}{4\ell_F\gamma^2|\mathcal{S}|^{\frac{3}{2}}(|\mathcal{A}| + |\mathcal{B}|)^4}$$

The condition number $\kappa$ is defined as $\kappa := \frac{\ell_U^\mu}{\sqrt{\mu_{\mathrm{QG}}\mu_{\mathrm{PL}}}}$ and is equal to:

$$\kappa = \frac{16\ell_F^{\frac{3}{2}}\gamma^3|\mathcal{S}|^{\frac{9}{4}}(|\mathcal{A}| + |\mathcal{B}|)^6}{(\min_s \varrho(s))^3(1 - \gamma)^{15}\mu^{\frac{3}{2}}}$$

Finally, we define function $\Phi^\mu := \max_{y \in \mathcal{Y}} U^\mu(x, y)$ and observe for its Lipschitz modulus, $L_\Phi^\mu$, and its gradient's Lipschitz modulus, $\ell_\Phi^\mu$:

$$L_\Phi^\mu = L_U^\mu = \frac{L_F|\mathcal{S}|^{\frac{3}{2}}(|\mathcal{A}| + |\mathcal{B}|)^3}{(1 - \gamma)^6}; \quad \ell_\Phi^\mu = 2\ell_U^\mu\kappa = \frac{512\ell_F^{\frac{9}{2}}\gamma^8|\mathcal{S}|^{\frac{23}{4}}(|\mathcal{A}| + |\mathcal{B}|)^{\frac{23}{2}}}{(1 - \gamma)^{23}(\min_s \varrho(s))^3\mu^{\frac{3}{2}}}.$$

#### E.1.2. PROPERTIES OF CMGS WITH STRONGLY CONCAVE UTILITIES

We carry over the same calculations for the cMG with strongly concave utilities. Since we do not perturb the utilities, the relevant Lipschitz moduli remain the same. This is not the case for $\mu_{\mathrm{QG}}, \mu_{\mathrm{PL}}$. We write:

$$\mu_{\mathrm{QG}} = \frac{(\min_s \varrho(s))^2(1 - \gamma)^2\mu}{4}; \quad \mu_{\mathrm{PL}} = \frac{(\min_s \varrho(s))^4(1 - \gamma)^7\mu^2}{4\ell_F\gamma|\mathcal{S}|^{\frac{1}{2}}(|\mathcal{A}| + |\mathcal{B}|)^{\frac{3}{2}}}$$

Further, $\kappa := \frac{\ell_U}{\sqrt{\mu_{\mathrm{QG}}\mu_{\mathrm{PL}}}}$ which is equal to:

$$\kappa = \frac{4\sqrt{2}\ell_F^{\frac{3}{2}}\gamma^{\frac{3}{2}}|\mathcal{S}|^{\frac{3}{4}}(|\mathcal{A}| + |\mathcal{B}|)^{\frac{9}{4}}}{(\min_s \varrho(s))^3(1 - \gamma)^{\frac{9}{2}}\mu^{\frac{3}{2}}}.$$

Finally, the Lipschitz modulus of $\Phi$ and its gradient will be:

$$L_\Phi = L_U = \frac{L_F|\mathcal{S}|^{\frac{1}{2}}(|\mathcal{A}| + |\mathcal{B}|)}{(1 - \gamma)^2}; \quad \ell_\Phi = \frac{16\sqrt{2}\ell_F^{\frac{5}{2}}\gamma^{\frac{5}{2}}|\mathcal{S}|^{\frac{5}{4}}(|\mathcal{A}| + |\mathcal{B}|)^{\frac{15}{4}}}{(\min_s \varrho(s))^3(1 - \gamma)^{\frac{15}{2}}\mu^{\frac{3}{2}}}.$$

### E.2. Stochastic Estimators

Using a trajectory $\xi := \left(s^{(0)}, a^{(0)}, s^{(1)}, a^{(1)}, \dots\right)$ and $\xi := \left(s^{(0)}, b^{(0)}, s^{(1)}, b^{(1)}, \dots\right)$ respectively, we define the estimates of $\lambda_1, \lambda_2$,

$$\hat{\lambda}_{1,\xi}^{s,a} := \sum_{h=1}^{H_\xi} \gamma^h \mathbb{1}\{s^{(h)} = s, a^{(h)} = a\};$$

$$\hat{\lambda}_{2,\xi}^{s,b} := \sum_{h=1}^{H_\xi} \gamma^h \mathbb{1}\{s^{(h)} = s, b^{(h)} = b\}.$$

Additionally, for a pseudo-reward vector $z \in \mathbb{R}^{|\mathcal{S}| \times |\mathcal{A}|}$ and $\xi := \left(s^{(0)}, a^{(0)}, s^{(1)}, a^{(1)}, \dots\right)$—or $z \in \mathbb{R}^{|\mathcal{S}| \times |\mathcal{B}|}$ corresponding and $\xi := \left(s^{(0)}, b^{(0)}, s^{(1)}, b^{(1)}, \dots\right)$—we define gradient estimates $\hat{\nabla}_{x,\xi}^t, \hat{\nabla}_{y,\xi}^t$:

$$\hat{g}_x(\xi | x_t, y_t, z) = \sum_{h=0}^{H-1} \left[ \gamma^h z\left(s^{(h)}, a^{(h)}\right) \cdot \left( \sum_{h'=0}^{h} \nabla_x \log x\left(a^{(h')} | s^{(h')}\right) \right) \right]$$

$$\hat{g}_y(\xi | x_t, y_t, z) = \sum_{h=0}^{H-1} \left[ \gamma^h z\left(s^{(h)}, b^{(h)}\right) \cdot \left( \sum_{h'=0}^{h} \nabla_y \log y\left(b^{(h')} | s^{(h')}\right) \right) \right]$$

At every iteration $t$, each agent constructs $\hat{r}_{x,t} \leftarrow \nabla_{\lambda_1} F(\hat{\lambda}_{1,t})$ describe how to construct $\hat{r}_x, \hat{r}_y$ batch versions of the latter estimators will be,

$$\hat{\nabla}_x^t := \sum_{i=1}^{M_x} \hat{g}_x(\xi_i | x_t, y_t, \hat{r}_{x,t}), \quad \text{and} \quad \hat{\nabla}_y^t := \sum_{i=1}^{M_y} \hat{g}_y(\xi_i | x_t, y_t, \hat{r}_{y,t}). \tag{40}$$

Now, for each one of the two cases we let $z \leftarrow \nabla_{\lambda_1} F\left(\hat{\lambda}_{1,t}; y_t\right)$ and $z \leftarrow \nabla_{\lambda_1} F\left(\hat{\lambda}_{1,t}; y_t\right)$ correspondingly.

*Remark* 4. This stochastic gradient estimator assumes access to a gradient oracle for $\nabla_{\lambda_1} F$. Access to an oracle for $F$ or $\nabla_\lambda F$ is an assumption made in virtually all the references that we have encountered concerning policy gradient methods for MDPs with general utilities (Zhang et al., 2020; 2021; Barakat et al., 2023). Designing a gradient estimator that entirely relies on samples lies well-beyond the scope of our paper. Nevertheless, this assumption is not a strong one as even with a complete knowledge of $F$, agents cannot control the function's arguments except for affecting them implicitly.

**The occupancy measure estimator** Let a trajectory sample $\xi$ of length $H_\xi$, we define the occupancy estimator for player 1 to be,

$$\hat{\lambda}_1^{s,a}(\xi) := \sum_{h=1}^{H_\xi} \gamma^h \mathbb{1}\{s^{(h)} = s, a^{(h)} = a\}$$

$$\hat{\lambda}_2^{s,b}(\xi) := \sum_{h=1}^{H_\xi} \gamma^h \mathbb{1}\{s^{(h)} = s, b^{(h)} = b\}$$

Assume policies $x_t, y_t$ and corresponding sampled trajectories $\{\xi_{i,t}\}_{i=1}^{M_x}$ and $\{\xi'_{i,t}\}_{i=1}^{M_x}$ for player 1 and 2 correspondingly. The occupancy batch estimators $\hat{\lambda}_{1,t}, \hat{\lambda}_{2,t}$

$$\hat{\lambda}_{1,t} := \frac{1}{M_x} \sum_{i=1}^{M_x} \hat{\lambda}_1(\xi_{i,t}), \qquad \hat{\lambda}_{2,t} := \frac{1}{M_x} \sum_{i=1}^{M_x} \hat{\lambda}_2(\xi'_{i,t})$$

$$\lambda_{1,H}^{s,a}(x,y) := \mathbb{E}_{x,y}\left[ \sum_{h=1}^{H} \gamma^h \mathbb{1}\{s^{(h)} = s, a^{(h)} = a\} \mathbb{P}\left(s^{(h+1)} | a^{(h)}, b^{(h)}, s^{(h)}\right) \Big| s^{(0)} \sim \varrho \right]$$

$$\lambda_{2,H}^{s,b}(x,y) := \mathbb{E}_{x,y}\left[\sum_{h=1}^{H}\gamma^h \mathbb{1}\{s^{(h)} = s, b^{(h)} = b\}\,\mathbb{P}\left(s^{(h+1)}|a^{(h)}, b^{(h)}, s^{(h)}\right)\Big|s^{(0)} \sim \varrho\right]$$

**Lemma E.2.** *Let an empirical estimate $\hat{\lambda}_1$ of the truncated-horizon occupancy measure $\lambda_{1,H}$. Then, the variance of the estimate can be bounded as:*

$$\mathbb{E}\left[\left\|\hat{\lambda}_1 - \lambda_{1,H}\right\|^2\right] \le \frac{1}{M_x(1-\gamma)^2}.$$

**The policy gradient estimator**   Assuming $\varepsilon$-greedy parametrization to control the variance of our estimators, we can go forward and state our main lemmata regarding the gradient policy gradient estimator's variances.

**Lemma E.3.** *Let $\hat{\nabla}_x^t$ be defined as in* (40)*. The estimators variance can be bounded as:*

$$\mathbb{E}\left[\left\|\hat{\nabla}_x^t - \nabla_x F(\lambda_{1,H}(x_t, y_t); y_t)\right\|^2\right] \le \frac{27 L_F^2}{M_x(1-\gamma)^6 \varepsilon^2}.$$

*Proof.* We essentially carry over the same computation as Step 2 in the proof of (Zhang et al., 2021)[Lemma F.2] using our notation and assumptions. To simplify the excessively busy notation and improve readability, let us introduce some shortcuts,

- $\overline{\lambda}_1 := \lambda_{1,H}(x_t, y_t)$,

- $\hat{\lambda}_1 := \hat{\lambda}_{1,t}$,

- $r_* := \nabla_{\lambda_1} F\left(\lambda_{1,H}(x_t, y_t); y_t\right) = \nabla_{\lambda_1} F\left(\overline{\lambda}_1; y_t\right)$,

- $\mathbf{J} := \nabla_x \lambda_{1,H}(x_t, y_t) = \nabla_x \overline{\lambda}_1$,

- $\hat{r} := \nabla_{\lambda_1} F\left(\hat{\lambda}_1; y_t\right)$,

- $g := \nabla_x F\left(\lambda_{1,H}(x_t, y_t); y_t\right) = \nabla_x F\left(\overline{\lambda}_1; y_t\right)$,

- $\hat{g}_i := \hat{g}_x(\xi_i | x_t, y_t, \hat{r})$,

- $\hat{g}_{i,*} := \hat{g}_x(\xi_i | x_t, y_t, r_*)$.

We can finally write:

$$\mathbb{E}\left[\|\hat{\nabla}_x^t - \mathbf{J}^\top \hat{r}\|^2\right] = \mathbb{E}\left[\left\|\frac{1}{M_x}\sum_i^{M_x}\hat{g}_{i,*} + \frac{1}{M_x}\sum_i^{M_x}\hat{g}_{i,*} - g + g - \mathbf{J}^\top \hat{r}\right\|^2\right]$$

$$\le 3\mathbb{E}\left[\left\|\frac{1}{M_x}\sum_i^{M_x}\left(\hat{g}_i - \hat{g}_{i,*}\right)\right\|^2\right] + 3\mathbb{E}\left[\left\|g - \mathbf{J}^\top \hat{r}_t\right\|^2\right] + 3\mathbb{E}\left[\left\|\frac{1}{M_x}\sum_i^{M_x}\hat{g}_{i,*} - g\right\|^2\right].$$

For the first term we write,

$$\begin{aligned}
\mathbb{E}\left[\left\|\frac{1}{M_x}\sum_i^{M_x}\left(\hat{g}_i - \hat{g}_{i,*}\right)\right\|^2\right] &\le \frac{1}{M_x}\sum_i^{M_x}\mathbb{E}\left[\left\|\hat{g}_i - \hat{g}_{i,*}\right\|^2\right] \\
&\le \frac{4}{(1-\gamma)^4\varepsilon^2}\cdot\mathbb{E}\left[\|\hat{r} - r_*\|_\infty^2\right] \\
&\le \frac{4 L_F^2}{(1-\gamma)^4\varepsilon^2}\mathbb{E}\left[\left\|\hat{\lambda}_1 - \overline{\lambda}_1\right\|^2\right] \\
&\le \frac{4 L_F^2}{M_x(1-\gamma)^6\varepsilon^2}.
\end{aligned} \tag{41}$$

Where (41) follows from the following fact:

$$
\begin{aligned}
\|\hat{g}(\xi|x,r_1) - \hat{g}(\xi|x,r_2)\| &= \left\| \sum_{t=0}^{H-1} \gamma^t \cdot (r_1(s_t,a_t) - r_2(s_t,a_t)) \left( \sum_{t'=0}^{t} \nabla_x \log \pi_x(a_{t'}|s_{t'}) \right) \right\| \\
&\leq \sum_{t=0}^{H-1} \frac{1}{\varepsilon} \gamma^t (t+1) \|r_1 - r_2\|_\infty \\
&\leq \frac{2\|r_1 - r_2\|}{(1-\gamma)^2 \varepsilon}.
\end{aligned}
$$

The second and third terms can be similarly bounded as:

$$
\mathbb{E}\left[ \left\| g^t - \mathbf{J}^\top \hat{r}_t \right\|^2 \right] \leq \frac{L_F^2}{M_x (1-\gamma)^4};
$$

and

$$
\mathbb{E}\left[ \left\| \frac{1}{M_x} \sum_{i}^{M_x} \hat{g}_{i,*} - g \right\|^2 \right] \leq \frac{4L_F^2}{M_x (1-\gamma)^6 \varepsilon^2}.
$$

$\square$

**Lemma E.4.** *For any $H > \frac{1}{\ln(1\sqrt{\gamma})}$ and a greedy exploration parameter $\varepsilon$, the following inequality holds true for the bias of the policy gradient estimator defined in Equation* (40)*:*

$$
\|\nabla_x F(\lambda_H(x,y); y) - \nabla_x F(\lambda_1(x,y), y)\|^2 \leq \frac{256 L_F}{(1-\gamma)^6 \varepsilon^2} \exp\Big(-(1-\gamma)(H-1)\Big).
$$

*Proof.* By (Zhang et al., 2021, Lemma E.3), we have that,

$$
\|\nabla_x F(\lambda_H(x,y); y) - \nabla_x F(\lambda_1(x,y), y)\|^2 \leq \left( \frac{8L_F^2}{(1-\gamma)^6 \varepsilon^2} + 16 \frac{L_F^2}{\varepsilon^2} \left( \frac{(H+1)^2}{(1-\gamma)^2} + \frac{1}{(1-\gamma)^4} \right) \right) \gamma^{2H}.
$$

Further, we can bound the RHS as:

$$
\left( \frac{8L_F^2}{(1-\gamma)^6 \varepsilon^2} + 16 \frac{L_F^2}{\varepsilon^2} \left( \frac{(H+1)^2}{(1-\gamma)^2} + \frac{1}{(1-\gamma)^4} \right) \right) \gamma^{2H} \leq \frac{256 L_F (H+1)^2}{(1-\gamma)^6 \varepsilon^2} \gamma^{2H}.
$$

Now, in order to simplify the display, compute an $H > 0$ such that:

$$
\frac{(H+1)^2}{(1/\gamma)^{H+1}} \leq 1
$$

equivalently,

$$
2 \log_{1/\gamma}(H+1) \leq H+1
$$

changing the base of the logarithm, we get,

$$
\frac{1}{\ln(1/\sqrt{\gamma})} \ln(H+1) \leq H+1,
$$

but the latter is true for any $H + 1 > 1/\ln(1/\sqrt{\gamma})$. Hence, since $\gamma < 1$ and noting that $\gamma = 1 - (1-\gamma)$,

$$
\frac{256 L_F (H+1)^2}{(1-\gamma)^6 \varepsilon^2} \gamma^{2H} \leq \frac{256 L_F}{(1-\gamma)^6 \varepsilon^2} \gamma^{H-1} \leq \frac{256 L_F}{(1-\gamma)^6 \varepsilon^2} \exp\Big(-(1-\gamma)(H-1)\Big).
$$

$\square$

## E.3. Additional Supporting Claims

In this subsection we bound the errors incurred due to the $\varepsilon$-greedy parametrization, and the regularization.

**Error due to $\varepsilon$-greedy parametrization.**

**Claim E.5.** *Assume an $L$-Lipschitz continuous function $f : \mathcal{X} \to \mathbb{R}$. Further, let $\mathcal{X}$ be a concatenation of $n$ $m$-dimensional probability simplices. Further let $w$ be the mapping $w(x) := (1 - \varepsilon)x + \varepsilon/m$ for some $0 < \varepsilon < 1$.*

$$|f(x) - (f \circ w)(x)| \leq \frac{\sqrt{n}L\varepsilon}{m}.$$

*Further, if for some $\epsilon > 0$ and $x \in \mathcal{X}$,*

$$f(x) - f^\star \leq \epsilon,$$

*then*

$$(f \circ w)(x) - f^\star \leq \epsilon + \frac{\sqrt{n}L\varepsilon}{m}.$$

*The second follows directly.*

*Proof.* The claim follows from the Lipschitz continuity of $f$. It is the case that $(f \circ w)(x) = f((1 - \varepsilon)x + \varepsilon/m)$,

$$|f(x) - f((1 - \varepsilon)x + \varepsilon/m)| \leq L \|x - ((1 - \varepsilon)x + \varepsilon/m)\|$$
$$\leq \frac{\sqrt{n}L\varepsilon}{m}.$$

$\square$

**Claim E.6.** *Let an $L$-Lipschitz continuous function $f : \mathcal{X} \times \mathcal{Y} \to \mathbb{R}$. Let $\mathcal{X}, \mathcal{Y}$ be concatenations of $n_x$ and $n_y$ $m_x$- and $m_y$-dimensional probability simplices respectively. Further let $w$ be the mapping $w(x; \varepsilon, m) := (1 - \varepsilon)x + \varepsilon/m$ for some $0 < \varepsilon < 1$. Further, for some $\varepsilon_x, \varepsilon_y > 0$ define $f_w(x, y) := f(w(x; \varepsilon_x, m_x), w(y; \varepsilon_y, m_y))$, $\Phi_w(x) := \max_{y \in \mathcal{Y}} f_w(x, y)$, and $\Phi_w^\star := \min_{x \in \mathcal{X}} \Phi_w(x)$. Then, the following inequalities hold true,*

- $|\Phi_w(x) - \Phi(x)| \leq \frac{\sqrt{n_x}L\varepsilon_x}{m_x} + \frac{\sqrt{n_y}L\varepsilon_y}{m_y}$
- $|\Phi^\star - \Phi_w^\star| \leq \frac{\sqrt{n_x}L\varepsilon_x}{m_x} + \frac{\sqrt{n_y}L\varepsilon_y}{m_y}.$

*Proof.* The proof of the first item directly follows from an application of Claim E.5 and the $L$-Lipschitz continuity of $\Phi$ given that $f$ is $L$-Lipschitz continuous.

For the second item, we use the inequality we just attained and write,

$$\Phi_w(x) \leq \Phi(x) + \frac{\sqrt{n_x}L\varepsilon_x}{m_x} + \frac{\sqrt{n_y}L\varepsilon_y}{m_y}.$$

Minimizing over $x$ for the two sides yields,

$$\Phi_w(x_w^\star) \leq \Phi(x^\star) + \frac{\sqrt{n_x}L\varepsilon_x}{m_x} + \frac{\sqrt{n_y}L\varepsilon_y}{m_y}.$$

Where, $x^\star, x_w^\star$ are such that $x^\star \in \arg\min_{x \in \mathcal{X}} \Phi(x)$ and $x_w^\star \in \arg\min_{x \in \mathcal{X}} \Phi_w(x)$. Applying the same trick to the other side of the other direction of the inequality:

$$\Phi(x^\star) \leq \Phi_w(x_w^\star) + \frac{\sqrt{n_x}L\varepsilon_x}{m_x} + \frac{\sqrt{n_y}L\varepsilon_y}{m_y}.$$

As such,

$$|\Phi(x^\star) - \Phi_w(x_w^\star)| \leq \frac{\sqrt{n_x}L\varepsilon_x}{m_x} + \frac{\sqrt{n_y}L\varepsilon_y}{m_y}.$$

$\square$

**Claim E.7.** *Assume an $L$-Lipschitz continuous and $\ell$-smooth function $f : \mathcal{X} \to \mathbb{R}$ that is $\mu$-pPL. Further, let $\mathcal{X}$ be a concatenation of $n$ $m$-dimensional probability simplices. Further let $w$ be the mapping $w(x) := (1 - \varepsilon)x + \varepsilon/m$ for some $0 < \varepsilon < 1$. Define $\mathcal{D}_{\mathcal{X}}^{f}(x, \alpha), \mathcal{D}_{\mathcal{X}}^{f \circ w}(x, \alpha)$ for some $\alpha > 0$ to be:*

$$\mathcal{D}_{\mathcal{X}}^{f}(x, \alpha) := -2\alpha \min_{y \in \mathcal{X}} \left\{ \langle \nabla f(x), y - x \rangle + \frac{\alpha}{2} \|x - y\|^2 \right\}$$

$$\mathcal{D}_{\mathcal{X}}^{f \circ w}(\pi_x, \alpha) := -2\alpha \min_{y \in \mathcal{X}} \left\{ \langle \nabla f \circ w(x), y - x \rangle + \frac{\alpha}{2} \|x - y\|^2 \right\}.$$

*Then, it is the case that:*

$$\frac{1}{2} \mathcal{D}_{\mathcal{X}}^{f \circ w}(x, \alpha) \geq \mu \left( f(x) - f^\star \right) - 8\sqrt{n}\ell L \alpha \mathrm{D}_{\mathcal{X}} \varepsilon.$$

*Proof.* We essentially need to prove that in fact:

$$\left| \mathcal{D}_{\mathcal{X}}^{f \circ w}(x, \alpha) - \mathcal{D}_{\mathcal{X}}^{f}(x, \alpha) \right| \leq 16\sqrt{n}\ell L \alpha \mathrm{D}_{\mathcal{X}} \varepsilon.$$

We write $\mathcal{D}_{\mathcal{X}}^{f}(x, \alpha)$ equivalently as:

$$\mathcal{D}_{\mathcal{X}}^{f}(x, \alpha) = \max_{y \in \mathcal{X}} \left\{ 2\alpha \langle \nabla f(x), x - y \rangle - \alpha^2 \|x - y\|^2 \right\}.$$

We define $G(\cdot; y)$ after we isolate the display inside the $\max$-operator and in place of the gradient put any vector $v$ of the same dimension:

$$G(v; y) := 2\alpha \langle v, x - y \rangle - \alpha^2 \|x - y\|^2.$$

We see that $\nabla_v G(v; y) = 2\alpha(x - y)$ and $\|x - y\| \leq \mathrm{D}_{\mathcal{X}}$. As such, $G$ is $2\alpha \mathrm{D}_{\mathcal{X}}$-Lipschitz continuous in $v$. This consequently means that $\Psi(v) := \max_{y \in \mathcal{Y}} G(v; y)$ is also $2\alpha \mathrm{D}_{\mathcal{X}}$-Lipschitz in $v$. Now, all that is left to do is to bound the distance between $\nabla_x (f \circ w)(x)$ and $\nabla_x f(x)$. By the chain rule:

$$\nabla_x (f \circ w)(x) = (1 - \varepsilon) \nabla_x f(z) \Big|_{z = w(x)}.$$

Further, set $x_\varepsilon := w(x)$ and observe that

$$\begin{aligned}
\|\nabla_x f(x) - \nabla_x f(x_\varepsilon)\| &\leq \ell \|x - x_\varepsilon\| \\
&= \ell \|x - (1 - \varepsilon)x - \varepsilon/m\| \\
&= \ell \|\varepsilon x - \varepsilon/m\| \\
&\leq \ell \varepsilon (\sqrt{n} + 1/m).
\end{aligned}$$

Where in the last inequality we use the fact that for a probability vector $y$, $\|y\| \leq 1$ and the triangle inequality. Assuming that $n, m > 1$, we can further simplify into:

$$\|\nabla_x f(x) - \nabla_x f(x_\varepsilon)\| \leq 2\sqrt{n}\ell\varepsilon \tag{42}$$

Now, by the fact that $\Psi(\nabla_x f(x)) = \mathcal{D}_x^f(x, \alpha)$, $\Psi(\nabla_x (f \circ w)(x)) = \mathcal{D}_x^{f \circ w}(x, \alpha)$, and the Lipschitz continuity of $\Psi$, we write:

$$\begin{aligned}
\left| \mathcal{D}_{\mathcal{X}}^{f}(x, \alpha) - \mathcal{D}_{\mathcal{X}}^{f \circ w}(x, \alpha) \right| &\leq 2\alpha \mathrm{D}_{\mathcal{X}} \|\nabla_x (f \circ w)(x) - \nabla_x f(x)\| \\
&\leq 2\alpha \mathrm{D}_{\mathcal{X}} \|\nabla_x f(x) - \nabla_x f(x_\varepsilon)\| + 2\alpha \mathrm{D}_{\mathcal{X}} \|\varepsilon \nabla_x f(x_\varepsilon)\| \\
&\leq 4\sqrt{n}\ell \alpha \mathrm{D}_{\mathcal{X}} \varepsilon + 2\alpha \mathrm{D}_{\mathcal{X}} \varepsilon L \\
&\leq 16\sqrt{n}\ell L \alpha \mathrm{D}_{\mathcal{X}} \varepsilon.
\end{aligned}$$

The claim follows. $\qquad \square$

**Claim E.8.** *Assume an $L$-Lipschitz continuous and $\ell$-smooth function $f : \mathcal{X} \to \mathbb{R}$ such that:*

$$\max_{x' \in \mathcal{X}, \|x-x'\| \leq 1} \langle \nabla_x f(x), x - x' \rangle \geq \mu \left( f(x) - f^\star \right).$$

*Further, let $\mathcal{X}$ be a concatenation of $n$ $m$-dimensional probability simplices. Further let $w$ be the mapping $w(x) := (1 - \varepsilon)x + \varepsilon/m$ for some $0 < \varepsilon < 1$. It is the true that:*

$$\max_{x' \in \mathcal{X}, \|x-x'\| \leq 1} \langle \nabla_x (f \circ w)(x), x - x' \rangle \geq \mu \left( f(x) - f^\star \right) - 8\sqrt{n}\ell L \mathrm{D}_\mathcal{X} \varepsilon.$$

*Proof.* Similar to the previous case, we need to show that $G(v; x') := \langle v, x - x' \rangle$ is Lipschitz continuous in $v$. Indeed, $\nabla_v G(v; x') := x - x'$ and as such $\|\nabla_v G(v; x')\| \leq \mathrm{D}_\mathcal{X}$. Consequently, $\Psi(v) := \max_{x' \in \mathcal{X}} G(v; x')$ is also $\mathrm{D}_\mathcal{X}$-Lipschitz continuous. Further, as previously shown in (42),

$$\|\nabla_x f(x) - \nabla_x f(x_\varepsilon)\| \leq 2\sqrt{n}\ell\varepsilon.$$

We can finally write,

$$\max_{x' \in \mathcal{X}, \|x-x'\| \leq 1} \langle \nabla_x (f \circ w), x - x' \rangle \geq \max_{x' \in \mathcal{X}, \|x-x'\| \leq 1} \langle \nabla_x f, x - x' \rangle - 8\sqrt{n}\ell L \mathrm{D}_\mathcal{X} \varepsilon.$$

$\square$

**Error due to the regularizer.**

**Claim E.9.** *Let a two-player zero-sum cMG with utility $U : \mathcal{X} \times \mathcal{Y} \to \mathbb{R}$. Assume that the maximizing player employs regularization to their utility of the form:*

$$U^{\mu_{\mathrm{reg.}}}(x, y) := U(x, y) - \frac{\mu_{\mathrm{reg.}}}{2} \|\lambda_2(x, y)\|^2,$$

*where $\mu_{\mathrm{reg.}} > 0$. Then, the following inequality is true:*

$$\|\nabla_x U(x, y) - \nabla_x U^\mu(x, y)\| \leq \mu_{\mathrm{reg.}} \|L_\lambda\|.$$

*Proof.*

$$\begin{aligned}
\|\nabla_x U(x, y) - \nabla_x U^\mu(x, y)\| &= \mu_{\mathrm{reg.}} \|\nabla_x \lambda_2(x, y)\| \\
&\leq \mu_{\mathrm{reg.}} \|\lambda_2\| \|\nabla_x \lambda_2(x, y)\| \\
&\leq \mu_{\mathrm{reg.}} L_\lambda.
\end{aligned}$$

$\square$

## E.4. Nested Policy Gradient

### E.4.1. CMG WITH CONCAVE UTILITIES

**Theorem E.10.** *Assume a two-player zero-sum cMG and a desried accuracy $\epsilon > 0$. Running Algorithm 1 a tuning of Algorithm 1 with, $\mu_{\mathrm{reg.}} = \Theta \left( \frac{(1-\gamma)(\min_s \varrho(s))\epsilon}{L_F} \right)$ and,*

- *step-sizes, $\tau_x = \Theta \left( \frac{(\min_s \varrho(s))^5 (1-\gamma)^{\frac{18}{2}} \epsilon^{\frac{3}{2}}}{\ell_F^{\frac{5}{2}} \gamma^{\frac{5}{2}} |\mathcal{S}|^{\frac{5}{2}} (|\mathcal{A}|+|\mathcal{B}|)^{\frac{15}{4}}} \right)$ and $\tau_y = \Theta \left( \frac{(1-\gamma)^8}{4\ell_F \gamma^2 |\mathcal{S}|^{\frac{3}{2}} (|\mathcal{A}|+|\mathcal{B}|)^4} \right)$;*

- *exploration parameters $\varepsilon_x = \left( \frac{\epsilon(\min_s \varrho(s))^6 \mu^3 (1-\gamma)^{17}}{512 L_F |\mathcal{S}|^4 \ell_F^5 \gamma^5 (|\mathcal{A}|+|\mathcal{B}|)^{\frac{17}{2}}} \right)$ and $\varepsilon_y = \Theta \left( \frac{\epsilon(\min_s \varrho(s))^9 \mu_x^{\frac{9}{2}} (1-\gamma)^{\frac{21}{2}}}{|\mathcal{S}|^{\frac{5}{4}} \ell_F^{\frac{3}{2}} \gamma^{\frac{1}{2}} (|\mathcal{A}|+|\mathcal{B}|)^{\frac{9}{4}}} \right)$;*

- *batch-sizes $M_x = \Theta\left(\frac{L_F^4 \ell_F^2 |\mathcal{S}|^7 (|\mathcal{A}|+|\mathcal{B}|)^{11}}{\epsilon^4 (\min_s \varrho(s))^4 (1-\gamma)^{32}}\right)$ and $M_y = \Theta\left(\frac{\ell_F^{10} L_F^2 \gamma^8 |\mathcal{S}|^{10} (|\mathcal{A}|+|\mathcal{B}|)^{22}}{(\min_s \varrho(s))^{16} (1-\gamma)^{52} \epsilon^8}\right)$;*

- *an outer-loop number of iterations at least $T_x = \Theta\left(\frac{L_F \ell_F^{\frac{9}{2}} \gamma^8 |\mathcal{S}|^{\frac{29}{4}} (|\mathcal{A}|+|\mathcal{B}|)^{\frac{29}{2}}}{(1-\gamma)^{\frac{55}{2}} (\min_s \varrho(s))^{\frac{13}{2}} \epsilon^{\frac{7}{2}}}\right)$ and inner loop iterations that are at least*

$T_y = \Theta\left(\frac{\ell_U L_\lambda^2 \ell_F \gamma^2 |\mathcal{S}|^{\frac{3}{2}} (|\mathcal{A}|+|\mathcal{B}|)^4}{(\min_s \varrho(s))^6 (1-\gamma)^{14} \epsilon^2} \ln\left(\frac{\ell_F L_F \gamma |\mathcal{S}| (|\mathcal{A}|+|\mathcal{B}|)}{(\min_s \varrho(s))(1-\gamma)\epsilon}\right)\right)$;

*will output an iterate $x_{t^\star}, y_{t^\star}$, such that:*

$$\mathbb{E}\left[U(x_{t^\star}, y_{t^\star+1}) - \min_{x' \in \mathcal{X}} U(x', y_{t^\star+1})\right] \le \epsilon;$$

$$\mathbb{E}\left[\max_{y \in \mathcal{Y}} U(x_t, y') - \mathbb{E}U(x_{t^\star}, y_{t^\star})\right] \le \epsilon.$$

*Proof.* In order to simplify our arguments, we formalize $\varepsilon$-greedy parametrization as a composition of the utility function $U(x, y)$ with mappings $w(x; \varepsilon, m)$ where $w(x) := (1-\varepsilon)x + \varepsilon \mathbf{1}/m$. In particular, our convergence guarantees are w.r.t. the function $U_w^\mu(x, y) := U^\mu(w(x; \varepsilon_x, |\mathcal{A}|), w(y; \varepsilon_y, |\mathcal{B}|))$ and $\tilde{\Phi}(x) := \max_{y \in \mathcal{Y}} U_w^\mu(x, y)$.

We will tune $T_x, \tau_x, H_x, \varepsilon_x, M_x$, by, Theorem D.5.

$$\frac{1}{T} \sum_{t=0}^{T-1} \left\|\hat{\mathcal{G}}_{x,\tau_x}^{\tilde{\Phi},t}\right\|^2 \le \frac{5\ell_{\tilde{\Phi}} L_{\tilde{\Phi}} D_{\mathcal{X}}}{T} + 6\delta_x^2 + \frac{1}{M_x} \cdot \frac{4L_F^2}{(1-\gamma)^6 \varepsilon_x^2}.$$

First, we see that in order to achieve an $\epsilon$-approximate best-response w.r.t. to $x$, then, $x$ needs to be an $(1-\gamma)\min_s \varrho(s)\epsilon$-approximate stationary point. In general, then, we need to ensure that,

$$\delta_x^2 = \Theta(\epsilon^2 (1-\gamma)^2 (\min_s \varrho(s))^2).$$

Hence, by Claim E.9, we can pick the regulizer's coefficient to be:

$$\mu_{\text{reg.}} = \Theta\left(\frac{(1-\gamma)(\min_s \varrho(s))\epsilon}{L_F}\right).$$

Lemma E.4 dictates the tuning of $H_x, H_y$,

$$H_x, H_y = \Theta\left(\frac{1}{1-\gamma} \ln\left(\frac{\ell_F L_F \gamma |\mathcal{S}| (|\mathcal{A}|+|\mathcal{B}|)}{\epsilon(1-\gamma)\min_s \varrho(s)\mu_x}\right)\right).$$

Further,

$$\varepsilon_x = \Theta\left(\frac{(1-\gamma)(\min_s \varrho(s))\epsilon}{|\mathcal{S}| \ell_F L_F \ell_\lambda L_\lambda^4}\right) = \Theta\left(\frac{\epsilon \min_s \varrho(s) (1-\gamma)^{12}}{2 L_F \ell_F |\mathcal{S}|^{\frac{7}{2}} \gamma (|\mathcal{A}|+|\mathcal{B}|)^{\frac{11}{2}}}\right)$$

$$\tau_y = \Theta\left(\frac{1}{\ell_U^\mu}\right) = \Theta\left(\frac{1}{\ell_F \ell_\lambda^2 L_\lambda}\right) = \Theta\left(\frac{(1-\gamma)^8}{4\ell_F \gamma^2 |\mathcal{S}|^{\frac{3}{2}} (|\mathcal{A}|+|\mathcal{B}|)^4}\right)$$

$$\sigma_x^2 = \frac{L_F^4 \ell_F^2 |\mathcal{S}|^7 (|\mathcal{A}|+|\mathcal{B}|)^{11}}{\epsilon^2 (\min_s \varrho(s))^2 (1-\gamma)^{30}}$$

$$M_x = \Theta\left(\frac{L_F^4 \ell_F^2 |\mathcal{S}|^7 (|\mathcal{A}|+|\mathcal{B}|)^{11}}{\epsilon^4 (\min_s \varrho(s))^4 (1-\gamma)^{32}}\right)$$

As for the inner-loop, we need to set

$$\epsilon_y = \Theta\left(\frac{\mu_{\mathrm{QG}}(1-\gamma)^2(\min_s \varrho(s))^2\epsilon^2}{2\ell_U^2}\right) = \Theta\left(\frac{\min_s \varrho(s)^5(1-\gamma)^{11}\epsilon^3}{\ell_F^2\gamma^2|\mathcal{S}|(|\mathcal{A}|+|\mathcal{B}|)^3}\right)$$

and hence,

$$\varepsilon_y = \Theta\left(\frac{\mu_{\mathrm{QG}}(1-\gamma)^2(\min_s \varrho(s))^2\epsilon^2}{\ell_U^4 L_U|\mathcal{S}|}\right) = \Theta\left(\frac{\min_s \varrho(s)^5(1-\gamma)^{19}\epsilon^3}{|\mathcal{S}|^{\frac{7}{2}}L_F\ell_F^4\gamma^4(|\mathcal{A}|+|\mathcal{B}|)^7}\right)$$

Now, we are ready to tune the number of inner-loop iterations, $T_y$,

$$\frac{\ell_U L_\lambda^2\ell_F\gamma^2|\mathcal{S}|^{\frac{3}{2}}(|\mathcal{A}|+|\mathcal{B}|)^4}{(\min_s \varrho(s))^6(1-\gamma)^{14}\epsilon^2}\ln\left(\frac{\ell_F L_F\gamma|\mathcal{S}|(|\mathcal{A}|+|\mathcal{B}|)}{(\min_s \varrho(s))(1-\gamma)\epsilon}\right)$$

and the batch-size, $M_y$, to be:

$$M_y = \Theta\left(\frac{\ell_F^{10}L_F^2\gamma^8|\mathcal{S}|^{10}(|\mathcal{A}|+|\mathcal{B}|)^{22}}{(\min_s \varrho(s))^{16}(1-\gamma)^{52}\epsilon^8}\right).$$

Finally,

$$T_x = \frac{L_F\ell_F^{\frac{9}{2}}\gamma^8|\mathcal{S}|^{\frac{29}{4}}(|\mathcal{A}|+|\mathcal{B}|)^{\frac{29}{2}}}{(1-\gamma)^{\frac{55}{2}}(\min_s \varrho(s))^{\frac{13}{2}}\epsilon^{\frac{7}{2}}}.$$

$\square$

### E.4.2. CMG WITH STRONGLY-CONCAVE UTILITIES

**Theorem E.11.** *Assume a two-player zero-sum cMG with a utilities that are strongly concave with modulie $\mu_1, \mu_2 > 0$. Let $\epsilon > 0$ be given. Then, a tuning of Algorithm 1 with,*

- *step-sizes, $\tau_x = \Theta\left(\frac{(\min_s \varrho(s))^3(1-\gamma)^{\frac{15}{2}}\mu_1^{\frac{3}{2}}}{\ell_F^{\frac{5}{2}}\gamma^{\frac{5}{2}}|\mathcal{S}|^{\frac{5}{2}}(|\mathcal{A}|+|\mathcal{B}|)^{\frac{15}{4}}}\right)$ and $\tau_y = \Theta\left(\frac{(1-\gamma)^3}{2\ell_F\gamma|\mathcal{S}|^{\frac{1}{2}}(|\mathcal{A}|+|\mathcal{B}|)^{\frac{3}{2}}}\right)$;*

- *exploration parameters $\varepsilon_x = \left(\frac{\epsilon(\min_s \varrho(s))^6\mu^3(1-\gamma)^{17}}{512L_F|\mathcal{S}|^4\ell_F^5\gamma^5(|\mathcal{A}|+|\mathcal{B}|)^{\frac{17}{2}}}\right)$ and $\varepsilon_y = \Theta\left(\frac{\epsilon(\min_s \varrho(s))^9\mu_x^{\frac{9}{2}}(1-\gamma)^{\frac{21}{2}}}{|\mathcal{S}|^{\frac{5}{4}}\ell_F^{\frac{3}{2}}\gamma^{\frac{1}{2}}(|\mathcal{A}|+|\mathcal{B}|)^{\frac{9}{4}}}\right)$;*

- *batch-sizes $M_x = \Theta\left(\frac{L_F^4|\mathcal{S}|^{\frac{33}{4}}\mathrm{D}_\mathcal{X}^2\ell_F^{\frac{27}{2}}\gamma^{\frac{27}{2}}(|\mathcal{A}|+|\mathcal{B}|)^{\frac{89}{4}}}{\epsilon^2(\min_s \varrho(s))^{19}\mu_x^{\frac{19}{2}}(1-\gamma)^{\frac{97}{2}}}\right)$ and $M_y = \Theta\left(\frac{4L_F^2|\mathcal{S}|^{\frac{7}{2}}\ell_F^5\gamma^3(|\mathcal{A}|+|\mathcal{B}|)^{\frac{15}{2}}}{\epsilon^2(\min_s \varrho(s))^{22}\mu_x^9\mu_y^2(1-\gamma)^{37}}\right)$;*

- *an outer-loop number of iterations at least $T_x = \Theta\left(\frac{16\sqrt{2}|\mathcal{S}|^{\frac{7}{4}}\ell_F^{\frac{7}{2}}\gamma^{\frac{7}{2}}(|\mathcal{A}|+|\mathcal{B}|)^{\frac{21}{4}}}{\min_s \varrho(s)^7\mu_x^{\frac{7}{2}}(1-\gamma)^{\frac{29}{2}}}\right)$ and inner loop iterations that are at least $T_y = \Theta\left(\frac{2\ell_F^2\gamma^2|\mathcal{S}|(|\mathcal{A}|+|\mathcal{B}|)^3}{(\min_s \varrho(s))^4(1-\gamma)^7\mu_y^2}\right)$;*

*will output an iterate $x_{t^\star}, y_{t^\star}$, such that:*

$$\mathbb{E}\left[U(x_{t^\star}, y_{t^\star+1}) - \min_{x'\in\mathcal{X}}U(x', y_{t^\star+1})\right] \le \epsilon;$$

$$\mathbb{E}\left[\max_{y'\in\mathcal{Y}}U(x_t, y') - \mathbb{E}U(x_{t^\star}, y_{t^\star})\right] \le \epsilon.$$

*Proof.* An optimality gap $\mathbb{E}\tilde{\Phi}(x_{T+1}) - \tilde{\Phi}^\star \le \epsilon$ for the utility function that is composed with the greedy exploration mapping, translates to an optimality error of the original utility function:

$$\mathbb{E}\Phi(x_{T+1}) - \Phi^\star \le \epsilon + \frac{|\mathcal{S}|^{\frac{1}{2}}L_\Phi\varepsilon_x}{|\mathcal{A}|} + \frac{|\mathcal{S}|^{\frac{1}{2}}L_\Phi\varepsilon_y}{|\mathcal{B}|} + 8|\mathcal{S}|^{\frac{1}{2}}\ell_\Phi^2\mathrm{D}_\mathcal{X}\varepsilon_x.$$

First, we can set the number of iterations to be at least,

$$T_x = \Theta\left(\frac{\ell_F \ell_\Phi \gamma |\mathcal{S}|^{\frac{1}{2}} (|\mathcal{A}| + |\mathcal{B}|)^{\frac{3}{2}}}{(\min_s \varrho(s))^4 (1-\gamma)^7 \mu_x^2}\right) = \Theta\left(\frac{16\sqrt{2}|\mathcal{S}|^{\frac{7}{4}} \ell_F^{\frac{7}{2}} \gamma^{\frac{7}{2}} (|\mathcal{A}| + |\mathcal{B}|)^{\frac{21}{4}}}{\min_s \varrho(s)^7 \mu_x^{\frac{7}{2}} (1-\gamma)^{\frac{29}{2}}}\right).$$

As such, we tune the exploration parameter $\varepsilon_x$ to be:

$$\varepsilon_x = \min\left\{\frac{|\mathcal{A}|\epsilon}{|\mathcal{S}|^{\frac{1}{2}} L_U}, \frac{\epsilon}{8|\mathcal{S}|^{\frac{1}{2}} \ell_\Phi^2 D_{\mathcal{X}}}\right\}$$

So we can set, $\varepsilon_x = \frac{\epsilon}{|\mathcal{S}|^{\frac{1}{2}} L_U \ell_\Phi^2 D_{\mathcal{X}}}$. Substituting yields:

$$\varepsilon_x = \Theta\left(\frac{\epsilon(\min_s \varrho(s))^6 \mu^3 (1-\gamma)^{17}}{L_F |\mathcal{S}|^{\frac{9}{2}} \ell_F^5 \gamma^5 (\mathcal{A} + |\mathcal{B}|)^{\frac{16}{2}}}\right) = \Theta\left(\frac{\epsilon(\min_s \varrho(s))^6 \mu^3 (1-\gamma)^{17}}{512 L_F |\mathcal{S}|^4 \ell_F^5 \gamma^5 (|\mathcal{A}| + |\mathcal{B}|)^{\frac{17}{2}}}\right)$$

$$M_x = \Theta\left(\frac{L_F^4 |\mathcal{S}|^{\frac{33}{4}} D_{\mathcal{X}}^2 \ell_F^{\frac{27}{2}} \gamma^{\frac{27}{2}} (|\mathcal{A}| + |\mathcal{B}|)^{\frac{89}{4}}}{\epsilon^2 (\min_s \varrho(s))^{19} \mu_x^{\frac{19}{2}} (1-\gamma)^{\frac{97}{2}}}\right)$$

Further, we require that $\varepsilon_y \leq \min\left\{\frac{|\mathcal{B}|\epsilon}{|\mathcal{S}|^{\frac{1}{2}} L_\Phi}, \frac{\epsilon}{8|\mathcal{S}|^{\frac{1}{2}} \ell_U^2 D_{\mathcal{Y}}}\right\}$. In order to tune $\delta_x$, we see that it is a sum of two terms, the bias of the gradient estimator and the expected distance of $y_t$ from $y^\star(x_t)$ at each iterate $t$. We see that we need to set,

$$\delta_x \leq c\frac{\sqrt{\mu_{\mathrm{qg},x}\epsilon}}{\sqrt{\ell_\Phi}}$$

for some constant $c > 0$ sufficiently small. As such, we will control the horizon of the stochastic gradient estimator, $H_x$, to be,

$$H_x = \Theta\left(\frac{1}{1-\gamma} \ln\left(\frac{\ell_F L_F \gamma |\mathcal{S}| (|\mathcal{A}| + |\mathcal{B}|)}{\epsilon(1-\gamma)\min_s \varrho(s)\mu_x}\right)\right)$$

For the inner loop, we need to ensure that $\mathbb{E}\|y_t - y^\star(x_t)\| \leq \frac{\sqrt{\mu_{\mathrm{pl},x}\epsilon}}{\sqrt{\ell_\Phi}}$. Hence, by the quadratic growth condition w.r.t. $y$, the optimality gap of the inner loop, $\epsilon_y$, needs to be bounded as:

$$\epsilon_y = \Theta\left(\frac{\epsilon\mu_{\mathrm{pl},x}\mu_{\mathrm{qg},y}}{\ell_\Phi \ell_U^2}\right) = \Theta\left(\frac{\epsilon(\min_s \varrho(s))^6 (1-\gamma)^{14} \mu_x^2 \mu_y}{\ell_F \gamma^2 |\mathcal{S}|^{\frac{3}{2}} (|\mathcal{A}| + |\mathcal{B}|)^4}\right).$$

To achieve such an optimality gap at the inner-loop on every outer-loop iteration $t$, by Claim E.5, we need to set $\varepsilon_y = \frac{\epsilon\mu_{\mathrm{pl},x}\mu_{\mathrm{qg},y}}{|\mathcal{S}|^{\frac{1}{2}} \ell_\Phi \ell_U^2}$.

$$\varepsilon_y = \Theta\left(\frac{\epsilon(\min_s \varrho(s))^9 \mu_x^{\frac{9}{2}} (1-\gamma)^{\frac{21}{2}}}{|\mathcal{S}|^{\frac{5}{4}} \ell_F^{\frac{3}{2}} \gamma^{\frac{1}{2}} (|\mathcal{A}| + |\mathcal{B}|)^{\frac{9}{4}}}\right),$$

the latter leads to the bound on $\sigma_y^2$,

$$\sigma_y^2 = \Theta\left(\frac{L_F^2 |\mathcal{S}|^{\frac{5}{2}} \ell_F^3 \gamma (|\mathcal{A}| + |\mathcal{B}|)^{\frac{9}{2}}}{\epsilon^2 (\min_s \varrho(s))^{18} \mu_x^9 (1-\gamma)^{27}}\right)$$

while, the previous bound calls for a batch-size,

$$M_y = \Theta\left(\frac{4 L_F^2 |\mathcal{S}|^{\frac{7}{2}} \ell_F^5 \gamma^3 (|\mathcal{A}| + |\mathcal{B}|)^{\frac{15}{2}}}{\epsilon^2 (\min_s \varrho(s))^{22} \mu_x^9 \mu_y^2 (1-\gamma)^{37}}\right).$$

Finally, the horizon of the inner-loop gradient estimator will be set to be:

$$H_x = \Theta\left(\frac{1}{1-\gamma}\ln\left(\frac{\ell_F L_F \gamma |\mathcal{S}|(|\mathcal{A}|+|\mathcal{B}|)}{\epsilon(1-\gamma)\min_s \varrho(s)\mu_{\mathrm{PL}}\mu_y}\right)\right),$$

while the step-size needs to be:

$$\tau_y = \Theta\left(\frac{1}{\ell_U}\right) = \Theta\left(\frac{(1-\gamma)^3}{2\ell_F\gamma|\mathcal{S}|^{\frac{1}{2}}(|\mathcal{A}|+|\mathcal{B}|)^{\frac{3}{2}}}\right),$$

and the total number of iterations needs to be at least,

$$T_y \geq \Theta\left(\frac{\ell_U}{\mu_{\mathrm{pl},y}}\ln\left(\frac{\gamma L_F \ell_F |\mathcal{S}|(|\mathcal{A}|+|\mathcal{B}|)}{\min_s \varrho(s)(1-\gamma)\epsilon}\right)\right)$$

$$= \Theta\left(\frac{2\ell_F^2\gamma^2|\mathcal{S}|(|\mathcal{A}|+|\mathcal{B}|)^3}{(\min_s \varrho(s))^4(1-\gamma)^7\mu_y^2}\right)$$

$\square$

## E.5. Alternating Policy Gradient Descent-Ascent

### E.5.1. CMG WITH CONCAVE UTILITIES

**Theorem E.12.** *Let $\epsilon > 0$ be a desired accuracy. After at most $T = \Theta\left(\frac{\ell_F^4 L_F^4 \gamma^8 |\mathcal{S}|^6(|\mathcal{A}|+|\mathcal{B}|)^{19}}{(1-\gamma)^{49}(\min_s \varrho(s))^{11}\epsilon^6}\right)$ iterations of Algorithm* 2,

- *with step-sizes, $\tau_x = \Theta\left(\frac{(\min_s \varrho(s))^9(1-\gamma)^{20}\epsilon^3}{\ell_F^2\gamma^5|\mathcal{S}|^3(|\mathcal{A}|+|\mathcal{B}|)^{\frac{17}{2}}}\right)$, and $\tau_y = \Theta\left(\frac{(1-\gamma)^8}{4\ell_F\gamma^2|\mathcal{S}|^{\frac{3}{2}}(|\mathcal{A}|+|\mathcal{B}|)^4}\right)$;*

- *exploration parameters $\varepsilon_x = \Theta\left(\frac{\min_s \varrho(s)(1-\gamma)^{12}\epsilon}{2L_F\ell_F |\mathcal{S}|^{\frac{7}{2}}\gamma(|\mathcal{A}|+|\mathcal{B}|)^{\frac{11}{2}}}\right)$ and $\varepsilon_y = \Theta\left(\frac{\min_s \varrho(s)(1-\gamma)^{23}\epsilon}{16L_F |\mathcal{S}|^{\frac{11}{2}}\ell_F^2\gamma^4(|\mathcal{A}|+|\mathcal{B}|)^{11}}\right)$;*

- *batch-sizes, $M_x = \Theta\left(\frac{L_F^4\ell_F^2|\mathcal{S}|^7(|\mathcal{A}|+|\mathcal{B}|)^{11}}{(\min_s \varrho(s))^4(1-\gamma)^{32}\epsilon^4}\right)$, and $M_y = \Theta\left(\frac{\ell_F^7\gamma^{14}|\mathcal{S}|^{\frac{31}{2}}(|\mathcal{A}|+|\mathcal{B}|)^{34}}{(\min_s \varrho(s))^{10}(1-\gamma)^{85}\epsilon^5}\right)$;*

- *sampling horizons $H_x, H_y = \Theta\left(\frac{1}{(1-\gamma)}\ln\left(\frac{\gamma\ell_F L_F |\mathcal{S}|(|\mathcal{A}|+|\mathcal{B}|)}{(1-\gamma)(\min_s \varrho(s))\epsilon}\right)\right)$,*

*there exists an iterate $t^\star$, such that*

$$\mathbb{E}U(x_{t^\star}, y_{t^\star}) - \Phi^\star \leq \epsilon;$$
$$\mathbb{E}\Phi(x_{t^\star}) - \mathbb{E}U(x_{t^\star}, y_{t^\star}) \leq \epsilon.$$

*Proof.* We invoke Theorem D.7 and tune the parameters appropriately using Lemma E.4 and claims E.5 and E.6. We tune the coefficient of the regularizer first. Needing to achieve an $(1-\gamma)(\min_s \varrho(s))\epsilon$-first order stationary point, we tune the regularizer as:

$$\mu_{\mathrm{reg.}} = \Theta\left(\frac{(1-\gamma)(\min_s \varrho(s))\epsilon}{L_F}\right).$$

We compute the pPL modulus given the regularizer tuning,

$$\mu_{\mathrm{PL}} = \Theta\left(\frac{(\min_s \varrho(s))^6(1-\gamma)^{14}\epsilon^2}{\ell_F L_F^2\gamma^2|\mathcal{S}|^{\frac{1}{2}}(|\mathcal{A}|+|\mathcal{B}|)^4}\right).$$

Further, by claims E.6 to E.8, we see that $\varepsilon_x, \varepsilon_y$ need to be tuned as,

$$\varepsilon_x = \Theta\left(\frac{(1-\gamma)(\min_s \varrho(s))\epsilon}{|\mathcal{S}|\ell_F L_F \ell_\lambda L_\lambda^4}\right) = \Theta\left(\frac{\epsilon\min_s \varrho(s)(1-\gamma)^{12}}{2L_F\ell_F |\mathcal{S}|^{\frac{7}{2}}\gamma(|\mathcal{A}|+|\mathcal{B}|)^{\frac{11}{2}}},\right)$$

and

$$\varepsilon_y = \Theta\left(\frac{\min_s \varrho(s)(1-\gamma)^{23}\epsilon}{16L_F |\mathcal{S}|^{\frac{11}{2}}\ell_F^2\gamma^4(|\mathcal{A}|+|\mathcal{B}|)^{11}}\right)$$

The resulting bounds on the variance are:

$$\sigma_x^2 = \frac{L_F^4\ell_F^2|\mathcal{S}|^7 (|\mathcal{A}|+|\mathcal{B}|)^{11}}{\epsilon^2(\min_s \varrho(s))^2(1-\gamma)^{30}},$$

and

$$\sigma_y^2 = \Theta\left(\frac{16L_F^4 |\mathcal{S}|^{11}\ell_F^4\gamma^8(|\mathcal{A}|+|\mathcal{B}|)^{22}}{\min_s \varrho(s)(1-\gamma)^{52}\epsilon^2}\right).$$

The latter dictates the batch-sizes,

$$M_x = \Theta\left(\frac{L_F^4\ell_F^2|\mathcal{S}|^7 (|\mathcal{A}|+|\mathcal{B}|)^{11}}{\epsilon^4(\min_s \varrho(s))^4(1-\gamma)^{32}}\right),$$

and

$$M_y = \Theta\left(\kappa^2\sigma_y^2\right) = \Theta\left(\frac{\ell_F^3\gamma^6|\mathcal{S}|^{\frac{9}{2}}(|\mathcal{A}|+|\mathcal{B}|)^{12}\sigma_y^2}{(\min_s \varrho(s))^9(1-\gamma)^{33}\epsilon^3}\right) = \Theta\left(\frac{\ell_F^7\gamma^{14}|\mathcal{S}|^{\frac{31}{2}}(|\mathcal{A}|+|\mathcal{B}|)^{34}}{(\min_s \varrho(s))^{10}(1-\gamma)^{85}\epsilon^5}\right)$$

The step-size is tuned straightforwardly as,

$$\tau_x = \Theta\left(\frac{(\min_s \varrho(s))^9(1-\gamma)^{20}\epsilon^3}{\ell_F^2\gamma^5|\mathcal{S}|^3 (|\mathcal{A}|+|\mathcal{B}|)^{\frac{17}{2}}}\right).$$

By Lemma E.4 we see easily that $H_x, H_y$ need to be,

$$H_x, H_y = \Theta\left(\frac{1}{(1-\gamma)}\ln\left(\frac{\gamma\ell_F L_F|\mathcal{S}| (|\mathcal{A}|+|\mathcal{B}|)}{(1-\gamma)(\min_s \varrho(s))\epsilon}\right)\right)$$

After we have tuned the regularizer, we can compute the upper bound on the number of iterations:

$$T = \Theta\left(\frac{\ell_F^4 L_F^4\gamma^8|\mathcal{S}|^6 (|\mathcal{A}|+|\mathcal{B}|)^{19}}{(1-\gamma)^{49}(\min_s \varrho(s))^{11}\epsilon^6}\right).$$

Now, tuning $\eta_y$ is also straightforward,

$$\tau_y = \Theta\left(\frac{1}{\ell_U^\mu}\right) = \Theta\left(\frac{1}{\ell_F\ell_\lambda^2 L_\lambda}\right) = \Theta\left(\frac{(1-\gamma)^8}{4\ell_F\gamma^2|\mathcal{S}|^{\frac{3}{2}}(|\mathcal{A}|+|\mathcal{B}|)^4}\right).$$

$\square$

### E.5.2. cMG WITH STRONGLY-CONCAVE UTILITIES

**Theorem E.13.** *Assume $\epsilon > 0$ and a two-player zero-sum cMG with utilities that are strongly concave with moduli $\mu_1, \mu_2$. Then if the two players are following Algorithm 2 with*

- *exploration parameters $\varepsilon_x, \varepsilon_y = \Theta\left(\frac{\epsilon(\min_s \varrho(s))^4(1-\gamma)^{15}\min\{\mu_x^2,\mu_y^2\}}{4L_F|\mathcal{S}|^{\frac{5}{2}}\ell_F^2\gamma^2(|\mathcal{A}|+|\mathcal{B}|)^4}\right)$,*

- *step-sizes, $\tau_x = \Theta\left(\frac{\mu_2^4(\min_s \varrho(s))^8(1-\gamma)^{30}}{32\ell_F^2\gamma^2|\mathcal{S}|(|\mathcal{A}|+|\mathcal{B}|)}\right)$, $\tau_y = \left(\frac{(1-\gamma)^3}{2\ell_F\gamma|\mathcal{S}|^{\frac{1}{2}}(|\mathcal{A}|+|\mathcal{B}|)^{\frac{3}{2}}}\right)$ and*

- *batch-sizes,* $M_x = \Theta\left(\frac{16L_F^4|\mathcal{S}|^{\frac{13}{2}}\ell_F^5\gamma^6(|\mathcal{A}|+|\mathcal{B}|)^{12}}{\epsilon^2(\min_s \varrho(s))^{20}(1-\gamma)^{42}\min\{\mu_x^6,\mu_y^6\}}\right)$, $M_y = \Theta\left(\frac{16L_F^4|\mathcal{S}|^{\frac{19}{2}}\ell_F^5\gamma^{10}(|\mathcal{A}|+|\mathcal{B}|)^{20}}{\epsilon^2(\min_s \varrho(s))^{28}(1-\gamma)^{50}\min\{\mu_x^{10},\mu_y^{10}\}}\right)$,

*then, it is the case that:*

$$\mathbb{E}U(x_T, y_T) - \Phi^\star \le \epsilon;$$
$$\mathbb{E}\Phi(x_T) - \mathbb{E}U(x_T, y_T)\Phi^\star \le \epsilon.$$

*after at most $T$ iterations, with $T = \Theta\left(\frac{4\ell_F\gamma^6|\mathcal{S}|^{\frac{9}{2}}(|\mathcal{A}|+|\mathcal{B}|)^{12}}{\min_s \varrho(s)^{12}(1-\gamma)^{36}\mu_1^2\mu_2^4}\log\left(\frac{L_F\ell_F\gamma|\mathcal{S}|(|\mathcal{A}|+|\mathcal{B}|)}{(\min_s \varrho(s))(1-\gamma)\mu_1,\mu_2}\right)\right)$.*

Assume the function $\tilde{U}(x, y) := U\left(w(x; \varepsilon_x, |\mathcal{A}|), w(y; \varepsilon_y, |\mathcal{B}|)\right)$ where $w(z; \varepsilon, m) := (1 - \varepsilon)z + \frac{\varepsilon\mathbf{1}}{m}$. claims E.6 and E.7 make sure that we can bound the optimality gap on the initial function $U$ by running Algorithm 2 on $\tilde{U}$. Hence, combining the aforementioned claims with Theorem D.8 and Lemma E.3, we see that we need to set:

$$\varepsilon_x, \varepsilon_y = \Theta\left(\frac{\epsilon(\min_s \varrho(s))^4(1-\gamma)^{15}\min\{\mu_x^2, \mu_y^2\}}{4L_F|\mathcal{S}|^2\ell_F^2\gamma^2(|\mathcal{A}|+|\mathcal{B}|)^4(\mathrm{D}_\mathcal{X} + \mathrm{D}_\mathcal{Y})}\right)$$

The resulting variances will be:

$$\sigma_x^2, \sigma_y^2 = \Theta\left(\frac{16L_F^4|\mathcal{S}|^4\ell_F^4\gamma^4(|\mathcal{A}|+|\mathcal{B}|)^8(\mathrm{D}_\mathcal{X} + \mathrm{D}_\mathcal{Y})^2}{\epsilon^2(\min_s \varrho(s))^8(1-\gamma)^{30}\min\{\mu_x^4, \mu_y^4\}}\right).$$

We will control the resulting variances using batches of the following size, which will also counter the $\mu_1$

$$M_x = \Theta\left(\frac{16L_F^4|\mathcal{S}|^{\frac{13}{2}}\ell_F^5\gamma^6(|\mathcal{A}|+|\mathcal{B}|)^{12}}{\epsilon^2(\min_s \varrho(s))^{20}(1-\gamma)^{42}\min\{\mu_x^6, \mu_y^6\}}\right);$$

$$M_y = \Theta\left(\frac{16L_F^4|\mathcal{S}|^{\frac{19}{2}}\ell_F^5\gamma^{10}(|\mathcal{A}|+|\mathcal{B}|)^{20}}{\epsilon^2(\min_s \varrho(s))^{28}(1-\gamma)^{50}\min\{\mu_x^{10}, \mu_y^{10}\}}\right).$$

We can easily see that the sampling horizons need to be:

$$H_x, H_y = \Theta\left(\frac{1}{1-\gamma}\ln\left(\frac{\ell_F L_F\gamma|\mathcal{S}|(|\mathcal{A}|+|\mathcal{B}|)}{\epsilon(1-\gamma)\min_s \varrho(s)\mu_x\mu_y}\right)\right)$$

The step-sizes are tuned to be:

$$\tau_x = \frac{\mu_{2,H}^4(\min_s \varrho(s))^8(1-\gamma)^{30}}{32\ell_F^2\gamma^2|\mathcal{S}|(|\mathcal{A}|+|\mathcal{B}|)}; \quad \tau_y = \frac{(1-\gamma)^3}{2\ell_F\gamma|\mathcal{S}|^{\frac{1}{2}}(|\mathcal{A}|+|\mathcal{B}|)^{\frac{3}{2}}}$$

Finally, the iteration complexity is at least:

$$T = \Theta\left(\frac{4\ell_F\gamma^6|\mathcal{S}|^{\frac{9}{2}}(|\mathcal{A}|+|\mathcal{B}|)^{12}}{\min_s \varrho(s)^{12}(1-\gamma)^{36}\mu_{H,1}^2\mu_{H,2}^4}\log\left(\frac{L_F\ell_F\gamma|\mathcal{S}|(|\mathcal{A}|+|\mathcal{B}|)}{(\min_s \varrho(s))(1-\gamma)\mu_{1,H},\mu_{2,H}}\right)\right).$$

