# OpenReview forum: "Solving Zero-Sum Convex Markov Games"
_ICML.cc/2025/Conference — ICML 2025 poster_

### Official Review · Reviewer_s9rc · 2025-02-17

**Overall Recommendation:** 3

**Summary:**

In this paper, the authors provide two (policy gradients-like) algorithms which learns $\epsilon$-Nash equilibria in convex Markov games. The authors provide bounds to the number of iterations required to compute the approximate ($\epsilon$) Nash. In order to do so, the authors leverage properties of hidden convex functions and Polyak-Lojasiewicz ones (namely, functions which satisfy the proximal Polyak-Lojasiewicz condition).

**Claims And Evidence:**

The claims seem convincing.

**Essential References Not Discussed:**

I believe that the paper is missing the following reference. I think that a comparison with [1] should be inserted in the paper.

[1] "Convex-Concave Zero-Sum Markov Stackelberg Games", Goktas et al. 2023

**Experimental Designs Or Analyses:**

I checked the soundness of the experimental design in the main paper.

**Methods And Evaluation Criteria:**

The proposed methods and evaluation criteria make sense and are intuitive.

**Other Comments Or Suggestions:**

3. Shouldn't be (GDmax) the algorithm employed in Theorem 3.1 and 3.3. ?
4. Theorem 4.6. two "with"'s in the statement.

**Other Strengths And Weaknesses:**

Overall, I believe this is a good paper. The theoretical analysis seems solid and the results are convincing. The paper is presented in an intuitive way, so that the reader can grasp the main idea behind the techniques.

Possible weakness of the work is the dependence of some parameters in the number of iterations necessary to approximate the Nash. For examples:
1. Is the dependency on $\min_s 1/\rho(s)$ necessary?
2. The $Poly$ notation hides some really bad coefficients with respect to many of the parameters in the bound (e.g. w.r.t. $1/(1-\gamma)$)

Nonetheless, I do not believe these reasons are sufficient for rejections. Thus, for now, I lean towards the acceptance of the work.

**Questions For Authors:**

See Weaknesses.

Additionally,

5. Why many of the results are proved with the $\Theta$ notation, while others (see Theorem 4.5, 4.6) with the $O$ one? Moreover, are you sure the employing the $\Theta$ notation is correct of a "sample complexity" kind of bound as the ones presented in the work?
6. Are there lower-bounds for the number of iterations required to compute the $\epsilon$-Nash?
7. I do not fully understand why (GDmax) and (Alt-GDA) have no dependency on $\delta_x,\delta_y$ in the bounds (or in the batch-sizes).

**Relation To Broader Scientific Literature:**

I think that both the techniques and the result presented in this paper would be of interest for the Game Theory community.

**Theoretical Claims:**

I did not check any proofs in details, nevertheless, the results seem convincing.

---

> ### Author Rebuttal · Authors · 2025-04-01
>
> ## Dependence on $\min_s 1/\rho(s)$.
>
> This quantity is merely a variant of the dependence on the *mismatch coefficient* [1, page 6 in the arxiv version][2]. The quantity $\min_s 1/\rho(s)$ upper bounds the mismatch coefficient when $\rho(s)>0\forall s$. The single-agent policy gradient convex MDP works also suffer this dependence on the mismatch coefficient [3,4].
>
> The main policy gradient algorithm that manages to circumvent dependence on this parameter is the natural policy gradient [1] (NPG). Yet, provable convergence for this approach is known only for single-agent conventional MDPs and not even convex ones. Further, to our knowledge, there is no natural policy gradient approach for two-player zero-sum Markov games.
>
> Concluding, dropping this dependence seems distant for two particular reasons:
> 	(a) No provable guarantees for the natural policy gradient method insingle-agent convex RL exist.
> 	(b) No provable guarantees for NPG in two-player zero-sum Markov games exist.
>
> As such, it should not be surprising that this dependence is present in our work as well
>
>
> ## Dependence on $1/(1-\gamma)$.
>
> Large dependence on the quantity $1/(1-\gamma)$ in sample complexity is to be expected as is the case in similar works, e.g., [2], where the exponent of $1/(1-\gamma)$ is $48.5$ and [5] $10 \times 4 = 40$.
>
> ## Dependence on $\delta_x,\delta_y$ and their tuning.
>
> Thank you for bringing this matter up. We ought to have been more clear on this. Checking the formal statements of our theorems, it should be apparent how convergence depends on these quantities. Namely, the $\delta$'s dictate the accuracy that is possible to get in terms of stationarity and duality gap (for convex and strongly convex utilities respectively).
>
> For the particular case of Alt-PGA and Nest-PG:
>
> * $\delta_y$ is merely the sampling bias of estimating the gradients by picking trajectory samples of fixed deterministic horizon, $H$. Yet, $\delta_y$ decays exponentially with $H$.
>
> * $\delta_x$ suffers from the same sampling bias **plus** the fact that player $1$ does not have access to the precise gradient of the regularized function. I.e., player $1$ can only estimate the gradient of the un-regularized function and suffer an error that is *bounded by the regularization coefficient's size times the upper bound of the regularizer's gradient norm*, in which case is the regularizer's Lipschitz modulus,  i.e.,
>
> $$\delta_x \leq O(\exp(-H)) + \mu_{\mathrm{reg}} L_{\mathrm{reg}}.$$
>
> For this reason we tune the regualrizers coefficient as $\mu_{\mathrm{reg}}\gets O\left(\frac{\epsilon}{L_{\mathrm{reg.}}}\right)$.
>
> ## Further Notes
>
> * (3.) Absolutely correct, it should be GDmax
> 4-5. We will fix these errors shortly. We agree that we should be using $O(\cdot)$ notation.
>
> * (6.), this is an interesting question. To the best of our-knowledege there are no lower bounds. We think that base on [6], the lower bound should not be more that $O(1/\epsilon^{3})$.
>
> * We agree that a comparison with [Goktas et al 23] should be discussed along with improvement of our related work section. Thank you for bringing up this work.
>
> -----
>
> [1] Agarwal, A., Kakade, S.M., Lee, J.D. and Mahajan, G., 2021. On the theory of policy gradient methods: Optimality, approximation, and distribution shift. JMLR
>
> [2] Daskalakis C, Foster DJ, Golowich N. Independent policy gradient methods for competitive reinforcement learning. NeurIPS 2020
>
>
> [3] Zhang, J., Koppel, A., Bedi, A.S., Szepesvari, C. and Wang, M., 2020. Variational policy gradient method for reinforcement learning with general utilities. NeurIPS 2020
>
> [4] Zhang, J., Ni, C., Szepesvari, C. and Wang, M., 2021. On the convergence and sample efficiency of variance-reduced policy gradient method. NeurIPS 2021
>
> [5] Wei, C.Y., Lee, C.W., Zhang, M. and Luo, H., 2021, July. Last-iterate convergence of decentralized optimistic gradient descent/ascent in infinite-horizon competitive Markov games. COLT 2021
>
> [6] Vavasis SA. Black-box complexity of local minimization. SIAM Journal on Optimization. 1993

---

> > ### Comment · Reviewer_s9rc · 2025-04-01
> >
> > I would like to thank the Authors for the response. I will keep my positive evaluation.

---

### Official Review · Reviewer_Jsh5 · 2025-03-09

**Overall Recommendation:** 3

**Summary:**

The paper studies two-player zero-sum convex Markov games and considers a regularization-based policy gradient approach for finding the Nash equilibrium. Two algorithms are proposed, and their complexity are provided.

**Claims And Evidence:**

I did not carefully review the paper due to a serious ethical concern. If the authors can address my concern (below), I am happy to provide a careful evaluation in the rebuttal period.

## Updated March 30

The main claims on how the two-player zero-sum cMG is abstracted by a nonconvex-nonconcave minimax optimization and the convergence guarantees of the proposed nested-loop and single-loop algorithms are sound and credible.

**Essential References Not Discussed:**

## Updated March 30
There are two important works which are currently referenced and discussed in the paper in a superficial way, Karimi et al. [2016] and Yang et al. [2020]. More discussion on the technical novelty of this work over the two papers should be made more clear.

Karimi et al. [2016] studies nonconvex (possibly non-smooth) optimization under the PL condition and draws the connections between a range of related conditions such as PL, quadratic growth, restricted secant, etc., and studies the convergence of (proximal) gradient descent. I do not find most results on page 5 unexpected as similar versions of them appeared in Karimi et al. [2016].

Yang et al. [2020] studies nonconvex-nonconcave minimax optimization under the PL condition and shows the convergence of alternating gradient descent ascent under deterministic and stochastic gradient oracles.

**Ethical Review Concerns:**

The paper contains a large number of non-existent references. This raises a potential concern for LLM-generated content which has not been carefully reviewed by the authors.

**Ethical Review Flag:**

Flag this paper for an ethics review.

**Ethics Expertise Needed:**

["Research Integrity Issues (e.g., plagiarism)"]

**Experimental Designs Or Analyses:**

N/A.

**Methods And Evaluation Criteria:**

## Updated March 30

The convergence metric ($\epsilon$-NE introduced in Definition 2) is a standard choice in the literature and makes sense.

**Other Comments Or Suggestions:**

In the theorem statements in Section 4, the authors should consider specifying order-wise how to choose $\varepsilon_x,\varepsilon_y,\tau_x,\tau_y$ as functions of the desired precision $\epsilon$.

Regarding the simulations,
1) It would be nice to see the single-loop algorithm compared against the nested-loop one.
2) Is the rock-paper-scissors-dummy problem a convex, nonlinear MG or is it actually linear?

More discussion of Theorem 4.3-4.6 is needed. Can the authors clarify if my understanding is correct: when the problem itself is not hidden strongly concave, a regularization on the order of the desired precision needs to be added to make it so. If this is true, why does the statement of Theorem 4.3 not involve $\mu$?

I find it hard to locate the proof of Theorem 4.1. Can the authors point out where it is in the appendix?

I do not understand the discussion of the technical challenge in line 055-062 (first column on page 2). At least in the linear utility setting, it is clear that we can define the aggregate value functions, which satisfies a fixed point equation involving a contractive operator. See Perolat et al. [2015]. Value-iteration-type methods can be used to find the fixed point. I suppose something similar should exist in the general convex setting as well.

References

Perolat, J., Scherrer, B., Piot, B. and Pietquin, O., 2015, June. Approximate dynamic programming for two-player zero-sum Markov games. In International Conference on Machine Learning (pp. 1321-1329). PMLR.

**Other Strengths And Weaknesses:**

There are a large number of non-existent references, including but not limited to

"Learning in markov games: Algorithms and guarantees",

"Decentralized learning in stochastic games: Convergence to coarse correlated equilibria",

"Convergence to cce in multi-agent markov games",

"An empirical study on offline multi-agent reinforcement learning",

"Fast convergence of equilibria in stochastic games".

This raises a concern for potential misbehavior.

## Updated March 30

On the positive side, the paper is very clearly written in the methodology. The development of technical results is easy to follow and provides a tutorial value in that aspect. The technical claims are sound and credible.

My main concern is that the paper seems to lack important technical contributions over the prior works I mentioned above (especially Yang et al. [2020]). The difference between this paper and Yang et al. [2020] that I can see is 1) Yang et al. [2020] studies the unconstrained case, while this paper studies the constrained problem and the algorithm makes projections to the constrained set. However, making this extension should be straightforward; 2) this paper models the zero-sum cMG problem in the minimax optimization framework, which also seems technically insignificant. I encourage the authors to correct me if my understanding is incorrect. In any case, more discussion on the exact technical contributions over the prior work should be made in the introduction section.

**Questions For Authors:**

N/A

**Relation To Broader Scientific Literature:**

The pPL condition may come up in other problems beyond zero-sum cMGs, in which the analysis of gradient-descent-ascent-based algorithms presented in this paper can provide insight.

**Theoretical Claims:**

## Updated March 30

I checked the mathematical claims in the main paper and part of the appendix. I do not see any major errors and find the claims solid and credible.

---

> ### Author Rebuttal · Authors · 2025-04-01
>
> Dear Reviewer,
>
> Thank you for your time and effort in reviewing the paper and pointing out this mistake in our bibliography. We apologize for our negligent mistake, thankfully, the wrong LLM-generated bib items are limited to the related work sec of Appendix B. As we clarified to the AC, all hallucinated references correspond to real citations. We were able to communicate a list of them to the AC but due to space constraints we cannot include them here.
>
> Before proceeding to address your concerns, we want to point out that our contributions are the following:
> * Designing finite-time finite-sample algorithms that converge to saddle-points in constrained nonconvex-pPL, pPL-pPL objectives (Sec 3).
> * Using the latter to design policy gradient algorithms that converge to Nash equilibria (NE) in cMGs (Sec. 4).
> * Our work is theoretical but our numerical experiments strengthen our claims. Although RPS is initially "linear", after regularization it is no different than a generic cMG.
>
> *We do not claim* to be (i) coining the proximal-PL condition, neither that (ii) formulating the NE as a min-max optimization problem is our contribution.
>
> Let us elaborate on two points that we think are critical in demonstrating the technical hurdles we needed to overcome.
>
> ## Bellman equations invalid
>
> As pointed out in convex MDP literature, the Bellman equations, ($V(s) = \max_{a}\[ r(s,a) + \gamma \mathbb{E}_{s'} P(s'|s,a)V \] $) , fail to hold. This is due to the non-additivity of rewards. The total utility is a convex function wrt the occupancy measure and value is not generally defined state-wise (e.g. the entropy of the state occupancy measure). See, e.g., introduction of [Zhang et. al, 2020, Intro - 3rd paragraph ]. The original contraction argument of (Shapley 53) which subsumes additivity cannot work. All these rule out the possibility that some value-iteration scheme can work.
>
> ## From unconstrained to constrained AGDA
>
> We needed substantial work to extend to the constrained case. It is not always true that incorporating a projection step does not require significant modification of the proof. One might get that idea when comparing the proof of convergence of gradient descent (GD) vs. projected-GD in nonconvex smooth optimization. Two facts make the latter straightforward for nonconvex smooth minimization case: (1) the Lyapunov function used to prove convergence is merely the objective function itself and the (2) the projection operator is 1-Lipschitz continuous.
>
> Nonetheless, in proving convergence of alternating gradient descent ascent for the challenging min-max objective, the Lyapunov function is a weighted sum of the two individual optimality gaps.
>
> For the unconstrained case [1,2], proving that the Lyapunov function progressively decreases revolves around manipulating the quantities $\nabla_x f(x,y), \nabla_x \Phi(x),$ and $\nabla_x f(x,y)$. These quantities are very easy to work with due to their Lipschitz continuity which follows from the assumptions.
>
> In the constrained case, proving progressive decrease in the Lyapunov function requires working with the gradient mapping $\| x - \Pi(x - \eta \nabla_x f(x,y)\|$ and the quantities $D_{X}^f, {D}_{X}^{\Phi}, D_Y$. The same assumptions that we make in the unconstrained case, do not make it easy to appropriately manipulate these quantities in order to show convergence.
>
> Without diving deeper, it is not obvious for example why $D_{\mathcal{X}}(x,a;y)$ should satisfy the following inequality,
> $$|D_{\mathcal{X}}(x, a; y) − D_{\mathcal{X}}(x, a; y)| ≤ 3\ell^2 \| y − y′\|^2 ;$$
> especially when it is defined as an optimization problem in itself: $$D_{\mathcal{X}}(y,a;x) := -2a \min_{y'} \{ \langle -\nabla_y f(x,y) , y' - y \rangle  + \frac{a}{2}\|y-y'\|^2 \}.$$
>
> Please, notice that $D_{\mathcal{X}}$ appears with an exponent of $1$ while $ \| y − y′\|$ with an exponent of $2$. Instead, en route for our convergence proof, we had to prove Claims C.10 through C.13. We have not encountered these claims in previous min-max optimization literature before.
>
>
> ## Further concerns
>
> >  when the problem itself is not hidden strongly concave, a regularization [...] needs to be added to make
>
> You are correct and we should provide the regularization's order. For the time being, please see Thm. F.10 (formal version of Thm. 4.3) for a precise tuning. $\mu$ is tuned at line 2945.
>
> > Proof of Theorem 4.1
>
> Please refer to its formal version Theorem E.1.
>
>
>
>
> [1] Yang, J., Kiyavash, N., and He, N. Global convergence and variance reduction for a class of nonconvex-nonconcave minimax problems. NeurIPS 2020.
>
> [2] Yang, J., Orvieto, A., Lucchi, A., and He, N. Faster single-loop algorithms for minimax optimization without strong
> concavity. AISTATS 2022.

---

> > ### Comment · Reviewer_Jsh5 · 2025-04-01
> >
> > I thank the authors for the clarification, especially on the technical challenges, which I did not understand. My rating is adjusted.

---

> > > ### Author Response · Authors · 2025-04-04
> > >
> > > Dear Reviewer,
> > >
> > > We are happy to know that we adequately addressed your concerns. We would like to additionally let you know that we ran some additional experiments to compare Alt-PGDA against Nest-PG, encouraged by your suggestion. We will add this comparison to our manuscript.
> > >
> > > Across experiments and compared against Alt-PGDA, Nest-PG enjoys smaller variance of the exploitability for the same number of outer-loop iterations. More inner-loop iterations further decrease the exploitability variance accross experiments. Nevertheless, this comes to the expense of more inner-loops iterations making Alt-PGDA more attractive for practical applications.
> > >
> > > Thank you.

---

### Official Review · Reviewer_4bBk · 2025-03-12

**Overall Recommendation:** 4

**Summary:**

This paper addresses global convergence to Nash equilibria in two-player zero-sum convex Markov games (cMGs)—a recently introduced framework generalizing Markov decision processes by allowing convex utilities over occupancy measures. The main contribution is proving that independent policy gradient algorithms, with a specialized regularization for stability, converge to ϵ-Nash equilibria in polynomial time. The authors present two methods—Nested Policy Gradient (Nest-PG) and Alternating Policy Gradient Descent-Ascent (Alt-PGDA)—and provide both theoretical guarantees and a simple empirical demonstration on an iterated Rock-Paper-Scissors gam

**Claims And Evidence:**

Key claims: (1) These are the first algorithms with global convergence guarantees for zero-sum cMGs, and (2) the proposed regularization induces a structure (proximal PL) that stabilizes independent learning. The paper offers solid theoretical proofs (theorem 4.3-4.5 and 4.1 respectively) and an experiment showing that exploitability decreases as expected.

**Essential References Not Discussed:**

No

**Experimental Designs Or Analyses:**

The single experiment on iterated RPS is limited but indicative. It confirms the predicted behavior: with regularization, exploitability drops and remains near a small positive threshold determined by the regularization parameter.

**Methods And Evaluation Criteria:**

The authors use policy gradient with a custom occupancy-based regularization to ensure smooth best responses. The design is coherent with the cMG problem structure and empirical success of PG based methods. The evaluation focuses on reaching
ϵ-Nash equilibrium and analyzing the iteration/sample complexity, which matches common MARL metrics.

**Other Comments Or Suggestions:**

No.

**Other Strengths And Weaknesses:**

The presentation is clear and easy to follow. Though I am not very familiar with this topic, the structure, starting with basics to their contribution guided me smoothly to the main point of this paper.

**Questions For Authors:**

Do you anticipate straightforward extensions to general-sum cMGs or more than two players?

**Relation To Broader Scientific Literature:**

The paper's main contribution theoretically supports the empirical success of PG based methods in MARL literature.

**Theoretical Claims:**

The authors prove that, under regularization, best responses become Lipschitz in opponent policies. They then establish convergence to ϵ-Nash equilibria via proximal PL arguments. The proofs look consistent and are comparable to known gradient-based results under nonconvex–nonconcave settings.

---

> ### Author Rebuttal · Authors · 2025-04-01
>
> Thank you for your positive reception of the paper. We are glad that you recognize our technical contributions that we deem important in broadening our understanding of multi-agent RL and nonconvex optimization.
>
> Regarding your question **''Do you anticipate straightforward extensions to general-sum cMGs or more than two players?''**,
> the initial paper [1] defining the setting, in fact defines cMGs for any number of agents and general utilities. So yes, there is a straightforward extension. A very interesting avenue of future work is finding nontrivial families of games where equilibrium computation is computationally feasible.
>
> Again, thank you and let us know if you have further concerns that we could address.
>
> [1] Gemp I, Haupt A, Marris L, Liu S, Piliouras G. Convex Markov Games: A Framework for Creativity, Imitation, Fairness, and Safety in Multiagent Learning.

---

### Official Review · Reviewer_9QVE · 2025-03-13

**Overall Recommendation:** 3

**Summary:**

The paper studies convergence of policy gradient methods in zero-sum convex Markov games, giving the first convergence result to $\epsilon$-Nash equilibria for such games. The approach uses the inherent hidden convex-concave structure present with respect to the occupancy measures. This structure is formalized the proximal PL condition, and is connected with other known assumptions such as quadratic growth. With the hidden structure and added regularization, two general policy gradient methods are studied, double loop nested PG method and alternating gradient descent-ascent method.
Complexity analysis is thoroughly given in the general context, min-max with hidden structure, and then linked back to the problem of convex zero-sum Markov games.

**Claims And Evidence:**

Claims are mostly theoretical, see the appropriate section below.

**Essential References Not Discussed:**

Due to the relatively new setting of convex Markov games, I am not aware of essential references that should be included.

**Experimental Designs Or Analyses:**

The experiment  used to test the theory is limited to a toy repeated RPS example. However, due to the theoretical focus of the paper this is a minor point.

**Methods And Evaluation Criteria:**

N/A

**Other Comments Or Suggestions:**

- what does it mean for the BR mapping to be convex? isn't this a set valued mapping?
- Def 1 seems a bit off, the way it is written suggests that we are asking $F$ to be concave over the cross-product of occupancy measures? I think what is meant is that it is convex-concave? for example this would not hold in a matrix game e.g. x*y is convex/concave but not concave in $(x,y)$.
- is Nest-PG the same as GDmax? Nest PG is only defined much later in section 4.1 after it is first referred to in earlier Theorems.
- what is a $\mu$-modulus transformation?(280)

**Other Strengths And Weaknesses:**

Strengths
The paper makes some fundamental observations regarding the structure of convex Markov games in connections to well-known non-convex optimization concepts, and uses these general assumptions to provide new guarantees in a new class of hidden convex-concave games. These two main contributions seem both important in their own right, i.e. for understand convex Markov games, and understanding what is useful structure in hidden convex-concave problems.

Weaknesses
The requirement of regularization seems to be a weakness albeit a minor one. It is well-known that regularization in games can utilized to establish convergence of gradient descent-ascent methods (i.e Nesterov smoothing, or entropy regularization in quantal response equilibria), however, I would not interpret those results giving an affirmative answer to whether policy gradient methods converge in min-max problems. I think most would agree that policy gradient methods can diverge or cycle in convex-concave games as it is assumed there is no additional curvature structure (i.e. regularization). Therefore I find the claim a bit strong to that the methods proposed in the paper show that policy gradient methods can converge in zero-sum convex Markov games.

**Questions For Authors:**

1. In Lemma 2.5 it is unclear how the constants $\mu_c, D_{\mathcal{U}}$ in Proposition 2.4 correspond to the constants in Lemma 2.5 where the proposition is applied. Is there an assumption missing? maybe one needs to apply claim 2.3? More precisely, where does constant $\frac{(1-\gamma)\min_s{\rho(s)}}{2\sqrt{2}}$ come from in Lemma 2.5?

**Relation To Broader Scientific Literature:**

The paper seems to be well placed within the broader scientific literature. It is much appreciated the effort that the authors have done through to connect results to the general optimization literature. There can be some more references included to better complete background and make the paper more approachable. For example, references for the proximal PL condition and Quadratic growth would be helpful. I believe the proximal PL condition is defined in:
Karimi, H., Nutini, J., & Schmidt, M. (2016, September). Linear convergence of gradient and proximal-gradient methods under the polyak-łojasiewicz condition. In Joint European conference on machine learning and knowledge discovery in databases (pp. 795-811). Cham: Springer International Publishing.

The approach also seems related to the idea of Nesterov smoothing, especially for the nested GD method, where a min-max problem can be turned into a smooth convex problem if you smooth out the best response of the max player. Some discussion in connection to this technique could be helpful/insightful.

**Theoretical Claims:**

The theoretical claims seem reasonable. I was not able to check all the claims in the paper. I do have some questions on some of the results that are not clear.

## Lemma 2.5
In Lemma 2.5 it is unclear how the constants $\mu_c, D_{\mathcal{U}}$ in Proposition 2.4 correspond to the constants in Lemma 2.5 where the proposition is applied. Is there an assumption missing? maybe one needs to apply claim 2.3? More precisely, where does constant $\frac{(1-\gamma)\min_s{\rho(s)}}{2\sqrt{2}}$ come from in Lemma 2.5?

## The the tuning of the bias $\delta_x, \delta_y$

The first order stochastic oracle model from line 325 specified potential systematic error (bias) in the gradients, however it is unclear how this irreducible bias is present in the results within the main paper. For example Theorems 3.1-3.4 specify tuning for stepsizes and batchsizes without mentioning how the bias error affects the convergence. Looking at the appendix it is mentioned that this bias needs to be tuned (e.g. line 1908). As expected there should be an neighbourhood of convergence that depends on this $\delta$ if it is not tuned. If tuned this should be stated in the main body of the paper, i.e. what upper bound is needed on the bias to attain the complexity guarantees presented in the paper similar to how the stepsizes and batchsizes are picked.

---

> ### Author Rebuttal · Authors · 2025-04-01
>
> We thank the reviewer for their positive reception of our work and recognizing the technicality and contributions of our paper. Allow us to address your concerns.
>
>
> ## Policy gradient terminology
>
> * Given that literature labels as "policy gradient methods" most algorithms that use a gradient wrt the policy in order to maximize, we believe that we are not abusing the term.
>
> * Indeed, as we point out in lines 76-100 left col., simply running a directly paramatrized policy gradient descent ascent, would not work. For this purpose we use alteration in gradient updates and regularization that is ubiquituous in optimization.
>
> ## The the tuning of the bias $\delta_x, \delta_y$:
>
> We are glad you bring up this detail we overlooked clarifying. Checking the formal statements of our theorems, it should be apparent how convergence depends on these quantities. Namely, the $\delta$'s dictate the accuracy that is possible to get in terms of stationarity and duality gap (for convex and strongly convex utilities respectively).
>
> For the particular case of Alt-PGA and Nest-PG:
>
> * $\delta_y$ is merely the sampling bias of estimating the gradients by picking trajectory samples of fixed deterministic horizon, $H$. Yet, $\delta_y$ decays exponentially with $H$.
>
> * $\delta_x$ suffers from the same sampling bias **plus** the fact that player $1$ does not have access to the precise gradient of the regularized function. I.e., player $1$ can only estimate the gradient of the un-regularized function and suffer an error that is *bounded by the regularization coefficient's size times the upper bound of the regularizer's gradient norm*, in which case is the regularizer's Lipschitz modulus,  i.e.,
>
> $$\delta_x \leq O(\exp(-H)) + \mu_{\mathrm{reg}} L_{\mathrm{reg}}.$$
>
> For this reason we tune the regualrizers coefficient as $O\left(\frac{\epsilon}{L_{\mathrm{reg.}}}\right)$.
>
>
> ## Further Notes
>
> * What we meant to say is that the BR-mapping maps to a set of policies that is convex.
>
> * Lipschitz modulus in Lemma 2.5:
>
>   (i) $\mathcal{D}_{\mathcal{U}}$ is the Euclidean diameter of the state-action occupancy measure. Since it is a simplex, the diameter is $\sqrt{2}$.
>
>   (ii) Then, $\mu_c$ is the inverse of the Lipschitz modulus of the transform from occupancy measures to policies and it is bounded by $\frac{2}{(1-\gamma \min_{s})\rho(s) }$. [1; Lemma C.3]
>
> ---
>
> Thank you,
>
> Please let us know in case you have further concerns.
>
> ---
>
> [1] Kalogiannis, F., Yan, J. and Panageas, I., 2024. Learning Equilibria in Adversarial Team Markov Games: A Nonconvex-Hidden-Concave Min-Max Optimization Problem. NeurIPS 2024.

---

> > ### Comment · Reviewer_9QVE · 2025-04-06
> >
> > Thank you for the clarifications.
> >
> > ## Constants in Lemma 2.5:
> > Looking at the results again after the explanation, I see that my confusion comes from the fact that Claim 2.3 (or more precisely Lemma 2.1) is used in proving Lemma 2.5 but is never actually invoked in the proof. To me this was difficult to follow, I would suggest explicitly referencing the earlier results in the use of the constants $\mu_c, D_{\mathcal{U}}$.
> >
> > ## Structure of bias in Convex Markov games:
> > Thank you for the clarification. It would be nice to have a comment on how the bias is automatically tuned via the horizon and regularization constant. Otherwise the results seems a bit suspicious. Adding this comment would also provide more motivation when introducing the oracle assumptions (line 325).
> >
> > ## Bias in the main theorem:
> > It makes sense to me that the bias has more structure in the specific context of convex Markov games, however, in the general results (Theorem 3.1-3.2) there is no Markov game context. It is a bit weird to me that the tuning conditions of the bias has been left out in the theorem when the stepsize and batchsize conditions are included. Why not include the bias condition as well? For example the condition in lines 1908-1910.

---

> > > ### Author Response · Authors · 2025-04-06
> > >
> > > Dear Reviewer,
> > >
> > > We are glad we were able to address your concerns. Thank you for your careful review and thoughtful comments. We will incorporate your suggestions in order to improve the text of our manuscript.
> > >
> > > We will clarify the points about the bias error to reflect its (i) effect on convergence, (ii) the fact that our algorithmic design that automatically tunes it in cMGs, and (iii) make clearer that it is the stochastic extension of the inexact gradient oracle [1].
> > >
> > >
> > > ----
> > > **Additional comments:**
> > >
> > > We also meant to address the rest of your concerns. By mistake we did not include them in the first rebuttal.
> > >
> > > * Regarding regularity conditions: *There can be some more references included to better complete background and make the paper more approachable.*
> > >
> > > We have included some references in the appendix section D, but we are more than happy to elaborate on this point.
> > >
> > > * *It is well-known that regularization in games can utilized to establish convergence of gradient descent-ascent methods*
> > >
> > > Although it is well known that regularization leads to convergence in games, all known results have to do with monotone games (a multi-player generalization of convex-concave), Markov games with value functions that are additive in the rewards, extensive-form games in the sequence form (i.e., convex-concave games).
> > >
> > > We provide a more general result for convergence through generalization for any zero-sum game over constrained domains with utility functions that are concave for each player.
> > >
> > > * *Connection to Nesterov smoothing*
> > >
> > > This is a nice point and it is indeed worth it including a note on our text. In our case, we introduce a nonconvex and PL regularizer (wrt policies) that induces the PL condition on the perturbed problem,  whereas the former work adds a strongly-convex regularizer to smoothen out the problem.
> > >
> > > * *Def 1 seems a bit off*
> > >
> > > Although the definition is not wrong you are right that it is a bit convoluted. A clearer equivalent definition would be to say that each player's utility is a concave function of their individual state-action occupancy measure when all other players' policies are fixed. We do not call for the utility to be jointly convex.
> > >
> > > A good example would be the maximum entropy of the state occupancy measure where one player is trying to explore the state space by maximizing the state-occupancy while the second player is trying to prevent exploration of the whole state-space by minimizing the entropy.
> > >
> > > * *is Nest-PG the same as GDmax?*
> > >
> > > This is a typo. Thank you for bringing this up. Indeed, in section 3, GDmax should take the place of Nest-PG.
> > >
> > > * *what is a $\mu$-modulus transformation?*
> > >
> > > What we meant by that is that the modulus of hidden convexity, $\mu$, is equivalent to a modulus of the proximal-PL condition, $\mu'$, with $\mu' = \mathrm{poly}(\mu)$. I.e., it transformed to a polynomial of the former. We will clarify this point.
> > >
> > > ---
> > >
> > > To conclude, **thank you** for your detailed comments and we are hopeful to have sufficiently addressed all of your concerns.
> > >
> > > ---
> > > [1] Devolder, O., Glineur, F. and Nesterov, Y., 2014. First-order methods of smooth convex optimization with inexact oracle. Mathematical Programming.

---

### Decision · Program_Chairs · 2025-05-01

**Decision:**

Accept (poster)

**Comment:**

This paper studies the convergence of policy gradient methods in zero-sum convex Markov games to an approximate Nash equilibrium. The key technique is to leverage the hidden convexity of the objective in terms of the occupancy measure, together with the use of regularization. The theoretical results are solid and reasonable, albeit many techniques are adopted from the literature of minimax optimization and recent development of regularized policy optimization in RL, weakening the technical novelty of the results. There were also some comments on the presentation and clarity of the paper. Overall, the new model and results are still of merit to the multi-agent RL literature. I recommend that the authors incorporate the feedback from the reviewers in preparing the camera-ready version of the paper.